# Corruption-Robust Linear Bandits: Minimax Optimality and Gap-Dependent Misspecification

**Haolin Liu**[*]
University of Virginia
srs8rh@virginia.edu

**Artin Tajdini**
University of Washington
artin@cs.washington.edu

**Andrew Wagenmaker**
University of California, Berkeley
ajwagen@berkeley.edu

**Chen-Yu Wei**
University of Virginia
chenyu.wei@virginia.edu

## Abstract

In linear bandits, how can a learner effectively learn when facing corrupted rewards? While significant work has explored this question, a holistic understanding across different adversarial models and corruption measures is lacking, as is a full characterization of the minimax regret bounds. In this work, we compare two types of corruptions commonly considered: *strong corruption*, where the corruption level depends on the learner's chosen action, and *weak corruption*, where the corruption level does not depend on the learner's chosen action. We provide a unified framework to analyze these corruptions. For stochastic linear bandits, we fully characterize the gap between the minimax regret under strong and weak corruptions. We also initiate the study of corrupted adversarial linear bandits, obtaining upper and lower bounds with matching dependencies on the corruption level.

Next, we reveal a connection between corruption-robust learning and learning with *gap-dependent misspecification*—a setting first studied by Liu et al. (2023a), where the misspecification level of an action or policy is proportional to its suboptimality. We present a general reduction that enables any corruption-robust algorithm to handle gap-dependent misspecification. This allows us to recover the results of Liu et al. (2023a) in a black-box manner and significantly generalize them to settings like linear MDPs, yielding the first results for gap-dependent misspecification in reinforcement learning. However, this general reduction does not attain the optimal rate for gap-dependent misspecification. Motivated by this, we develop a specialized algorithm that achieves optimal bounds for gap-dependent misspecification in linear bandits, thus answering an open question posed by Liu et al. (2023a).

## 1  Introduction

The real world is rarely truly stochastic—in practice, our observations are often corrupted—and furthermore, rarely are the modeling assumption typically made in theory—that the true data-generating process lives in our model class—met in reality. Therefore, robustly handling these deviations from idealized assumptions is crucial. These challenges are particularly pronounced in interactive decision-making settings, where deviations from idealized assumptions could lead an algorithm to take unsafe or severely suboptimal actions. In this work, we seek to address these challenges, and develop a unified understanding for robust learning in corruption-robust and misspecified settings.

---

[*]Authors are listed in alphabetical order by last name.

38th Conference on Neural Information Processing Systems (NeurIPS 2024).

We first consider the corruption-robust learning setting. Robust learning in the presence of corruptions requires designing algorithms whose guarantee have a tight scaling in the corruption level. That is, although some amount of suboptimality is inevitable if our observations are corrupted, we would hope to obtain the minimum amount of suboptimality possible at a given corruption level. While much work has been done on learning with corrupted observations, existing work has failed to yield a tight characterization of this scaling in the corruption level, even in simple settings such as linear bandits. We address this shortcoming, and develop an algorithm which achieves the optimal scaling in the corruption level, and further extend this to a novel corrupted adversarial linear bandit setting, where in addition to corrupted observations, the rewards themselves may be adversarially chosen from round to round. We obtain the first provably efficient bounds in this setting.

Model misspecification, another extensively studied problem in the literature, can be thought of as a form of corruption, where the corruption level is the amount of misspecification between the "closest" model in the model class and the true environment. Standard discussions on misspecification usually assume that the misspecification for every action has a uniform upper bound, and the final regret guarantee scales linearly with the amount of misspecification. The work of Liu et al. (2023a) initiated the study on the *gap-dependent misspecification* setting, where the misspecification level for a given action scales with the suboptimality of that action. They demonstrated that the linear scaling in regret is not necessary in this case. We revisit this problem, and show a general reduction from the gap-dependent misspecified setting to the corruption setting. We utilize this reduction to show that settings previously not known to be learnable—for example, linear MDPs with policy gap-dependent misspecification—are in fact efficiently learnable with existing corruption robust algorithms.

Together, our results present a unified picture of optimally learning in the presence of observation corruption, and (certain types of) model misspecification. We summarize our contributions as follows (see Section 2 and Section 3 for formal definitions of the mentioned quantities):

1. In Section 4, we develop a stochastic linear bandit algorithm with $\widetilde{\mathcal{O}}(d\sqrt{T} + \min\{dC, \sqrt{d}C_\infty\})$ regret, where $d$ is the feature dimension, $T$ is the number of rounds, $C$ is the strong corruption measure, and $C_\infty$ is the weak corruption measure. These bounds are unimprovable.
2. In Section 5, we initiate the study of adversarial linear bandits with corruptions. We obtain $\widetilde{\mathcal{O}}(d\sqrt{T} + \sqrt{d}C_\infty)$ and $\widetilde{\mathcal{O}}(\sqrt{d^3T} + dC)$ regret for weak and strong corruptions, respectively.
3. We prove a general reduction that efficiently handles gap-dependent misspecification with corruption-robust algorithms. We apply our reduction to show that linear MDPs with gap-dependent misspecification are efficiently learnable (Section 6).
4. Finally, while the reduction in item 3 is general, it is unable to obtain the tightest possible rate for gap-dependent misspecification. We thus develop a specialized algorithm which, in the linear bandit setting, obtains the optimal rate. This resolves the open problem of Liu et al. (2023a).

In Section 2 we present our problem setting, and in Section 3, compare the corruption notions in previous and our work. More related works are discussed in Appendix A. In Section 4–Section 6, we present our main results as outlined above.

## 2 Problem Setting and Preliminaries

We consider the corrupted linear bandit problem. The learner interacts with the environment for $T$ rounds. The learner is given an action set $\mathcal{A} \subset \mathbb{R}^d$. At the beginning of round $t$, the environment determines a reward vector $\theta_t \in \mathbb{R}^d$ and a corruption function $\epsilon_t(\cdot) : \mathcal{A} \to [-1, 1]$, which are both hidden from the learner. The learner then selects an action $a_t \in \mathcal{A}$. Then a reward value $r_t = a_t^\top \theta_t + \epsilon_t(a_t) + \zeta_t$ is revealed to the learner, for some zero-mean noise $\zeta_t \in [-1, 1]$[2]. We assume that $\|a\|_2 \le 1$, $\|\theta_t\|_2 \le \sqrt{d}$, and $a^\top \theta_t \in [-1, 1]$ for any $a \in \mathcal{A}$ and any $t = 1, 2, \ldots, T$. We define $\epsilon_t = \max_{a \in \mathcal{A}} |\epsilon_t(a)|$.

In the stochastic setting, the environment is restricted to choose $\theta_t = \theta^\star$ for all $t$, while in the adversarial setting, $\theta_t$ can arbitrarily depend on the history up to round $t - 1$. The regret of the learner is defined as

$$\text{Reg}_T = \max_{u \in \mathcal{A}} \sum_{t=1}^T u^\top \theta_t - \sum_{t=1}^T a_t^\top \theta_t.$$

---

[2]We assume both the corruption function and the noise are bounded for simplicity. All our results can be generalized to the case where the corruption is unbounded and the noise is sub-Gaussian. See the "additional note on corruption" in Page 5 of Wei et al. (2022) for reducing this case to the bounded case.

Note that although the non-stationarity of $\theta_t$ in the adversarial setting captures a certain degree of corruption, this form of corruption is limited to a linear form $a^\top(\theta_t - \theta^\star)$, which is not as general as $\epsilon_t(a)$ that could be an arbitrarily function. Therefore, the corrupted linear bandit problem cannot be reduced to an adversarial linear bandit problem.

**Notation.** We denote $[n] = \{1, 2, \ldots, n\}$. Let $\Delta(\mathcal{A})$ be the set of distribution over $\mathcal{A}$. For any $p \in \Delta(\mathcal{A})$, define the lifted covariance matrix $\widehat{\mathrm{Cov}}(p) = \mathbb{E}_{a \sim p} \begin{bmatrix} aa^\top & a \\ a^\top & 1 \end{bmatrix} \in \mathbb{R}^{(d+1) \times (d+1)}$. For $A, B \in \mathbb{R}^{d \times d}$, define $\langle A, B \rangle = \mathrm{Tr}(AB^\top)$. $\mathbb{E}_t[\cdot]$ is the expectation conditioned on history up to $t - 1$.

**G-Optimal Design.** A G-optimal design over $\mathcal{A}$ is a distribution $\rho \in \Delta(\mathcal{A})$ such that $\|a\|^2_{G^{-1}} \leq d$ for all $a \in \mathcal{A}$, where $G = \sum_{a \in \mathcal{A}} \rho(a) aa^\top$. Note that such a distribution is guaranteed to exist, and can be efficiently computed (Pukelsheim, 2006; Lattimore and Szepesvári, 2020).

## 3  Two Equivalent Views: On Adversary Adaptivity and Corruption Measure

Previous works have studied corruption with various assumptions on the adaptivity of the adversary and different measures for the corruption level. In this work, we consider both the *strong* and *weak* guarantees, which can cover different notions of corruptions studied in previous works. We provide two different viewpoints to understand them. In the first viewpoint, the weak and strong guarantee differ by the *adaptivity* of the adversary, while in the second viewpoint, the two guarantees differ in the *measure of corruption*. Then we argue that the two viewpoints are equivalent.

**Adversary Adaptivity (AA) Viewpoint.** In this viewpoint, the corruption is specified only for the *chosen* action. That is, in each round $t$, the adversary only decides a single corruption level $\epsilon_t \in \mathbb{R}_{\geq 0}$ and ensures $|\mathbb{E}[r_t] - \langle a_t, \theta^\star \rangle| \leq \epsilon_t$. We consider two kinds of adversary: strong adversary who decides $\epsilon_t$ *after* seeing the chosen action $a_t$, and weak adversary who decides $\epsilon_t$ *before* seeing $a_t$. The robustness of the algorithm is measured by how the regret depends on $\sum_{t=1}^T \epsilon_t$.

**Corruption Measure (CM) Viewpoint.** In this viewpoint, the corruption is individually specified for *every* action. That is, at each round $t$, the adversary decides $\epsilon_t(a)$ for all action $a \in \mathcal{A}$ and ensures $\mathbb{E}[r_t | a_t = a] - \langle a, \theta^\star \rangle = \epsilon_t(a)$ for all $a$. The adversary always decides $\epsilon_t(\cdot)$ *before* seeing $a_t$. To evaluate the performance, we consider two different measures of the total corruption: the strong measure $\sum_{t=1}^T |\epsilon_t(a_t)|$ and the weak measure $\sum_{t=1}^T \max_{a \in \mathcal{A}} |\epsilon_t(a)|$.

We argue that the two viewpoints are equivalent in the sense that the performance guarantee of an algorithm under strong/weak adversary in the AA viewpoint are the same as those under strong/weak measure in the CM viewpoint, respectively. This is by the following observation. A strong adversary in the AA viewpoint who decides the corruption level $\epsilon_t$ after seeing $a_t$ can be viewed as deciding the corruption $\epsilon_t(a)$ for all action $a$ *before* seeing $a_t$, and set $\epsilon_t = |\epsilon_t(a_t)|$ after seeing $a_t$. In other words, $\epsilon_t(a)$ is the corruption *planned* (before seeing $a_t$) by a strong adversary assuming $a_t = a$, and the adversary simply carries out its plan after seeing $a_t$. It is clear that this is equivalent to the CM viewpoint with $\sum_{t=1}^T |\epsilon_t(a_t)|$ as the corruption measure. See Appendix B for more details. On the other hand, a weak adversary in the AA viewpoint has to decide an upper bound of the corruption level $\epsilon_t$ no matter which action $a_t$ is chosen by the learner. This can be viewed as deciding the corruption $\epsilon_t(a)$ for every action $a$ before seeing $a_t$ with the restriction $|\epsilon_t(a)| \leq \epsilon_t$ for all $a$. Therefore, this is equivalent to using $\sum_{t=1}^T \max_a |\epsilon_t(a)|$ to measure total corruption in the CM viewpoint.

In this work, we adopt the CM viewpoint as described in Section 2. With the CM viewpoint, for both strong and weak settings, the power of the adversary remains the same as the standard "adaptive adversary" (i.e., deciding the corruption function $\epsilon_t(\cdot)$ based on the history up to time $t - 1$), and we only need to derive regret bounds with different corruption measures. All our results can also be interpreted in the AA viewpoint, as the above argument suggests.

With this unified viewpoint, we categorize in Table 1 previous works on linear (contextual) bandits based on the corruption measure, all under the same type of adversary. According to the definitions in Table 1, $C$ and $C_\infty$ correspond to the strong measure and weak measure mentioned above, respectively. It is easy to see that $C \leq \{C_\infty, C_{\mathsf{sq}}\} \leq C_{\mathsf{sq},\infty} \leq C_{\mathsf{ms}}$, where $C_\infty$ and $C_{\mathsf{sq}}$ are incomparable.

Table 1: Classification of previous works based on the corruption measure. Foster et al. (2020), Takemura et al. (2021), and He et al. (2022) studied the more general linear *contextual* bandit setting where the action set can be chosen by an adaptive adversary in every round. Foster et al. (2020) and Takemura et al. (2021) reported their bounds in $C_{\mathsf{sq},\infty}$ and $C_{\mathsf{ms}}$, respectively, though one can make minor modifications to their analysis and show that their algorithms actually ensure the $C_{\mathsf{sq}}$ bound.

| Measure | Definition | Work |
|---|---|---|
| $C_{\mathsf{ms}}$ | $T \max_{t,a} \lvert \epsilon_t(a) \rvert$ | Lattimore et al. (2020), Neu and Olkhovskaya (2020) |
| $C_{\mathsf{sq},\infty}$ | $\left( T \sum_{t=1}^{T} \max_a \epsilon_t(a)^2 \right)^{1/2}$ | Liu et al. (2024) |
| $C_{\mathsf{sq}}$ | $\left( T \sum_{t=1}^{T} \epsilon_t(a_t)^2 \right)^{1/2}$ | Foster et al. (2020), Takemura et al. (2021) |
| $C_{\infty}$ | $\sum_{t=1}^{T} \max_a \lvert \epsilon_t(a) \rvert$ | Li et al. (2019), Bogunovic et al. (2020) |
| $C$ | $\sum_{t=1}^{T} \lvert \epsilon_t(a_t) \rvert$ | Bogunovic et al. (2021, 2022), He et al. (2022) |

Table 2: Regret bounds under corruption measure $C$ and $C_{\infty}$. See Table 1 for their definitions. He et al. (2022) studied the more general linear contextual bandits setting, though it also gives the state-of-the-art $C$ bound for linear bandits.

| Setting | | $C_{\infty}$ bound | $C$ bound |
|---|---|---|---|
| Upper bound | Stochastic LB | $d\sqrt{T} + \sqrt{d}C_{\infty}$ (Algorithm 1) | $d\sqrt{T} + dC$ (He et al., 2022) |
| | Adversarial LB | $d\sqrt{T} + \sqrt{d}C_{\infty}$ (Algorithm 2) | $\sqrt{d^3 T} + dC$ (Algorithm 3) |
| Lower bound | | $d\sqrt{T} + \sqrt{d}C_{\infty}$ (Lattimore et al., 2020) | $d\sqrt{T} + dC$ (Bogunovic et al., 2021) |

For stochastic linear bandits, considering the relations among different corruption measures, the Pareto frontiers of the existing upper bounds are $\widetilde{\mathcal{O}}(d\sqrt{T} + \sqrt{d}C_{\mathsf{sq}})$ by Foster et al. (2020) and Takemura et al. (2021), and $\widetilde{\mathcal{O}}(d\sqrt{T} + dC)$ by He et al. (2022). The lower bound frontiers are $\Omega(d\sqrt{T} + \sqrt{d}C_{\mathsf{ms}})$ by Lattimore et al. (2020) and $\Omega(d\sqrt{T} + dC)$ by Bogunovic et al. (2020). These results imply an $\widetilde{\mathcal{O}}(d\sqrt{T} + dC_{\infty})$ upper bound and an $\Omega(d\sqrt{T} + \sqrt{d}C_{\infty})$ lower bound, which still have a gap. In this work, we close the gap by showing an $\widetilde{\mathcal{O}}(d\sqrt{T} + \sqrt{d}C_{\infty})$ upper bound.

For adversarial linear bandits, we are only aware of upper bound $\widetilde{\mathcal{O}}(d\sqrt{T} + \sqrt{d}C_{\mathsf{sq},\infty})$ by Liu et al. (2024), and not aware of any upper bounds related to $C_{\infty}$ or $C$. In this work, we show $\widetilde{\mathcal{O}}(d\sqrt{T} + \sqrt{d}C_{\infty})$ and $\widetilde{\mathcal{O}}(\sqrt{d^3 T} + dC)$ upper bounds. The results are summarized in Table 2. As in most previous work, we assume that $C_{\infty}$ and $C$ (or their upper bounds) are known by the learner when developing the algorithms. The case of unknown $C_{\infty}$ or $C$ is discussed in Appendix C.

We emphasize that before our work, for both stochastic and adversarial linear bandits, it was unknown how to achieve $\widetilde{\mathcal{O}}(d\sqrt{T} + \sqrt{d}C_{\infty})$ regret. To see how $C_{\infty}$ is different from other notions such as $C_{\mathsf{ms}}$ and $C_{\mathsf{sq}}$, we observe that for stochastic linear bandits, while $\widetilde{\mathcal{O}}(d\sqrt{T} + \sqrt{d}C_{\mathsf{sq}})$ can be achieved via deterministic algorithms, it is not the case for $\widetilde{\mathcal{O}}(d\sqrt{T} + \sqrt{d}C_{\infty})$. The reason is that for deterministic algorithms, the adversary can control $C_{\infty}$ to be the same as $C$, for which $\Omega(d\sqrt{T} + dC)$ is unavoidable. We formalize this in Proposition 1, with the proof given in Appendix D. This precludes the possibility of many previous algorithms to actually achieve the $\widetilde{\mathcal{O}}(d\sqrt{T} + \sqrt{d}C_{\infty})$ upper bound, e.g., Lattimore et al. (2020), Takemura et al. (2021), Bogunovic et al. (2020, 2021), He et al. (2022).

**Proposition 1.** *For stochastic linear bandits, there exists a deterministic algorithm achieving* $\mathrm{Reg}_T = \widetilde{\mathcal{O}}(d\sqrt{T} + \sqrt{d}C_{\mathsf{sq}})$, *while any deterministic algorithm must suffer* $\mathrm{Reg}_T = \Omega(d\sqrt{T} + dC_{\infty})$.

**Algorithm 1:** Randomized Phased Elimination (for stochastic $C_\infty$ and $C$ bounds)

1 **Input**: $Z = \sqrt{d}C_\infty$ or $dC$, action space $\mathcal{A} \subset \mathbb{R}^d$, confidence level $\delta$.

2 Let $\mathcal{A}_1 = \mathcal{A}$ and $L = d\log(|\mathcal{A}|T/\delta)$.

3 **for** $k = 1, 2, \ldots$ **do**

4     Compute a G-optimal design (defined in Section 2) $p_k$ over $\mathcal{A}_k$, and let $G_k = \sum_a p_k(a)aa^\top$.
      Define $\mathcal{I}_k = [(2^{k-1}-1)L + 1, (2^k-1)L]$ and $m_k = |\mathcal{I}_k| = 2^{k-1}L$.

5     **for** $t \in \mathcal{I}_k$ **do** Draw $a_t \sim p_k$ and receive $r_t$ where $\mathbb{E}[r_t] = a_t^\top \theta^\star + \epsilon_t(a_t)$.

6     Define reward vector estimator $\widehat{\theta}_k = (m_k G_k)^{-1}\sum_{t \in \mathcal{I}_k} a_t r_t$ and active action set:

$$\mathcal{A}_{k+1} = \left\{ a \in \mathcal{A}_k : \max_{b \in \mathcal{A}_k} b^\top \widehat{\theta}_k - a^\top \widehat{\theta}_k \leq 8\sqrt{\frac{d\log(|\mathcal{A}|T/\delta)}{m_k}} + \frac{2Z}{m_k} \right\}. \tag{1}$$

## 4 Stochastic Linear Bandits

In this section, we introduce Algorithm 1, which achieves optimal regret for both $C$ and $C_\infty$.

Algorithm 1 is an elimination-based algorithm. At each epoch $k$, it samples actions from a fixed distribution $p_k \in \Delta(\mathcal{A}_k)$, which is a G-optimal design over the active action set $\mathcal{A}_k$ (Line 4). At the end of epoch $k$, only actions that are within the error threshold will be kept in the active action set of the next epoch (Eq. (1)). While previous works by Lattimore et al. (2020) and Bogunovic et al. (2021) have used a similar elimination framework to obtain $\widetilde{\mathcal{O}}(d\sqrt{T} + \sqrt{d}C_{\mathsf{ms}})$ and $\widetilde{\mathcal{O}}(d\sqrt{T} + d^{\frac{3}{2}}C)$ bounds, respectively, we note that their algorithms only specify the number of times the learner should sample for each action in each epoch. This is different from our algorithm that requires the learner to exactly use the distribution $p_k$ to sample actions in every round in epoch $k$. As argued in Proposition 1, if their algorithms are instantiated as a deterministic algorithm, then the regret will be at least $\Omega(d\sqrt{T} + dC_\infty)$. Thus, this subtle difference is important.

Note that to achieve the tight $C_\infty$ (or $C$) bound, $Z = \sqrt{d}C_\infty$ (or $Z = dC$) has to be input to the algorithm to decide the error threshold. The guarantee of Algorithm 1 is stated in Theorem 4.1.

**Theorem 4.1.** *With input $Z = \sqrt{d}C_\infty$ or $Z = dC$, Algorithm 1 ensures with probability at least $1 - \delta$ that* $\mathrm{Reg}_T \leq \mathcal{O}(d\sqrt{T\log(T/\delta)} + Z\log T)$.

Algorithm 1 can also be shown to ensure that $\mathrm{Reg}_T \leq \mathcal{O}(\sqrt{dT\log(|\mathcal{A}|T/\delta)} + Z\log T)$, which could be smaller than the bound given in Theorem 4.1 when $|\mathcal{A}|$ is small.

## 5 Adversarial Linear Bandits

In this section, we consider corrupted adversarial linear bandits. Although adversarial linear bandits have been widely studied, robustness under corruption is an under-explored topic: there is no prior work obtaining regret bounds that linearly depends on either $C_\infty$ or $C$.

### 5.1 $C_\infty$ bound in Adversarial Linear Bandits

Our algorithm (Algorithm 2) is based on follow-the-regularized-leader (FTRL) with logdet regularizer. Similar to previous works (Foster et al., 2020; Zimmert and Lattimore, 2022; Liu et al., 2024, 2023b) that utilize logdet regularizer, the feasible set $\mathcal{H}$ is in $\mathbb{R}^{(d+1)\times(d+1)}$, which is the space of the covariance matrix for distributions over the lifted action space (Line 2–Line 3). At round $t$, the algorithm obtains a covariance matrix $\boldsymbol{H}_t$ by solving the FTRL objective (Eq. (2)). The action distribution $p_t$ is such that the induced covariance matrix is equal to $\boldsymbol{H}_t$ (Eq. (3)). After sampling $a_t \sim p_t$ and obtaining the reward $r_t$, the algorithm constructs reward vector estimator $\widehat{\theta}_t$ (Line 8) and feeds it to FTRL. The reader may refer to Zimmert and Lattimore (2022) for more details.

In typical corruption-free adversarial linear bandits, the learner would construct an unbiased reward vector estimator. However, in the presence of corruption, the learner can no longer construct an unbiased estimator. To compensate the bias, we adopt the idea of "adding exploration bonus" inspired

**Algorithm 2:** FTRL with log-determinant barrier regularizer (for adversarial $C_\infty$ bound)

1 **Parameters**: $\alpha = \max\left\{\frac{C_\infty}{\sqrt{d\log(T)}}, \sqrt{T}\right\}$, $\eta = \sqrt{\frac{\log(T)}{T}}$ and $\gamma = \frac{d}{\sqrt{T}}$.

2 Let $\rho \in \Delta(\mathcal{A})$ be a G-optimal design over $\mathcal{A}$, and let $\Delta_\gamma(\mathcal{A}) = \{p : p = (1-\gamma)p' + \gamma\rho, \ p' \in \Delta(\mathcal{A})\}$.

3 Define feasible set $\mathcal{H} = \left\{\widehat{\mathrm{Cov}}(p) : p \in \Delta_\gamma(\mathcal{A})\right\}$. ($\widehat{\mathrm{Cov}}(p)$ is defined in Section 2)

4 Define $G(\boldsymbol{H}) = -\log\det(\boldsymbol{H})$ and $B_0 = 0$.

5 **for** $t = 1, 2, \ldots$ **do**

6 $\quad$ Solve the fixed-point problem Eq. (2)–Eq. (5).

$$\boldsymbol{H}_t = \operatorname*{argmax}_{\boldsymbol{H} \in \mathcal{H}} \{\eta\langle \boldsymbol{H}, \boldsymbol{\Lambda}_{t-1}\rangle - G(\boldsymbol{H})\} \quad \text{where } \boldsymbol{\Lambda}_{t-1} = \begin{bmatrix} \alpha B_t & \frac{1}{2}\sum_{s=1}^{t-1}\widehat{\theta}_s \\ \frac{1}{2}\sum_{s=1}^{t-1}\widehat{\theta}_s^\top & 0 \end{bmatrix} \quad (2)$$

$$p_t \in \Delta_\gamma(\mathcal{A}) \text{ be such that } \boldsymbol{H}_t = \widehat{\mathrm{Cov}}(p_t), \quad (3)$$

$$\Sigma_t = \sum_{a \in \mathcal{A}} p_t(a) a a^\top, \quad (4)$$

$$B_t = \textsc{Bonus}(B_{t-1}, \Sigma_t). \quad \text{(Defined in Figure 1a)} \quad (5)$$

7 $\quad$ Sample $a_t \sim p_t$. Observe reward $r_t$ with $\mathbb{E}[r_t] = a_t^\top \theta_t + \epsilon_t(a_t)$.

8 $\quad$ Construct reward estimator $\widehat{\theta}_t = \Sigma_t^{-1} a_t r_t$.

---

**Function** $B' = \textsc{Bonus}(B, \Sigma)$:

$\quad$ **if** $B \preceq \Sigma^{-1}$ **then return** $B' = \Sigma^{-1}$.

$\quad$ **else**

$\quad\quad$ Perform eigen-decomposition:

$\quad\quad$ $B^{-\frac{1}{2}}\Sigma^{-1}B^{-\frac{1}{2}} = \sum_{i=1}^d \lambda_i v_i v_i^\top$,

$\quad\quad$ where $\{v_i\}_{i=1}^d$ are unit eigenvectors.

$\quad\quad$ **return** $B' = B^{\frac{1}{2}}\left(\sum_{i=1}^d \max\{\lambda_i, 1\}v_i v_i^\top\right)B^{\frac{1}{2}}$.

(a) The bonus function

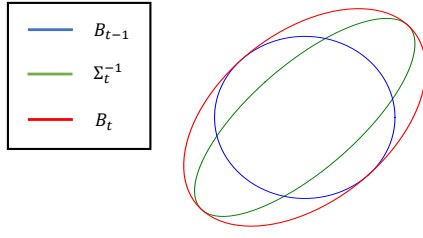

(b) Illustration for $B_t = \textsc{Bonus}(B_{t-1}, \Sigma_t)$. A psd matrix with eigenvalues $(\lambda_i)_{i=1}^d$ is represented as an ellipsoid with radius $(\sqrt{\lambda_i})_{i=1}^d$.

Figure 1: The bonus function and its illustration

by previous work on high-probability adversarial linear bandits (Lee et al., 2020; Zimmert and Lattimore, 2022). In the regret analysis, the exploration bonus creates a negative term that cancels the bias of the loss estimator. The bonus is represented by the $B_t$ in Eq. (5).

To decide the form of $B_t$, we first analyze the bias. With the standard construction of the reward estimator, the bias on the benchmark action $u$ can be calculated as (with $\epsilon_t := \max_a |\epsilon_t(a)|$)

$$\left|u^\top\left(\mathbb{E}_t[\Sigma_t^{-1}a_t r_t] - \theta_t\right)\right| = \left|u^\top \mathbb{E}_t[\Sigma_t^{-1}a_t\epsilon_t(a_t)]\right| \leq \epsilon_t\sqrt{u^\top\Sigma_t^{-1}\mathbb{E}_t[a_t a_t^\top]\Sigma_t^{-1}u} = \epsilon_t\|u\|_{\Sigma_t^{-1}}, \quad (6)$$

where $\Sigma_t$ is the feature covariance matrix induced by $p_t$ (defined in Eq. (4)). Below, we compare different bonus designs in previous and our work.

**Bonus design in previous work.** In Zimmert and Lattimore (2022), which is also based on logdet-FTRL but where the goal is only to get a high-probability bound, the bonus introduces an additional regret the form $-\alpha\|u\|_{\Sigma_t^{-1}}^2 + \alpha\sum_a p_t(a)\|a\|_{\Sigma_t^{-1}}^2$. This can be used to cancel off the bias in Eq. (6):

$$\sum_{t=1}^T \epsilon_t\|u\|_{\Sigma_t^{-1}} - \alpha\sum_{t=1}^T \|u\|_{\Sigma_t^{-1}}^2 + \alpha\sum_{t=1}^T\sum_{a \in \mathcal{A}} p_t(a)\|a\|_{\Sigma_t^{-1}}^2 \leq \sum_{t=1}^T \frac{\epsilon_t^2}{\alpha} + \alpha dT, \quad (7)$$

where we use AM-GM. Unfortunately, with the optimal $\alpha$, this only leads to an additive regret $\sqrt{dT\sum_t \epsilon_t^2} = \sqrt{d}C_{\mathsf{sq},\infty} > \sqrt{d}C_\infty$, which does not meet our goal.

**Bonus design in our work.** To obtain the tighter $\sqrt{d}C_\infty = \sqrt{d}\sum_t \epsilon_t$ bound, our idea is to construct a positive-definite matrix $B_t$ such that $B_t \succeq \Sigma_\tau^{-1}$ for all $\tau \in [t]$, and add bonus $B_t - B_{t-1}$ at round $t$. This way, the total negative regret on $u$ becomes $-\alpha\|u\|_{B_T}^2$ and the cancellation becomes

$$\sum_{t=1}^T \epsilon_t\|u\|_{\Sigma_t^{-1}} - \alpha\|u\|_{B_T}^2 + \alpha\sum_{t=1}^T\sum_{a\in\mathcal{A}} p_t(a)\|a\|_{B_t-B_{t-1}}^2 \le \frac{\left(\sum_{t=1}^T \epsilon_t\right)^2}{\alpha} + \alpha\sum_{t=1}^T \langle \Sigma_t, B_t - B_{t-1}\rangle, \tag{8}$$

where we use $B_T \succeq \Sigma_t^{-1}$ for all $t$ and AM-GM. With this, it suffices to find $B_t$ satisfying our condition $B_t \succeq \Sigma_\tau^{-1}$ for $\tau \le t$, and bound the overhead $\sum_{t=1}^T \langle \Sigma_t, B_t - B_{t-1}\rangle$ by $\widetilde{\mathcal{O}}(d)$.

It turns out that there exists a way to inductively construct $B_t$ so that $B_t \succeq \Sigma_\tau^{-1}$ for all $\tau \le t$ and $\sum_{t=1}^T \langle \Sigma_t, B_t - B_{t-1}\rangle \lesssim \log\det(B_T) = \widetilde{\mathcal{O}}(d)$. This is by letting $B_t$ to be a minimal matrix such that $B_t \succeq B_{t-1}$ and $B_t \succeq \Sigma_t^{-1}$. By induction, this ensures $B_t \succeq \Sigma_\tau^{-1}$ for all $\tau \le t$. The function $B_t = \text{BONUS}(B_{t-1}, \Sigma_t)$ is formally defined in Figure 1a. The geometric interpretation is finding the minimal ellipsoid that contains both ellipsoids induced by $B_{t-1}$ and $\Sigma_t^{-1}$. An illustration figure is given in Figure 1b.

We adopt the fixed-point formulation in Zimmert and Lattimore (2022) (see their FTRL-FB) that includes the bonus for round $t$ (i.e., $B_t$) in the FTRL objective when calculating the policy at round $t$ (Eq. (2)). Notice that $B_t$, in turn, depends on the policy at round $t$ (Eq. (5), where $\Sigma_t$ depends on $p_t$), and thus this forms a fixed-point problem. In the regret analysis, this avoids the "stability term" of the bonus to appear in the regret bound. While the fixed-point solution always exists, it may not be computationally efficient to find. For completeness, in Algorithm 4 (Appendix F), we present a version that does not require solving fixed point but has a suboptimal $d\sqrt{\log T}C_\infty$ additive regret. The guarantee of Algorithm 2 is stated in Theorem 5.1, with its proof deferred to Appendix F.

**Theorem 5.1.** *Algorithm 2 ensures with probability of $1-\delta$, $\text{Reg}_T = \widetilde{\mathcal{O}}(d\sqrt{T} + \sqrt{d}C_\infty)$, where $\widetilde{\mathcal{O}}(\cdot)$ hides $\log(T/\delta)$ factors.*

### 5.2   $C$ bound in Adversarial Linear Bandits

To see how to obtain a $C$ bound, we perform the bias analysis again. Similar but slightly different from Eq. (6), with the standard loss estimator, the bias on action $u$'s reward is bounded by

$$\left|u^\top\left(\mathbb{E}_t[\Sigma_t^{-1}a_t r_t] - \theta_t\right)\right| = \left|u^\top\mathbb{E}_t[\Sigma_t^{-1}a_t\epsilon_t(a_t)]\right| \le \|u\|_{\Sigma_t^{-1}}\mathbb{E}_t\left[\|a_t\|_{\Sigma_t^{-1}}|\epsilon_t(a_t)|\right]. \tag{9}$$

Unlike in Eq. (6), we do not relax $|\epsilon_t(a_t)|$ to $\epsilon_t = \max_a |\epsilon_t(a)|$ because we want the final bound to depend on $C = \sum_t |\epsilon_t(a_t)|$. The idea to ensure that the sum of Eq. (9) over $t$ can be related to $C$ is to make $\|a_t\|_{\Sigma_t^{-1}}$ bounded by a constant $\text{poly}(d)$, which allows us to further bound Eq. (9) by $\text{poly}(d)\|u\|_{\Sigma_t^{-1}}|\epsilon_t(a_t)|$. Such a property holds in standard linear bandit algorithms that operate in the continuous action space where $a_t$ is a point in the convex hull of $\mathcal{A}$, and utilize a more concentrated action sampling scheme. Algorithms that are of this type include SCRiBLe (Abernethy et al., 2008) and continuous exponential weights (CEW) (Ito et al., 2020).

For SCRiBLe and CEW, the work by Lee et al. (2020) and Zimmert and Lattimore (2022) developed techniques that incorporate bonus terms to get high probability regret bounds. The bonus terms introduced by Zimmert and Lattimore (2022) is similar to that discussed in Eq. (7), which only allows us to get a $C_{\text{sq}}$ bound. The bonus terms introduced by Lee et al. (2020) allows us to obtain a $C$ bound, but the overhead introduced by the bonus terms is much larger, resulting in a highly sub-optimal regret bound. Indeed, as shown in Appendix J, adopting their bonus construction results in an additional regret of $d^{\frac{5}{2}}C$. With several attempts, we are only able to obtain the tight corruption dependency $dC$ using the bonus in Section 5.1. To use that bonus, however, it is necessary to lift the problem to $(d+1)^2$-dimensional space. Unfortunately, existing SCRiBLe and CEW algorithms only operate in the original $d$-dimensional space, and as discussed above, we need them to ensure $\|a_t\|_{\Sigma_t^{-1}} \le \text{poly}(d)$.

In order to combine these two useful ideas (i.e., our bonus design in Section 5.1, and the concentrated sampling scheme by SCRiBLe or CEW), we end up with the algorithm that runs CEW over the lifted action space (Algorithm 3). In order to simplify the exposition, we assume without loss of generality

---

**Algorithm 3:** Continuous exponential weights (for adversarial $C$ bound)

---

1 **Parameters**: $\gamma = 1/T$, $\alpha = \sqrt{dT} + C$, $\beta = 4\log(10dT)$, $\eta = \sqrt{d/T}$.
2 **for** $t = 1, 2, \ldots, T$ **do**
3 $\quad$ Solve the fixed-point problem Eq. (10)-Eq. (13).

$$q'_t(h) = \frac{\exp\left(\eta\langle h, \phi(\boldsymbol{\Lambda}_{t-1})\rangle\right)}{\int_{h'\in\phi(\mathcal{H})}\exp\left(\eta\langle h', \phi(\boldsymbol{\Lambda}_{t-1})\rangle\right)\mathrm{d}h'} \quad \text{where} \quad \boldsymbol{\Lambda}_{t-1} = \begin{bmatrix} \alpha B_t & \frac{1}{2}\sum_{s=1}^{t-1}\widehat{\theta}_s \\ \frac{1}{2}\sum_{s=1}^{t-1}\widehat{\theta}_s^{\top} & 0 \end{bmatrix}.$$

4 $\qquad\qquad$ (10)

5 $\quad$ Let $q_t \in \Delta(\mathcal{H})$ and $p_t \in \Delta(\mathcal{A})$ be the distributions of $\boldsymbol{H} \in \mathcal{H}$ and $a \in \mathcal{A}$, respectively, generated by the following ($\boldsymbol{Z}$ is a $d \times (d+1)$ matrix):

$$h \sim q'_t, \quad \boldsymbol{H} = \phi^{-1}(h), \quad a = \begin{bmatrix} 1 & 0 & \cdots & 0 & 0 \\ 0 & 1 & \cdots & 0 & 0 \\ \vdots & \vdots & \ddots & \vdots & \vdots \\ 0 & 0 & \cdots & 1 & 0 \end{bmatrix} \boldsymbol{H}e_{d+1} := \boldsymbol{Z}\boldsymbol{H}e_{d+1}. \quad (11)$$

$$\tilde{p}_t(a) = \frac{p_t(a)\mathbb{1}\{\|a\|_{\Sigma_t^{-1}} \le \sqrt{d}\beta\}}{\int_{a'\in\mathcal{A}} p_t(a')\mathbb{1}\{\|a'\|_{\Sigma_t^{-1}} \le \sqrt{d}\beta\}\mathrm{d}a'}, \quad \text{where } \Sigma_t = \mathbb{E}_{a\sim p_t}[aa^{\top}]. \quad (12)$$

$$B_t = \text{BONUS}(B_{t-1}, \widetilde{\Sigma}_t), \quad \text{where } \widetilde{\Sigma}_t = \gamma I + \mathbb{E}_{a\sim\tilde{p}_t}[aa^{\top}]. \quad (13)$$

6 $\quad$ Sample $a_t \sim \tilde{p}_t$, and observe reward $r_t$ with $\mathbb{E}[r_t] = a_t^{\top}\theta_t + \epsilon_t(a_t)$.
7 $\quad$ Construct reward estimator $\widehat{\theta}_t = \widetilde{\Sigma}_t^{-1}a_t r_t$.

---

that $\mathcal{A} = \text{conv}(\mathcal{A})$. The lifted action space is $\mathcal{H} = \{\widehat{\text{Cov}}(p) : p \in \Delta(\mathcal{A})\} \subset \mathbb{R}^{(d+1)\times(d+1)}$. The price of the lifting is that the "regularization penalty term" in the regret analysis now grows from $\widetilde{\mathcal{O}}(d/\eta)$ to $\widetilde{\mathcal{O}}(d^2/\eta)$, which gives us the $\sqrt{d^3T}$ sub-optimal regret.

Note that CEW requires the assumption that the feasible set is a convex body with non-zero volume, but the effective dimension of $\mathcal{H}$ is strictly smaller than $(d+1)^2$ and thus have zero volume in $\mathbb{R}^{(d+1)^2}$. To correctly write the algorithm, we introduce an invertible linear transformation $\phi : \mathbb{R}^{(d+1)^2} \to \mathbb{R}^m$ that maps an $(d+1)^2$-dimensional action set $\mathcal{H}$ to an $m$-dimensional one, where $m$ is the effective dimension of $\mathcal{H}$. In Appendix I, we formally define this $\phi$. The algorithm uses $\phi$ to map all lifted actions and reward estimators from $\mathbb{R}^{(d+1)\times(d+1)}$ to $\mathbb{R}^m$.

The exponential weights runs over the space of $\phi(\mathcal{H})$ (see Eq. (10)). A point $h \in \phi(\mathcal{H})$ sampled from the exponential weights can be linearly mapped to an action $a \in \mathcal{A}$ according to Eq. (11). We use $q'_t$ to denote the exponential weight distribution in $\phi(\mathcal{H})$, and use $p_t$ to denote the corresponding distribution in $\mathcal{A}$. Instead of sampling $a_t$ from $p_t$, we sample it through rejection sampling that rejects samples with $\|a_t\|_{\Sigma_t^{-1}} > \widetilde{\Theta}(\sqrt{d})$ (Eq. (12)). This technique was developed by Ito et al. (2020), and this guarantees $\|a_t\|_{\Sigma_t^{-1}} \le \widetilde{\mathcal{O}}(\sqrt{d})$—which is our goal as discussed in Eq. (9)—while keeping the clipped distribution $\tilde{p}_t$ close enough to the original distribution $p_t$. This last property heavily relies on the log-concavity of the exponential weight distribution (Ito et al., 2020). The definition of the bonus term is similar to that in Algorithm 2 (Eq. (13)). The construction of the reward estimator (Line 7) and the way of lifting (Eq. (10)) are also similar to those in Algorithm 2. Again, we adopt the fixed-point formulation where the calculation of the policy at time $t$ involves the bonus at time $t$, which, in turn, depends on the policy at time $t$. It is unlikely that this algorithm can be polynomial time. As a remedy, we provide a polynomial time algorithm (Algorithm 6) in Appendix J with a much worse regret bound of $\widetilde{\mathcal{O}}(d^3\sqrt{T} + d^{5/2}C)$. The regret guarantee of Algorithm 3 is given in the following theorem.

**Theorem 5.2.** *Algorithm 3 ensures with probability at least $1 - \delta$, $\text{Reg}_T = \widetilde{\mathcal{O}}(\sqrt{d^3T} + dC)$, where $\widetilde{\mathcal{O}}(\cdot)$ hides polylog$(T/\delta)$ factors.*

# 6   Gap-Dependent Misspecification

Intimately related to corrupted settings are *misspecified* settings, settings where our model class is unable to capture the true environment we are working with. For example, we might consider a stochastic linear bandit problem where the underlying reward function $f(\cdot)$ is nearly linear, i.e., there exists some $\theta$ and $\epsilon^{\mathrm{mis}}(\cdot)$ such that $|f(a) - a^\top \theta| \le \epsilon^{\mathrm{mis}}(a)$ for each $a$. Indeed, in such settings, playing on our true (nearly linear) environment is equivalent to playing on the environment with reward mean $a^\top \theta$ and with corruption $\epsilon^{\mathrm{mis}}(a)$ at each step. Thus, if we can solve corruption settings, it stands to reason that we can solve misspecified settings.

Here we are particularly interested in obtaining bounds on misspecified decision-making that scale precisely with action-dependent misspecification, $\epsilon^{\mathrm{mis}}(a)$. While it is relatively straightforward to obtain bounds on learning in misspecified settings for a uniform level of misspecification $\epsilon \ge \max_{a \in \mathcal{A}} \epsilon^{\mathrm{mis}}(a)$, obtaining bounds on learning with action-dependent misspecification have proved more elusive. To formalize this, we consider, in particular, the following *gap-dependent* notion of misspecification defined in Liu et al. (2023a).

**Assumption 1** (Gap-Dependent Misspecification (Liu et al., 2023a)). *There exists some $\theta \in \mathbb{R}^d$ such that some $\rho > 0$, denoting $\Delta(a) = \max_{a'} f(a') - f(a)$, we have for any $a \in \mathcal{A}$,*
$$|f(a) - a^\top \theta| \le \rho \cdot \Delta(a).$$
*We let $\mathcal{M}^\star$ denote the original environment with reward function $f(a)$ (with $\mathrm{Reg}_T^{\mathcal{M}^\star}$ the corresponding regret), and $\mathcal{M}_0$ the environment with linear reward, $a^\top \theta$, (with $\mathrm{Reg}_T^{\mathcal{M}_0}$ the corresponding regret).*

Assumption 1 allows the reward to be misspecified, but the misspecification level for an action scales with how suboptimal that action is. This could correspond to real-world settings where, for example, significant attention has been given to modeling near-optimal behavior, such that it is accurately represented within our model class, but much less attention has been given to modeling suboptimal behavior. We assume access to a generic corruption-robust algorithm.

**Assumption 2.** *We have access to a regret minimization algorithm which takes as input some $C'$ and with probability at least $1 - \delta$ has regret bounded on $\mathcal{M}_0$ as*
$$\mathrm{Reg}_T^{\mathcal{M}_0} \le \mathcal{C}_1(\delta, T)\sqrt{T} + \mathcal{C}_2(\delta, T)C'$$
*if $C' \ge C \triangleq \sum_{t=1}^T \epsilon^{\mathrm{mis}}(a_t)$, and by $T$ otherwise, for $C$ as defined above and for (problem-dependent) constants $\mathcal{C}_1(\delta, T), \mathcal{C}_2(\delta, T)$ which may scale at most logarithmically with $T$ and $\frac{1}{\delta}$.*

Assumption 2 is essentially the guarantee of a corruption-robust algorithm in terms of strong corruption measure (defined in Section 3). Note, in particular, that Assumption 2 only needs to obtain a sub-linear regret guarantee in the known-corruption setting, and can have linear regret in the setting where the corruption level is unknown. We then have the following result.

**Theorem 6.1.** *Assume our environment satisfies Assumption 1 and that we have access to a corruption-robust algorithm satisfying Assumption 2. Then as long as $\rho \le \min\{\frac{1}{2}, \frac{1}{4}\mathcal{C}_2(\frac{\delta}{T}, T)^{-1}\}$, with probability at least $1 - 2\delta$ we can achieve regret bounded as:*
$$\mathrm{Reg}_T^{\mathcal{M}^\star} \le 6\mathcal{C}_1(\tfrac{\delta}{T}, T)\sqrt{T} + 4\sqrt{2T \log(1/\delta)} + 4.$$

Theorem 6.1 states that, assuming our environment exhibits gap-dependent misspecification with tolerance $\rho \le \min\{\frac{1}{2}, \frac{1}{4}\mathcal{C}_2(\frac{\delta}{T}, T)^{-1}\}$, then we can achieve regret on the true environment bounded as the leading-order term of our corruption-robust oracle, $\mathcal{C}_1(\frac{\delta}{T}, T)\sqrt{T}$, with additional overhead of only $\tilde{O}(\sqrt{T})$. This reduction is almost entirely black-box: it requires knowledge of $\mathcal{C}_1(\delta, T)$ and $\mathcal{C}_2(\delta, T)$, but does not require knowledge of $\rho$ or any other facts about the corruption-robust algorithm.

**Remark 1** (Anytime Algorithm). *The oracle of Assumption 5 must be* anytime*, achieving the above regret guarantee for any $T$ not given as an input. Though many existing corruption-robust algorithms take $T$ as input, the standard doubling trick can convert them into an anytime algorithm.*

## 6.1   Optimal Misspecification Rate for Linear Bandits

We are particularly interested in how stringent a condition on the misspecification level—how small a value of $\rho$—Theorem 6.1 requires. As we have shown, Theorem 4.1 obtains the optimal misspecification level of $dC$. We then have the following corollary.

**Corollary 6.1.1.** *Assume our environment is a misspecified linear bandit satisfying Assumption 1 with $\rho \leq \mathcal{O}(\frac{1}{d \log T})$. Then instantiating Assumption 2 with the algorithm of Theorem 4.1, we can achieve regret bounded with probability $1 - \delta$ as $\mathrm{Reg}_T^{\mathcal{M}^\star} \leq \mathcal{O}(d\sqrt{T \log(T/\delta)})$.*

While the regret bound of Corollary 6.1.1 achieves a scaling of $\widetilde{\mathcal{O}}(d\sqrt{T})$, which is tight for linear bandits (Lattimore and Szepesvári, 2020), it is unclear its requirement on $\rho$ of $\rho \leq \widetilde{\mathcal{O}}(\frac{1}{d})$ is optimal. The result below shows that it is not optimal because $\rho \leq \mathcal{O}(\frac{1}{\sqrt{d}})$ suffices for $\widetilde{\mathcal{O}}(d\sqrt{T})$ regret.

**Theorem 6.2.** *Assume our environment is a misspecified linear bandit satisfying Assumption 1 with $\rho \leq \mathcal{O}(\frac{1}{\sqrt{d}})$. Then there exists an algorithm that achieves, w.p. $1 - \delta$: $\mathrm{Reg}_T^{\mathcal{M}^\star} \leq \mathcal{O}(d\sqrt{T \log(T/\delta)})$.*

Theorem 6.2 relies on a specialized algorithm for the gap-dependent misspecification setting, and improves on the best-known bound for gap-dependent misspecification in linear bandits, which requires $\rho \leq \widetilde{\mathcal{O}}(\frac{1}{d})$ (Liu et al., 2023a). Moreover, for $\rho > c_T \frac{1}{\sqrt{d}}$ for some logarithmic term $c_T$, adapting the lower-bound instance from Lattimore et al. (2020), we show that achieving sub-linear regret is not possible (Theorem K.2). These results jointly show that $\rho \approx \frac{1}{\sqrt{d}}$ is the best $\rho$ we can hope for. This disproves the conjecture of Liu et al. (2023a) that $\rho = \Theta(1)$ is possible.

Note that the reduction in Theorem 6.1 is *not* able to achieve a tight $\rho$—while reducing from gap-dependent misspecification to corruption allows for black-box usage of existing algorithms, it requires more stringent conditions on the misspecification level than specialized algorithms for this setting.

## 6.2 Gap-Dependent Misspecification in Reinforcement Learning

Theorem 6.1 is a corollary of a more general result, Theorem L.1, which applies to misspecified reinforcement learning, where we there assume a generalized notion of gap-dependent misspecification: for each policy $\pi$, $\mathbb{E}^{\mathcal{M}^\star, \pi}[\sum_{h=1}^{H} \epsilon_h^{\mathrm{mis}}(s_h, a_h)] \leq \rho \cdot (V_0^\star - V_0^\pi)$, for $V_0^\pi$ the expected reward of policy $\pi$, and $\epsilon_h^{\mathrm{mis}}(s, a)$ a measure of the misspecification at step $h$, state $s$, and action $a$. To illustrate this general reduction, we consider the following setting, a generalization of linear MDPs (Jin et al., 2020).

**Assumption 3** (Gap-Dependent Misspecified Linear MDPs)**.** *Let $\phi(s, a) : \mathcal{S} \times \mathcal{A} \to \mathbb{R}^d$ denote some feature map and $\boldsymbol{\mu}_h \cdot \mathcal{S} \to \mathbb{R}^d$ some measure which satisfy $\|\phi(s, a)\|_2 \leq 1, \forall s, a$, and $\|\int_s |\mathrm{d}\boldsymbol{\mu}_h(s)|\|_2 \leq \sqrt{d}$. Assume that the transitions $P_h(\cdot \mid s, a)$ on our true environment satisfy:*

$$\|P_h(\cdot \mid s, a) - \langle \phi(s, a), \boldsymbol{\mu}_h(\cdot) \rangle\|_{\mathrm{TV}} \leq \epsilon_h^{\mathrm{mis}}(s, a)$$

*for some $\epsilon_h^{\mathrm{mis}}(s, a) \geq 0$ and $\|P - Q\|_{\mathrm{TV}}$ the total variation distance between $P$ and $Q$. Furthermore, assume that for any policy $\pi$, we have $\mathbb{E}^\pi[\sum_{h=1}^{H} \epsilon_h^{\mathrm{mis}}(s_h, a_h)] \leq \rho \cdot (V_0^\star - V_0^\pi)$.*

We then have the following result.

**Corollary 6.2.1.** *Assume our environment satisfies Assumption 3 with $\rho \leq \widetilde{\mathcal{O}}(\frac{1}{dH})$. Then there exists an algorithm that achieves regret bounded with probability $1 - \delta$ as $\mathrm{Reg}_T^{\mathcal{M}^\star} \leq \widetilde{\mathcal{O}}(\sqrt{d^3 H^2 T})$.*

To the best of our knowledge, Corollary 6.2.1 is the first result showing that it is possible to efficiently learn in linear MDPs with gap-dependent misspecification. Note that under Assumption 3, our MDP could be far from a linear MDP—we simply assume that if we play a "good" policy, it appears as approximately linear. This result is almost immediate by instantiating our reduction with a known corruption-robust algorithm for linear MDPs (Ye et al., 2023).

## 7 Open Problems

It remains open how to achieve $d\sqrt{T} + dC$ regret in corrupted adversarial linear bandits. The tight $C_\infty$ bound for corrupted linear *contextual* bandits, where the action set can be chosen by an adaptive adversary in every round, also remains open. The best known upper and lower bounds for this setting are $\widetilde{\mathcal{O}}(d\sqrt{T} + dC_\infty)$ by He et al. (2022) and $\Omega(d\sqrt{T} + \sqrt{d}C_\infty)$ by Lattimore and Szepesvári (2020).

With the AA viewpoint in Section 3, our work first shows the separation between the achievable regret under weak adversary and strong adversary in corrupted linear bandits. An interesting future direction is to investigate similar separation in general decision making (Foster et al., 2021).

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

# Appendices

# A    Related Work

**Model Misspecification**    Theoretical works on bandits or RL often assume that the underlying world is well-specified by a particular model. Algorithms that are purely built on this assumption are vulnerable to potential misspecifications. Therefore, some works, besides proposing the main results, also discuss the case where the model is misspecified, such as Jiang et al. (2017); Jin et al. (2020); Zanette et al. (2020); Wang et al. (2020); Li and Yang (2024). These discussions, however, usually assume that the amount of misspecification has a uniform upper bound for all actions / states / policies, and the performance degradation is proportional to this uniform upper bound.

For settings like stochastic linear bandits and stochastic linear contextual bandits, it was also found that some widely used algorithm such as LinUCB cannot achieve the tightest guarantee under misspecification (Du et al., 2019). Therefore, a line of work developed better algorithms that have optimal robustness against misspecification, such as Lattimore et al. (2020); Foster et al. (2020); Takemura et al. (2021).

While most work focus on the stochastic setting, Neu and Olkhovskaya (2020) took a first step in studying misspecification in linear contextual bandits with stochastic contexts and adversarial rewards. They established near-optimal regret dependencies on the amount of misspecification.

**Gap-dependent Misspecification**    Gap-dependent misspecification is a setting where the amount of misspecification for an action is bounded by a constant times that action's sub-optimality gap. To our knowledge, this setting is first studied by Liu et al. (2023a) for linear bandits. Another related work is Zhang et al. (2023), which assumes that the misspecification is bounded by a constant times the *minimal sub-optimality gap* among all actions. Although this assumption is more restrictive, they handle the more general linear contextual bandit setting, and derive instance-dependent logarithmic regret bounds.

**Corruption-robust Bandits**    The guarantees on model misspecification is rather pessimistic in the sense that if the misspecification is time-varying, and large misspecification only appears in a few rounds, then the existing guarantees for misspeicifcation still scale with the largest misspeicifcation. To refine such guarantee, previous works have consider different notions of time-varying corruption, and established more fine-grained regret guarantees. These include $C_{\mathsf{sq},\infty}, C_{\mathsf{sq}}, C_\infty$, and $C$ discussed in Section 3. Among them, $C_{\mathsf{sq},\infty}$ and $C_\infty$ are usually studied under the "weak adversary" framework where the adversary decides the corruption before seeing the action chosen by the learner. On the other hand, $C_{\mathsf{sq}}$ and $C$ are usually studied under the "strong adversary" framework where the adversary decides the corruption after seeing the action chosen the learner. In Section 3, we provide a unified view for them so that they can both be regarded as weak adversarial setting but with different corruption measure.

The algorithms of Foster et al. (2020) and Takemura et al. (2021) achieved the optimal bound with respect to $C_{\mathsf{sq}}$ for stochastic linear contextual bandits (i.e., $d\sqrt{T} + \sqrt{d}C_{\mathsf{sq}}$), and He et al. (2022) showed the optimal bound with respect to $C$ (i.e., $d\sqrt{T} + dC$). However, it is still unclear whether the tight dependency on $C_\infty$ is $\sqrt{d}C_\infty$ or $dC_\infty$. In this paper, we answer it for the context-free linear bandit setting, showing that $d\sqrt{T} + \sqrt{d}C_\infty$ is achievable. However, the question remains open for linear contextual bandits.

For the adversarial setting, Liu et al. (2024) showed $d^2\sqrt{T} + \sqrt{d}C_{\mathsf{sq},\infty}$ bound for linear contextual bandits with stochastic contexts and adversarial rewards, which can be improved to $d\sqrt{T} + \sqrt{d}C_{\mathsf{sq},\infty}$ when specialized to adversarial linear bandits. To our knowledge, no $C_\infty$ or $C$ bound has been shown for adversarial linear bandits, and our work make the first attempts on them.

We remark that for $A$-armed adversarial bandits, it is easy to see that $\sqrt{AT} + C_\infty$ bound is achievable simply by running standard adversarial multi-armed bandit algorithm that handles adaptive adversary (e.g., EXP3.P by Auer et al. (2002)). The work of Hajiesmaili et al. (2020) is the only one that we know to obtain $C$ bound for adversarial bandits. They showed a $\sqrt{AT} + AC$ bound for $A$-armed bandits, which is tight.

**Best-of-both-worlds Bounds**    The study of the best-of-both-world problem was initiated by Bubeck and Slivkins (2012) and extended by Seldin and Slivkins (2014); Auer and Chiang (2016); Seldin

and Lugosi (2017); Wei and Luo (2018); Zimmert and Seldin (2019); Zimmert et al. (2019); Ito (2021); Ito and Takemura (2023, 2024); Dann et al. (2023); Kong et al. (2023). The goal of this line of work is to have a single algorithm that achieves a $\widetilde{\mathcal{O}}(\sqrt{T})$ regret when the reward is adversarial and $\mathcal{O}(\log T)$ when the reward is stochastic, without knowing the type of reward in advance. These results should be viewed as refinements of the standard adversarial setting but not the corruption setting considered in our work, though they also used the term "corruption" in their work.

For example, Lee et al. (2021); Ito and Takemura (2024, 2023, 2024); Dann et al. (2023); Kong et al. (2023) studied the best-of-both-world linear bandits problem. The underlying world could be stochastic ($\theta_t = \theta^\star$ for all $t$) or adversarial ($\theta_t$'s are arbitrary). Their algorithm achieves a bound of $\mathcal{O}(d^2 \log(T)/\Delta)$ in the former case, where $\Delta$ is the reward gap between the best and the second-best arm, and $\widetilde{\mathcal{O}}(d\sqrt{T})$ in the latter phase. They also define the *corruption* $C' = \sum_t \max_a |a^\top (\theta_t - \theta^\star)|$ and show that their algorithm achieves a regret of $\mathcal{O}(d^2 \log(T)/\Delta + \sqrt{d^2 \log(T)C'/\Delta})$. Compared to our setting, their corruption is in a more limited form, but their target regret bound in the stochastic setting is tighter than ours.

## B  Equivalence Between AA and CM Viewpoints for Strong Corruption

We show that strong corruption in both definitions is equivalent, that is, for any adversary having strong corruption $C = \sum_t |\epsilon_t|$ from AA viewpoint, there exists an adversary using the equal amount of strong corruption $\sum_t |\epsilon'_t(a_t)|$ from CM viewpoint, where $|\epsilon_t| = |\epsilon'_t(a_t)|$ for all $t$, and vice versa.

Assume that $\epsilon(H_{t-1}, a_t)$ is the function used by an AA strong adversary to decide the corruption at time $t$, where $H_{t-1}$ is the history up to time $t-1$ and $a_t$ is the chosen action at time $t$. Then we define $\epsilon'_t(a) \triangleq \epsilon(H_{t-1}, a), \forall a$ for the CM viewpoint, thus $|\epsilon_t| = |\epsilon(H_{t-1}, a_t)| = |\epsilon'_t(a_t)|$. Note that the function $\epsilon'_t(\cdot)$ only depends on the history up to time $t-1$, so the definition of $\epsilon'_t$ is known to adversary before observing $a_t$. The other direction of this equivalence is achieved by setting the corruption in AA viewpoint as $\epsilon_t = \epsilon'_t(a_t)$. Note that since $a_t$ is known to a strong adversary in AA viewpoint, $\epsilon_t$ is also known.

## C  The Case of Unknown $C_\infty$ or $C$

In the corrupted stochastic setting, Wei et al. (2022) developed a black-box reduction that can turn any algorithm achieving $\beta_1\sqrt{T} + \beta_2 + \beta_3 C_\infty$ regret with the knowledge of $C_\infty$ into an algorithm achieving $\log(T) \times (\beta_1\sqrt{T} + \beta_2 + \beta_3 C_\infty)$ regret without knowledge of $C_\infty$. This reduction can be directly applied to our stochastic $C_\infty$ bound result (Theorem 4.1), which allows us to achieve almost the same regret bound without knowledge of $C_\infty$. The idea of Wei et al. (2022) has been extended to the adversarial setting by Jin et al. (2024) (see their Section 4). Similarly, for the adversarial setting, one can turn any algorithm achieving $\beta_1\sqrt{T} + \beta_2 + \beta_3 C_\infty$ regret with known $C_\infty$ into one achieving $\log(T) \times (\beta_1\sqrt{T} + \beta_2 + \beta_3 C_\infty)$ regret without knowing $C_\infty$. This can be directly applied to our adversarial $C_\infty$ result (Theorem 5.1).

The case of unknown $C$ is quite different. It has been proven by Bogunovic et al. (2021) that it is impossible to achieve a bound that has linear scaling in $C$ (e.g., $\beta_1\sqrt{T} + \beta_2 + \beta_3 C$) for all $C$ simultaneously if $C$ is not known by the learner. This is also mentioned in He et al. (2022) again. Hence, almost all previous work studying $C$ bound assumes knowledge on $C$. If $C$ is unknown, simply setting $\bar{C} = \sqrt{T}$ as an upper bound of $C$ yields a bound of $\mathcal{O}(\sqrt{T} + C^2)$—if $C \leq \sqrt{T}$ indeed holds, then $\bar{C}$ is a correct upper bound, so the regret can be bounded by $\mathcal{O}(\sqrt{T} + \bar{C}) = \mathcal{O}(\sqrt{T})$; if $C > \sqrt{T}$, then simply bound the regret by $T \leq \mathcal{O}(C^2)$.

## D  Proof of Proposition 1

First, we argue that there exists a deterministic algorithm achieving $\widetilde{\mathcal{O}}(d\sqrt{T} + \sqrt{d}C_{\mathsf{sq}})$ upper bound. The algorithm of Takemura et al. (2021) is such an algorithm, although they only showed an upper bound of $\widetilde{\mathcal{O}}(d\sqrt{T} + \sqrt{d}C_{\mathsf{ms}})$. To argue the stronger $\widetilde{\mathcal{O}}(d\sqrt{T} + \sqrt{d}C_{\mathsf{sq}})$ bound, we only need to slightly modify their analysis: In their proof of Lemma 2 (in their Page 6), the original proof bound

the per-step regret due to the misspecification as the following (the calculation below uses their original notation):

$$\left| \sum_{\tau \in \Psi_{t,s}} \epsilon_\tau(i_\tau) x_\tau(i_\tau)^\top V_{t-1,s}^{-1} x_t(i) \right| \le \epsilon \sqrt{|\Psi_{t,s}| \sum_{\tau \in \Psi_{t,s}} \left( x_\tau(i_\tau)^\top V_{t-1,s}^{-1} x_t(i) \right)^2},$$

$$\le \epsilon \sqrt{|\Psi_{t,s}| x_t(i)^\top V_{t-1,s}^{-1} x_t(i)}$$

$$\le \epsilon \sqrt{|\Psi_{t,s}| c^{-2s}}$$

$$\le \widetilde{\mathcal{O}}(\epsilon \sqrt{d}). \qquad \text{(by their Lemma 1)}$$

We can tighten their analysis by doing the following:

$$\left| \sum_{\tau \in \Psi_{t,s}} \epsilon_\tau(i_\tau) x_\tau(i_\tau)^\top V_{t-1,s}^{-1} x_t(i) \right| \le \sqrt{\left( \sum_{\tau \in \Psi_{t,s}} \epsilon_\tau(i_\tau)^2 \right) \left( \sum_{\tau \in \Psi_{t,s}} \left( x_\tau(i_\tau)^\top V_{t-1,s}^{-1} x_t(i) \right)^2 \right)},$$

$$\le \sqrt{\left( \sum_{\tau \in \Psi_{t,s}} \epsilon_\tau(i_\tau)^2 \right) x_t(i)^\top V_{t-1,s}^{-1} x_t(i)}$$

$$\le \sqrt{\left( \sum_{\tau \in \Psi_{t,s}} \epsilon_\tau(i_\tau)^2 \right) c^{-2s}}$$

$$\le \widetilde{\mathcal{O}} \left( \sqrt{\frac{d}{|\Psi_{t,s}|} \sum_{\tau \in \Psi_{t,s}} \epsilon_\tau(i_\tau)^2} \right). \qquad \text{(by their Lemma 1)}$$

Since the regret for every step in $\Psi_{t,s}$ can be bounded by this value, when summing the regret over $\Psi_{T+1,s}$, one can get a regret of order

$$\widetilde{\mathcal{O}} \left( \sqrt{d|\Psi_{T+1,s}| \sum_{\tau \in \Psi_{T+1,s}} \epsilon_\tau(i_\tau)^2} \right).$$

Further summing this over $s$ (there are logarithmically many different $s$) and using that $[T] = \bigcup_s \Psi_{T+1,s}$ and using Cauchy-Schwarz, we get a $\sqrt{dT \sum_{t=1}^T \epsilon_t(i_t)^2} = \sqrt{d} C_{\mathsf{sq}}$ bound.

To argue that any deterministic algorithm must suffer at least $\Omega(d\sqrt{T} + dC_\infty)$ regret, we only need to use the lower bound instance of $\Omega(d\sqrt{T} + dC)$. At the beginning of round $t$, the adversary simply change the corruptions $\epsilon_t(a)$ to be zero for all $a \ne a_t$ (the adversary knows what $a_t$ since the algorithm is deterministic). This makes $C = C_\infty$, and thus the lower bound $\Omega(d\sqrt{T} + dC_\infty)$ holds.

# E   Proof of Theorem 4.1

**Lemma E.1.** *With probability at least $1 - 2\delta$, for all $k$ and for all $b \in \mathcal{A}_k$,*

$$|\langle b, \widehat{\theta}_k - \theta^\star \rangle| \le 4 \sqrt{\frac{d \log(|\mathcal{A}|T/\delta)}{m_k}} + \frac{\min\{\sqrt{d}C', dC\}}{m_k}.$$

*Proof.* Let $\mathbb{E}_t[\cdot]$ be the expectation conditioned on the history up to round $t - 1$. We fix $k$ and $b$ and consider

$$X_t = b^\top (m_k G_k)^{-1} a_t r_t$$

for $t \in \mathcal{I}_k$. Notice that $\sum_{t \in \mathcal{I}_k} X_t = b^\top \widehat{\theta}_k$ and

$$\sum_{t \in \mathcal{I}_k} \mathbb{E}_t [X_t] = \sum_{t \in \mathcal{I}_k} b^\top (m_k G_k)^{-1} \mathbb{E}_t \left[ a_t (a_t^\top \theta^\star + \epsilon_t(a_t)) \right]$$

$$= b^\top \theta^\star + b^\top (m_k G_k)^{-1} \sum_{t \in \mathcal{I}_k} \mathbb{E}_t \left[ a_t \epsilon_t(a_t) \right].$$

Also, by the definition of $G_k$, we have $|X_t| = \left| b^\top (m_k G_k)^{-1} a_t r_t \right| \leq \frac{1}{m_k} \|b\|_{G_k^{-1}} \|a_t\|_{G_k^{-1}} \leq \frac{d}{m_k}$.
Thus, by Freedman's inequality, with probability at least $1 - \frac{\delta}{|\mathcal{A}|T}$, the following holds:

$$\left| \left\langle b, \widehat{\theta}_k - \theta^\star \right\rangle \right| = \left| \sum_{t \in \mathcal{I}_k} X_t - \sum_{t \in \mathcal{I}_k} \mathbb{E}_t [X_t] \right| + \left| b^\top (m_k G_k)^{-1} \sum_{t \in \mathcal{I}_k} \mathbb{E}_t \left[ a_t \epsilon_t(a_t) \right] \right|$$

$$\leq \underbrace{\sqrt{\log \left( \frac{|\mathcal{A}|T}{\delta} \right) \sum_{t \in \mathcal{I}_k} \mathbb{E}_t [X_t^2]} + \frac{d}{m_k} \log \left( \frac{|\mathcal{A}|T}{\delta} \right)}_{\textbf{term}_1} + \underbrace{\left| b^\top (m_k G_k)^{-1} \sum_{t \in \mathcal{I}_k} \mathbb{E}_t \left[ a_t \epsilon_t(a_t) \right] \right|}_{\textbf{term}_2}.$$

We bound $\textbf{term}_1$ by

$$\textbf{term}_1 = \sqrt{\log(|\mathcal{A}|T/\delta) \sum_{t \in \mathcal{I}_k} \mathbb{E}_t \left[ b^\top (m_k G_k)^{-1} a_t a_t^\top (m_k G_k)^{-1} b \right]}$$

$$= \sqrt{\log(|\mathcal{A}|T/\delta) \frac{1}{m_k^2} \sum_{t \in \mathcal{I}_k} b^\top G_k^{-1} b} \leq \sqrt{\frac{d \log(|\mathcal{A}|T/\delta)}{m_k}},$$

and $\textbf{term}_2$ by

$$\textbf{term}_2 = \frac{1}{m_k} \left| \sum_{t \in \mathcal{I}_k} \sum_{a \in \mathcal{A}_k} p_k(a) \epsilon_t(a) b^\top G_k^{-1} a \right|$$

$$\leq \frac{1}{m_k} \sum_{t \in \mathcal{I}_k} \sqrt{\sum_{a \in \mathcal{A}_k} p_k(a) \epsilon_t(a)^2} \sqrt{\sum_{a \in \mathcal{A}_k} p_k(a) \left( b^\top G_k^{-1} a \right)^2}$$

$$\leq \frac{1}{m_k} \sum_{t \in \mathcal{I}_k} \max_{a'} \epsilon_t(a') \sqrt{d} \leq \frac{\sqrt{d} C'}{m_k}.$$

or

$$\textbf{term}_2 \leq \frac{1}{m_k} \sum_{t \in \mathcal{I}_k} \mathbb{E}_t \left[ \epsilon_t(a_t) \left| b^\top G_k^{-1} a_t \right| \right]$$

$$\leq \frac{1}{m_k} \sum_{t \in \mathcal{I}_k} \epsilon_t(a_t) \left| b^\top G_k^{-1} a_t \right| + \frac{1}{m_k} \sqrt{\log \left( \frac{|\mathcal{A}|T}{\delta} \right) \sum_{t \in \mathcal{I}_k} \mathbb{E}_t \left[ \epsilon_t(a_t)^2 \left| b^\top G_k^{-1} a_t \right|^2 \right]}$$

$$+ \frac{\left| \epsilon_t(a_t) b^\top G_k^{-1} a_t \right|}{m_k} \log \left( \frac{|\mathcal{A}|T}{\delta} \right) \qquad \text{(Freedman's inequality)}$$

$$\leq \frac{d}{m_k} \sum_{t \in \mathcal{I}_k} \epsilon_t(a_t) + \frac{1}{m_k} \sqrt{\log \left( \frac{|\mathcal{A}|T}{\delta} \right) \sum_{t \in \mathcal{I}_k} \mathbb{E}_t \left[ b^\top G_k^{-1} a_t a_t^\top G_k^{-1} b \right]} + \frac{d}{m_k} \log \left( \frac{|\mathcal{A}|T}{\delta} \right)$$

$$\leq \frac{dC}{m_k} + \frac{1}{m_k} \sqrt{\log \left( \frac{|\mathcal{A}|T}{\delta} \right) \sum_{t \in \mathcal{I}_k} b^\top G_k^{-1} b} + \frac{d}{m_k} \log \left( \frac{|\mathcal{A}|T}{\delta} \right)$$

$$\leq \frac{dC}{m_k} + \sqrt{\frac{d \log(|\mathcal{A}|T/\delta)}{m_k}} + \frac{d}{m_k} \log \left( \frac{|\mathcal{A}|T}{\delta} \right).$$

Thus, for any $b \in \mathcal{A}_k$, with probability at least $1 - \frac{\delta}{|\mathcal{A}|T}$,

$$\left| \left\langle b, \widehat{\theta}_k - \theta^\star \right\rangle \right| \leq 2\sqrt{\frac{d \log(|\mathcal{A}|T/\delta)}{m_k}} + \frac{2d}{m_k} \log(|\mathcal{A}|T/\delta) + \frac{1}{m_k} \min\left\{ \sqrt{d}C', dC \right\}$$

$$\leq 4\sqrt{\frac{d \log(|\mathcal{A}|T/\delta)}{m_k}} + \frac{1}{m_k} \min\left\{ \sqrt{d}C', dC \right\}. \qquad (m_k \geq d \log(|\mathcal{A}|/\delta))$$

Taking a union bound over $k$ and $b \in \mathcal{A}_k$ finishes the proof. $\qquad \square$

**Lemma E.2.** *Let $a^\star = \operatorname{argmax}_{a \in \mathcal{A}} a^\top \theta^\star$. Then with probability at least $1 - 2\delta$, $a^\star \in \mathcal{A}_k$ for all $k$.*

*Proof.* Suppose that the high-probability event in Lemma E.1 holds. For any $k$, if $a^\star \in \mathcal{A}_k$, then for any $b \in \mathcal{A}_k$,

$$b^\top \widehat{\theta}_k - a^{\star\top} \widehat{\theta}_k \leq b^\top \theta^\star - a^{\star\top} \theta^\star + \left| b^\top (\widehat{\theta}_k - \theta^\star) \right| + \left| a^{\star\top} (\widehat{\theta}_k - \theta^\star) \right|$$

$$\leq 0 + 2 \left( 4\sqrt{\frac{d \log(|\mathcal{A}|T/\delta)}{m_k}} + \frac{\min\{\sqrt{d}C', dC\}}{m_k} \right).$$

By the definition of $\mathcal{A}_{k+1}$ in Eq. (1), we have $a^\star \in \mathcal{A}_{k+1}$. The lemma is then proven by an induction argument. $\qquad \square$

*Proof of Theorem 4.1.* We first calculate the regret in epoch $k > 1$ assuming that the event in Lemma E.2 holds.

$$\sum_{t \in \mathcal{I}_k} \left( \max_{a \in \mathcal{A}} a^\top \theta^\star - a_t^\top \theta^\star \right)$$

$$\leq \sum_{t \in \mathcal{I}_k} \left( \max_{a \in \mathcal{A}} a^\top \widehat{\theta}_{k-1} - a_t^\top \widehat{\theta}_{k-1} \right) + 2 m_k \max_{a \in \mathcal{A}_k} \left| a^\top \left( \widehat{\theta}_{k-1} - \theta^\star \right) \right|$$

$$\leq m_k \cdot \mathcal{O} \left( \sqrt{\frac{d \log(|\mathcal{A}|T/\delta)}{m_{k-1}}} + \frac{\min\{\sqrt{d}C', dC\}}{m_{k-1}} \right)$$

$$= \mathcal{O} \left( \sqrt{d m_k \log(|\mathcal{A}|T/\delta)} + \min\{\sqrt{d}C', dC\} \right).$$

Summing this over $k$ and using that $m_1 = d \log(|\mathcal{A}|T/\delta)$, we get

$$\sum_{t=1}^{T} \left( \max_{a \in \mathcal{A}} a^\top \theta^\star - a_t^\top \theta^\star \right) \leq \mathcal{O} \left( \sqrt{dT \log(|\mathcal{A}|T/\delta)} + d \log(|\mathcal{A}|T/\delta) + \min\{\sqrt{d}C', dC\} \log T \right).$$

Notice that without loss of generality we can assume $d \log(|\mathcal{A}|T/\delta) \leq T$ (otherwise the right-hand side is vacuous). Using this fact gives the desired bound.

$$\square$$

From Exercise 27.6 in Lattimore and Szepesvári (2020), the $\epsilon$-covering number of $\mathcal{A}$ is bounded by $\left( \frac{6d}{\epsilon} \right)^d$. Let $\mathcal{C}(\mathcal{A}, \epsilon)$ be the $\epsilon$-net of $\mathcal{A}$, we then have $\left| \mathcal{C}(\mathcal{A}, \frac{6d}{T}) \right| \leq T^d$. Thus, when $|\mathcal{A}| \geq T^d$, we can use $\mathcal{C}(\mathcal{A}, \frac{6d}{T})$ as $\mathcal{A}_1$ in Algorithm 1 to conduct phase elimination. In that case, following above proof, we have

$$\sum_{t=1}^{T} \left( \max_{a \in \mathcal{C}(\mathcal{A}, \frac{6d}{T})} a^\top \theta^\star - a_t^\top \theta^\star \right) \leq \mathcal{O} \left( \sqrt{dT \log \left( \left| \mathcal{C} \left( \mathcal{A}, \frac{6d}{T} \right) \right| T/\delta \right)} + \min\{\sqrt{d}C', dC\} \log T \right)$$

$$\leq \mathcal{O} \left( d\sqrt{T \log(T/\delta)} + \min\{\sqrt{d}C', dC\} \log T \right).$$

From the definition of covering number, there exists a $a_c^\star \in \mathcal{C}(\mathcal{A}, \frac{6d}{T})$ such that

$$\max_{a \in \mathcal{A}} a^\top \theta^\star - (a_c^\star)^\top \theta^\star \leq \frac{6d}{T}.$$

We have

$$\sum_{t=1}^{T} \left( \max_{a \in \mathcal{A}} a^\top \theta^\star - \max_{a \in \mathcal{C}(\mathcal{A}, \frac{6d}{T})} a^\top \theta^\star \right) \leq \sum_{t=1}^{T} \left( \max_{a \in \mathcal{A}} a^\top \theta^\star - (a_c^\star)^\top \theta^\star \right) + \sum_{t=1}^{T} \left( (a_c^\star)^\top \theta^\star - \max_{a \in \mathcal{C}(\mathcal{A}, \frac{6d}{T})} a^\top \theta^\star \right)$$
$$\leq 6d.$$

Thus,

$$\sum_{t=1}^{T} \left( \max_{a \in \mathcal{A}} a^\top \theta^\star - a_t^\top \theta^\star \right) \leq \mathcal{O}\left( d\sqrt{T \log(T/\delta)} + \min\{\sqrt{d}C', dC\} \log T \right).$$

## F   Proof of Theorem 5.1

In this section, we use the following notation:

$$\widehat{\boldsymbol{\gamma}}_t = \begin{bmatrix} 0 & \frac{1}{2}\widehat{\theta}_t \\ \frac{1}{2}\widehat{\theta}_t^\top & 0 \end{bmatrix}, \qquad \boldsymbol{D}_t = \begin{bmatrix} \alpha B_t - \alpha B_{t-1} & 0 \\ 0 & 0 \end{bmatrix}.$$

Algorithm 2 is equivalent to the FTRL update:

$$\boldsymbol{H}_t = \underset{\boldsymbol{H} \in \mathcal{H}}{\operatorname{argmax}} \left\{ \left\langle \boldsymbol{H}, \sum_{s=1}^{t-1} \widehat{\boldsymbol{\gamma}}_s + \sum_{s=1}^{t} \boldsymbol{D}_s \right\rangle - \frac{G(\boldsymbol{H})}{\eta} \right\}. \tag{14}$$

Algorithm 4 is equivalent to

$$\boldsymbol{H}_t = \underset{\boldsymbol{H} \in \mathcal{H}}{\operatorname{argmax}} \left\{ \sum_{s=1}^{t-1} \langle \boldsymbol{H}, \widehat{\boldsymbol{\gamma}}_s + \boldsymbol{D}_s \rangle - \frac{G(\boldsymbol{H})}{\eta} \right\}. \tag{15}$$

By the standard analysis for FTRL algorithms (e.g., Theorem 2 in Zimmert and Lattimore (2022)), the regret bounds of Eq. (14) and Eq. (15) are given by the following lemmas, respectively.

**Lemma F.1.** *The update rule Eq. (14) (Algorithm 2) ensures for any $\boldsymbol{U} \in \mathcal{H}$,*

$$\sum_{t=1}^{T} \langle \boldsymbol{U} - \boldsymbol{H}_t, \widehat{\boldsymbol{\gamma}}_t \rangle$$
$$\leq \frac{G(\boldsymbol{U}) - \min_{\boldsymbol{H} \in \mathcal{H}} G(\boldsymbol{H})}{\eta} - \sum_{t=1}^{T} \langle \boldsymbol{U} - \boldsymbol{H}_t, \boldsymbol{D}_t \rangle + \sum_{t=1}^{T} \max_{\boldsymbol{H} \in \mathcal{H}} \left\{ \langle \boldsymbol{H} - \boldsymbol{H}_t, \widehat{\boldsymbol{\gamma}}_t \rangle - \frac{D_G(\boldsymbol{H}, \boldsymbol{H}_t)}{\eta} \right\}.$$

**Lemma F.2.** *The update rule Eq. (15) (Algorithm 4) ensures for any $\boldsymbol{U} \in \mathcal{H}$,*

$$\sum_{t=1}^{T} \langle \boldsymbol{U} - \boldsymbol{H}_t, \widehat{\boldsymbol{\gamma}}_t \rangle$$
$$\leq \frac{G(\boldsymbol{U}) - \min_{\boldsymbol{H} \in \mathcal{H}} G(\boldsymbol{H})}{\eta} - \sum_{t=1}^{T} \langle \boldsymbol{U} - \boldsymbol{H}_t, \boldsymbol{D}_t \rangle + \sum_{t=1}^{T} \max_{\boldsymbol{H} \in \mathcal{H}} \left\{ \langle \boldsymbol{H} - \boldsymbol{H}_t, \widehat{\boldsymbol{\gamma}}_t + \boldsymbol{D}_t \rangle - \frac{D_G(\boldsymbol{H}, \boldsymbol{H}_t)}{\eta} \right\}.$$

We consider an arbitrary comparator $p_\star \in \Delta(\mathcal{A})$ with $u_\star = \mathbb{E}_{a \sim p_\star}[a]$. Define $p = (1 - \gamma)p_\star + \gamma\rho$. We have $p \in \Delta_\gamma(\mathcal{A})$, and define $u = \mathbb{E}_{a \sim p}[a]$ and $\boldsymbol{U} = \widehat{\operatorname{Cov}}(p)$. The regret with respect to $p_\star$ can be

decomposed as the following: With probability at least $1 - \delta$,

$$\text{Reg}_T(p_\star) \tag{16}$$

$$= \sum_{t=1}^{T} \langle u_\star - a_t, \theta_t \rangle$$

$$= \sum_{t=1}^{T} \langle u - a_t, \theta_t \rangle + 2\gamma T$$

$$\leq \sum_{t=1}^{T} \langle u - x_t, \theta_t \rangle + \mathcal{O}\left(\sqrt{T \log(1/\delta)}\right) + 2\gamma T \qquad \text{(Azuma's inequality)}$$

$$= \underbrace{\sum_{t=1}^{T} \left\langle u - x_t, \theta_t - \mathbb{E}_t[\widehat{\theta}_t] \right\rangle}_{\textbf{Bias}} + \underbrace{\sum_{t=1}^{T} \left\langle u - x_t, \mathbb{E}_t[\widehat{\theta}_t] - \widehat{\theta}_t \right\rangle}_{\textbf{Deviation}} + \underbrace{\sum_{t=1}^{T} \langle \boldsymbol{U} - \boldsymbol{H}_t, \widehat{\boldsymbol{\gamma}}_t \rangle}_{\textbf{FTRL}} + \mathcal{O}\left(\sqrt{T \log(1/\delta)} + \gamma T\right).$$

$$\tag{17}$$

By Lemma F.1, the **FTRL** term can further be bounded by

$$\textbf{FTRL} \leq \underbrace{\frac{G(\boldsymbol{U}) - \min_{\boldsymbol{H} \in \mathcal{H}} G(\boldsymbol{H})}{\eta}}_{\textbf{Penalty}} - \underbrace{\sum_{t=1}^{T} \langle \boldsymbol{U} - \boldsymbol{H}_t, \boldsymbol{D}_t \rangle}_{\textbf{Bonus}} + \underbrace{\sum_{t=1}^{T} \max_{\boldsymbol{H} \in \mathcal{H}} \langle \boldsymbol{H} - \boldsymbol{H}_t, \widehat{\boldsymbol{\gamma}}_t \rangle - \frac{D_G(\boldsymbol{H}, \boldsymbol{H}_t)}{\eta}}_{\textbf{Stability}}.$$

$$\tag{18}$$

In the following five lemmas, we bound the five terms **Bias**, **Deviation**, **Penalty**, **Bonus**, and **Stability**.

**Lemma F.3.**

$$\textbf{Bias} \leq C_\infty \max_t \|u\|_{\Sigma_t^{-1}} + \sqrt{d} C_\infty.$$

*Proof.*

$$\mathbb{E}_t \left[ \langle u - x_t, -\Sigma_t^{-1} a_t \epsilon_t(a_t) \rangle \right] \leq \mathbb{E}_t \left[ \sqrt{(u - x_t)^\top \Sigma_t^{-1} a_t a_t^\top \epsilon_t^2(a_t) \Sigma_t^{-1}(u - x_t)} \right]$$

$$\leq \sqrt{(u - x_t)^\top \Sigma_t^{-1} \mathbb{E}_t \left[ a_t a_t^\top \epsilon_t^2(a_t) \right] \Sigma_t^{-1}(u - x_t)}$$

$$\leq \epsilon_t \|u - x_t\|_{\Sigma_t^{-1}}$$

$$\leq \epsilon_t \|x_t\|_{\Sigma_t^{-1}} + \epsilon_t \|u\|_{\Sigma_t^{-1}}$$

$$\leq \sqrt{d} \epsilon_t + \epsilon_t \|u\|_{\Sigma_t^{-1}}. \qquad (\Sigma_t \succeq x_t x_t^\top)$$

Thus,

$$\textbf{Bias} = \underbrace{\sum_{t=1}^{T} \left\langle u - x_t, \theta_t - \mathbb{E}_t \left[ \Sigma_t^{-1} a_t a_t^\top \theta_t \right] \right\rangle}_{=0} + \mathbb{E}_t \left[ \sum_{t=1}^{T} \left\langle u - x_t, -\Sigma_t^{-1} a_t \epsilon_t(a_t) \right\rangle \right]$$

$$\leq C_\infty \max_t \|u\|_{\Sigma_t^{-1}} + \sqrt{d} C_\infty.$$

$\square$

**Lemma F.4.** *With probability of at least $1 - \delta$, we have*

$$\textbf{Deviation} \leq \max_t \|u\|_{\Sigma_t^{-1}} \left( 12\sqrt{T \log(T/\delta)} + \frac{12\sqrt{d} \log(T/\delta)}{\sqrt{\gamma}} \right) + 12\sqrt{dT \log(T/\delta)} + \frac{12d \log(T/\delta)}{\sqrt{\gamma}}$$

*Proof.* Notice that

$$
\begin{aligned}
\left|\langle u - x_t, \widehat{\theta}_t\rangle\right| &\leq \left|(u - x_t)^\top \Sigma_t^{-1} a_t\right| \\
&\leq \|u - x_t\|_{\Sigma_t^{-1}} \|a_t\|_{\Sigma_t^{-1}} \\
&\leq \frac{\sqrt{d}}{\sqrt{\gamma}} \|u - x_t\|_{\Sigma_t^{-1}}.
\end{aligned}
$$

By the strengthened Freedman's inequality (Lemma M.3), with probability at least $1 - \delta$,

$$
\begin{aligned}
\textbf{Deviation} &= \sum_{t=1}^{T} \left\langle u - x_t, \mathbb{E}_t[\widehat{\theta}_t] - \widehat{\theta}_t \right\rangle \\
&\leq 3 \sqrt{\sum_{t=1}^{T} \mathbb{E}_t \left[ \left\langle u - x_t, \widehat{\theta}_t \right\rangle^2 \right] \log(d^4 T^4/\delta)} + 2 \cdot \frac{\sqrt{d}}{\sqrt{\gamma}} \max_t \|u - x_t\|_{\Sigma_t^{-1}} \log(d^4 T^4/\delta) \\
&\leq \max_t \|u - x_t\|_{\Sigma_t^{-1}} \left( 12 \sqrt{T \log(T/\delta)} + \frac{12\sqrt{d} \log(T/\delta)}{\sqrt{\gamma}} \right) \\
&\leq \max_t \|u\|_{\Sigma_t^{-1}} \left( 12 \sqrt{T \log(T/\delta)} + \frac{12\sqrt{d} \log(T/\delta)}{\sqrt{\gamma}} \right) + 12\sqrt{dT \log(T/\delta)} + \frac{12 d \log(T/\delta)}{\sqrt{\gamma}}.
\end{aligned}
$$

$\square$

**Lemma F.5.**

$$
\textbf{Penalty} \leq \frac{(d+1)\log(T)}{\eta}.
$$

*Proof.* Define $\boldsymbol{H}_0 = \mathbb{E}_{a\sim\rho} \begin{bmatrix} aa^\top & a \\ a^\top & 1 \end{bmatrix}$. By the definition of the feasible set $\mathcal{H}$, for any $\boldsymbol{H} \in \mathcal{H}$, $\boldsymbol{H} \succeq \gamma \boldsymbol{H}_0 = \frac{d+1}{T} \boldsymbol{H}_0$ and $\boldsymbol{H} \preceq (d+1)\boldsymbol{H}_0$. Thus, **Penalty** can be upper bounded by

$$
\frac{G(\boldsymbol{U}) - \min_{\boldsymbol{H}\in\mathcal{H}} G(\boldsymbol{H})}{\eta} \leq \frac{G\left(\frac{d+1}{T}\boldsymbol{H}_0\right) - G\left((d+1)\boldsymbol{H}_0\right)}{\eta} = \frac{1}{\eta} \log \left( \frac{\det\left((d+1)\boldsymbol{H}_0\right)}{\det\left(\frac{d+1}{T}\boldsymbol{H}_0\right)} \right) = \frac{(d+1)\log(T)}{\eta}.
$$

$\square$

**Lemma F.6.**

$$
\textbf{Bonus} \leq 3\alpha d \log(T) - \alpha \max_t \|u\|_{\Sigma_t^{-1}}^2.
$$

*Proof.* Given $\mathbb{E}_{a\sim p_0}[aa^\top] \succeq \mathbb{E}_{a\sim p_0}[a]\mathbb{E}_{a\sim p_0}[a]^\top = uu^\top$, we have

$$
\sum_{t=1}^{T} \langle \boldsymbol{U}, \boldsymbol{D}_t \rangle = \left\langle \mathbb{E}_{a\sim p_0}[aa^\top], \alpha B_T \right\rangle \geq \left\langle uu^\top, \alpha B_T \right\rangle = \alpha \|u\|_{B_T}^2.
$$

Recall that $B_1 = \Sigma_1^{-1}$ and for $t \geq 2$,

$$
\Sigma_t^{-1} = B_{t-1}^{\frac{1}{2}} \left( \sum_{i=1}^{d} \lambda_{ti} v_{ti} v_{ti}^\top \right) B_{t-1}^{\frac{1}{2}}, \qquad (\{v_{ti}\}_{i=1}^d \text{ are unit eigenvectors})
$$

$$
B_{t-1} = B_{t-1}^{\frac{1}{2}} \left( \sum_{i=1}^{d} v_{ti} v_{ti}^\top \right) B_{t-1}^{\frac{1}{2}}, \qquad (\sum_{i=1}^d v_{ti} v_{ti}^\top = I)
$$

$$
B_t = B_{t-1}^{\frac{1}{2}} \left( \sum_{i=1}^{d} \max\{\lambda_{ti}, 1\} v_{ti} v_{ti}^\top \right) B_{t-1}^{\frac{1}{2}}, \qquad (19)
$$

which ensures $B_t \succeq B_{t-1}$ and $B_t \succeq \Sigma_t^{-1}$. By induction, it leads to $B_T \succeq \Sigma_t^{-1}$ for any $t$. This implies

$$\|u\|_{B_T}^2 \geq \max_t \|u\|_{\Sigma_t^{-1}}^2.$$

Thus, $\sum_{t=1}^T \langle \boldsymbol{U}, \boldsymbol{D}_t \rangle \geq \alpha \max_t \|u\|_{\Sigma_t^{-1}}^2$.

Next, we upper bound $\sum_{t=1}^T \langle \boldsymbol{H}_t, \boldsymbol{D}_t \rangle$. First, notice that $\langle \boldsymbol{H}_1, \boldsymbol{D}_1 \rangle = \alpha \mathrm{Tr}(\Sigma_t B_t) = \mathrm{Tr}(I) = \alpha d$. For $t \geq 2$,

$$
\begin{aligned}
\langle \boldsymbol{H}_t, \boldsymbol{D}_t \rangle &= \alpha \mathrm{Tr}\left( \Sigma_t \left( B_t - B_{t-1} \right) \right) \\
&= \alpha \mathrm{Tr}\left( B_{t-1}^{-\frac{1}{2}} \left( \sum_{i=1}^d \lambda_{ti} v_{ti} v_{ti}^\top \right)^{-1} B_{t-1}^{-\frac{1}{2}} B_{t-1}^{\frac{1}{2}} \left( \sum_{i=1}^d \max\{\lambda_{ti} - 1, 0\} v_{ti} v_{ti}^\top \right) B_{t-1}^{\frac{1}{2}} \right) \\
&\qquad\qquad\qquad\qquad\qquad\qquad\qquad\qquad\qquad\qquad\qquad\qquad\qquad\qquad \text{(by Eq. (19))} \\
&= \alpha \mathrm{Tr}\left( \left( \sum_{i=1}^d \lambda_{ti} v_{ti} v_{ti}^\top \right)^{-1} \left( \sum_{i=1}^d \max\{\lambda_{ti} - 1, 0\} v_{ti} v_{ti}^\top \right) \right) \\
&= \alpha \sum_{i=1}^d \max\left\{ 1 - \frac{1}{\lambda_{ti}}, 0 \right\} \\
&\leq \alpha \sum_{i=1}^d \max\{\log \lambda_{ti}, 0\}.
\end{aligned}
$$

We also have

$$
\begin{aligned}
&\log \det (B_t) - \log \det (B_{t-1}) \\
&= \log \left( \frac{\det\left( B_{t-1}^{\frac{1}{2}} \right) \det\left( \sum_{i=1}^d \max\{\lambda_{ti}, 1\} v_{ti} v_{ti}^\top \right) \det\left( B_{t-1}^{\frac{1}{2}} \right)}{\det\left( B_{t-1}^{\frac{1}{2}} \right) \det\left( \sum_{i=1}^d v_{ti} v_{ti}^\top \right) \det\left( B_{t-1}^{\frac{1}{2}} \right)} \right) \\
&= \log \left( \frac{\det\left( \sum_{i=1}^d \max\{\lambda_{ti}, 1\} v_{ti} v_{ti}^\top \right)}{\det\left( \sum_{i=1}^d v_{ti} v_{ti}^\top \right)} \right) \\
&= \sum_{i=1}^d \max\{\log \lambda_{ti}, 0\}.
\end{aligned}
$$

Thus,

$$\sum_{t=1}^T \langle \boldsymbol{H}_t, \boldsymbol{D}_t \rangle \leq \alpha d + \alpha \log \det (B_T) - \alpha \log \det (B_1) \leq \alpha d + \alpha \log \det (B_T). \qquad (20)$$

Finally, we bound $\log \det (B_T)$. Since $\Sigma_t = \sum_a p_t(a) a a^\top \succeq \gamma \sum_a \rho(a) a a^\top$, by Theorem 3 of Bubeck et al. (2012), we have $\Sigma_t \succeq \frac{\gamma}{d} I$ and $\Sigma_t^{-1} \preceq \frac{d}{\gamma} I$ for all $t$. Thus, $B_1 = \Sigma_1^{-1} \preceq \frac{d}{\gamma} I$. Below, we use induction to show that $B_t \preceq \frac{td}{\gamma} I$. Assume $B_{t-1} \preceq \frac{(t-1)d}{\gamma} I$. Then,

$$
\begin{aligned}
B_t &= B_{t-1}^{\frac{1}{2}} \left( \sum_{i=1}^d \max\{\lambda_{ti}, 1\} v_{ti} v_{ti}^\top \right) B_{t-1}^{\frac{1}{2}} \\
&\preceq B_{t-1}^{\frac{1}{2}} \left( \sum_{i=1}^d (\lambda_{ti} + 1) v_{ti} v_{ti}^\top \right) B_{t-1}^{\frac{1}{2}} \\
&= \Sigma_t^{-1} + B_{t-1} \preceq \frac{d}{\gamma} I + \frac{(t-1)d}{\gamma} I = \frac{td}{\gamma} I.
\end{aligned}
$$

By induction, we get $B_T \preceq \frac{Td}{\gamma}I$ and $\log \det(B_T) \leq 2d\log(T)$ by setting $\gamma = \frac{d}{\sqrt{T}}$. Overall, by Eq. (20), we have

$$\sum_{t=1}^{T} \langle \boldsymbol{H}_t, \boldsymbol{D}_t \rangle \leq 3\alpha d\log(T).$$

Combining the upper bound for $\sum_{t=1}^{T} \langle \boldsymbol{H}_t, \boldsymbol{D}_t \rangle$ and the lower bound for $\sum_{t=1}^{T} \langle \boldsymbol{U}, \boldsymbol{D}_t \rangle$ finishes the proof. $\square$

**Lemma F.7.** *With probability at least* $1 - \delta$

$$\textbf{Stability} \leq \mathcal{O}\left( d\eta T + \frac{\eta d\log(1/\delta)}{\gamma} \right).$$

*Proof.* For any $p$, define $\mu(p) = \mathbb{E}_{a \sim p}[a]$ and

$$\mathrm{Cov}(p) = \mathbb{E}_{a \sim p}[(a - \mu(p))(a - \mu(p))^\top], \quad \widehat{\mathrm{Cov}}(p) = \mathbb{E}_{a \sim p}\begin{bmatrix} \mathrm{Cov}(p) + \mu(p)\mu(p)^\top & \mu(p) \\ \mu(p)^\top & 1 \end{bmatrix}.$$

For any $\boldsymbol{H} = \begin{bmatrix} H + hh^\top & h \\ h^\top & 1 \end{bmatrix}$, given $\boldsymbol{H}_t = \begin{bmatrix} \mathrm{Cov}(p_t) + x_t x_t^\top & x_t \\ x_t^\top & 1 \end{bmatrix}$, we have

$$\langle \boldsymbol{H} - \boldsymbol{H}_t, \widehat{\boldsymbol{\gamma}}_t \rangle - \frac{D_G(\boldsymbol{H}, \boldsymbol{H}_t)}{2\eta} \leq \langle \boldsymbol{H} - \boldsymbol{H}_t, \widehat{\boldsymbol{\gamma}}_t \rangle - \frac{\|x_t - h\|_{\mathrm{Cov}(p_t)^{-1}}^2}{2\eta} \qquad \text{(Lemma M.1)}$$

$$= \left\langle h - x_t, \widehat{\theta}_t \right\rangle - \frac{\|x_t - h\|_{\mathrm{Cov}(p_t)^{-1}}^2}{2\eta}$$

$$\leq \eta\|\widehat{\theta}_t\|_{\mathrm{Cov}(p_t)}^2 \qquad \text{(AM-GM)}$$

$$= \eta r_t^2 a_t^\top \Sigma_t^{-1} \mathrm{Cov}(p_t) \Sigma_t^{-1} a_t$$

$$\leq \eta\|a_t\|_{\Sigma_t^{-1}}^2. \qquad (|r_t| \leq 1 \text{ and } \mathrm{Cov}(p_t) \preceq \Sigma_t)$$

By Freedman's inequality, since $\mathbb{E}_t\left[\|a_t\|_{\Sigma_t^{-1}}^2\right] = d$, and $\eta\|a_t\|_{\Sigma_t^{-1}}^2 \leq \frac{\eta d}{\gamma}$, with probability at least $1 - \delta$, we have

$$\eta \sum_{t=1}^{T} \|a_t\|_{\Sigma_t^{-1}}^2 \leq \mathcal{O}\left( d\eta T + \frac{\eta d\log(1/\delta)}{\gamma} \right).$$

$\square$

*Proof of Theorem 5.1.* Using Lemma F.3–Lemma F.7 in Eq. (17) and Eq. (18), we get
$\mathrm{Reg}_T$

$$\leq \mathcal{O}\left( \frac{d\log(T)}{\eta} + \eta dT + \alpha d\log(T) + \sqrt{dT\log(T/\delta)} + \frac{d\log(T/\delta)}{\sqrt{\gamma}} + \frac{\eta d\log(1/\delta)}{\gamma} + \sqrt{d}C_\infty + \gamma T \right)$$

$$+ \max_t \|u\|_{\Sigma_t^{-1}} \left( 12\sqrt{T\log(T/\delta)} + \frac{12\sqrt{d}\log(T/\delta)}{\sqrt{\gamma}} + C_\infty \right) - \alpha \max_t \|u\|_{\Sigma_t^{-1}}^2$$

$$\leq \mathcal{O}\left( \frac{d\log(T)}{\eta} + \eta dT + \alpha d\log(T) + \sqrt{dT\log(T/\delta)} + \frac{d\log(T/\delta)}{\sqrt{\gamma}} + \frac{\eta d\log(1/\delta)}{\gamma} \right.$$

$$\left. + \frac{(C_\infty)^2}{\alpha} + \frac{T\log(T/\delta)}{\alpha} + \frac{d\log^2(T/\delta)}{\gamma\alpha} + \sqrt{d}C_\infty + \gamma T \right). \qquad \text{(AM-GM)}$$

Therefore, the choice $\gamma = \frac{d}{\sqrt{T}}, \alpha = \max\left\{ \frac{C_\infty}{\sqrt{d\log(T)}}, \sqrt{T} \right\}$ and $\eta = \sqrt{\frac{\log(T)}{T}}$ gives

$$\mathrm{Reg}_T \leq \mathcal{O}\left( d\sqrt{T}\log(T/\delta) + C_\infty\sqrt{d\log(T)} \right).$$

$\square$

**Algorithm 4:** FTRL with log-determinant barrier regularizer

---

**1 Parameters:** $\alpha = \max\left\{\frac{C_\infty}{\sqrt{d\log(T)}}, \sqrt{T}\right\}$, $\eta = \min\left\{\frac{\sqrt{\log(T)}}{16C_\infty}, \sqrt{\frac{\log(T)}{T}}\right\}$, and $\gamma = \frac{d}{\sqrt{T}}$.

**2** Let $\rho \in \Delta(\mathcal{A})$ be John's exploration over $\mathcal{A}$, and let $\Delta_\gamma(\mathcal{A}) = \left\{p : p = (1-\gamma)p' + \gamma\rho, \ p' \in \Delta(\mathcal{A})\right\}$.

**3** Define feasible set $\mathcal{H} = \left\{\widehat{\text{Cov}}(p) : p \in \Delta_\gamma(\mathcal{A})\right\}$.

**4** Define $G(\boldsymbol{H}) = -\log\det(\boldsymbol{H})$ and $B_0 = 0$.

**5 for** $t = 1, 2, \ldots$ **do**

**6** $\quad$ Compute

$$\boldsymbol{H}_t = \underset{\boldsymbol{H} \in \mathcal{H}}{\arg\max}\left\{\eta\langle\boldsymbol{H}, \boldsymbol{\Theta}_{t-1}\rangle - G(\boldsymbol{H})\right\} \quad \text{where} \quad \boldsymbol{\Theta}_{t-1} = \begin{bmatrix} \alpha B_{t-1} & \frac{1}{2}\sum_{s=1}^{t-1}\widehat{\theta}_s \\ \frac{1}{2}\sum_{s=1}^{t-1}\widehat{\theta}_s^\top & 0 \end{bmatrix},$$

$$p_t \in \Delta_\gamma(\mathcal{A}) \text{ be such that } \boldsymbol{H}_t = \widehat{\text{Cov}}(p_t),$$

$$\Sigma_t = \sum_{a \in \mathcal{A}} p_t(a)aa^\top,$$

$$B_t = \text{BONUS}(B_{t-1}, \Sigma_t). \qquad \text{(defined in Figure 1a)}$$

**7** $\quad$ Sample $a_t \sim p_t$. Observe reward $r_t$ with $\mathbb{E}[r_t] = a_t^\top\theta_t + \epsilon_t(a_t)$.

**8** $\quad$ Construct reward estimator $\widehat{\theta}_t = \Sigma_t^{-1}a_t r_t$.

**9 end**

---

## G   Computationally Efficient Algorithm for Adversarial $C_\infty$ Bound

Most proof is the same as Appendix F. Namely, we follow Eq. (17) in Appendix F together with a different decomposition

$$\text{Reg}_T(p_\star) \leq \underbrace{\sum_{t=1}^T\left\langle u - x_t, \theta_t - \mathbb{E}_t[\widehat{\theta}_t]\right\rangle}_{\textbf{Bias}} + \underbrace{\sum_{t=1}^T\left\langle u - x_t, \mathbb{E}_t[\widehat{\theta}_t] - \widehat{\theta}_t\right\rangle}_{\textbf{Deviation}} + \underbrace{\sum_{t=1}^T\langle\boldsymbol{U} - \boldsymbol{H}_t, \widehat{\boldsymbol{\gamma}}_t\rangle}_{\textbf{FTRL}} + \mathcal{O}\left(\sqrt{T\log(1/\delta)} + \gamma T\right).$$

$$\text{(21)}$$

By Lemma F.2, we can further bound **FTRL** by

$$\textbf{FTRL} \leq \underbrace{\frac{G(\boldsymbol{U}) - \min_{\boldsymbol{H} \in \mathcal{H}} G(\boldsymbol{H})}{\eta}}_{\textbf{Penalty}} + \underbrace{\sum_{t=1}^T\langle\boldsymbol{U} - \boldsymbol{H}_t, -\boldsymbol{D}_t\rangle}_{\textbf{Bonus}}$$

$$+ \underbrace{\sum_{t=1}^T\max_{\boldsymbol{H} \in \mathcal{H}}\left\{\langle\boldsymbol{H} - \boldsymbol{H}_t, \widehat{\boldsymbol{\gamma}}_t\rangle - \frac{D_G(\boldsymbol{H}, \boldsymbol{H}_t)}{2\eta}\right\}}_{\textbf{Stability-1}} + \underbrace{\sum_{t=1}^T\max_{\boldsymbol{H} \in \mathcal{H}}\left\{\langle\boldsymbol{H}_t - \boldsymbol{H}, -\boldsymbol{D}_t\rangle - \frac{D_G(\boldsymbol{H}, \boldsymbol{H}_t)}{2\eta}\right\}}_{\textbf{Stability-2}}.$$

$$\text{(22)}$$

Among the terms above, **Bias**, **Penalty**, **Deviation**, **Bonus**, and **Stability-1** follow the same bounds as Lemma F.3, Lemma F.5, Lemma F.6, and Lemma F.7, respectively. It remains to bound **Stability-2**.

**Lemma G.1.** *If* $\eta \leq \frac{1}{16\sqrt{d}\alpha}$, *then*

$$\textbf{Stability-2} \leq 8\eta\alpha^2 d.$$

*Proof.* From the analysis of bias term and $\boldsymbol{H}_t$ and $\boldsymbol{D}_t$ are both positive semi-definite, we have

$$\sqrt{\text{Tr}\left(\boldsymbol{H}_t\boldsymbol{D}_t\boldsymbol{H}_t\boldsymbol{D}_t\right)} = \alpha\sqrt{\text{Tr}\left(\Sigma_t\left(B_t - B_{t-1}\right)\Sigma_t\left(B_t - B_{t-1}\right)\right)} \leq \alpha\sqrt{d}$$

where the last inequality is due to $\Sigma_t^{-1} \succeq B_t - B_{t-1}$. Since $\eta \leq \frac{1}{16\sqrt{d}\alpha}$, by Lemma M.2, with probability of at least $1 - \delta$, we have

$$
\begin{aligned}
\textbf{Stability-2} &\leq 8\eta \sum_{t=1}^{T} \text{Tr}\left(\boldsymbol{H}_t \boldsymbol{D}_t \boldsymbol{H}_t \boldsymbol{D}_t\right) \\
&\leq 8\eta\alpha^2 \sum_{t=1}^{T} \text{Tr}\left(\Sigma_t \left(B_t - B_{t-1}\right) \Sigma_t \left(B_t - B_{t-1}\right)\right) \\
&\leq 8\eta\alpha^2 d
\end{aligned}
$$

where the last step follows the similar analysis in Lemma F.6. □

*Proof of Theorem 5.1 (Option II).* Using Lemma F.3–Lemma G.1 in Eq. (21) and Eq. (22), we get

$$
\begin{aligned}
\text{Reg}_T = \mathcal{O}\Bigg(&\frac{d\log(T)}{\eta} + \eta dT + d\alpha\log(T) + \sqrt{dT\log(T/\delta)} + \frac{d\log(T/\delta)}{\sqrt{\gamma}} + \frac{\eta d\log(1/\delta)}{\gamma} + \eta\alpha^2 d \\
&+ \frac{(C_\infty)^2}{\alpha} + \frac{T\log(T/\delta)}{\alpha} + \frac{d\log^2(T/\delta)}{\gamma\alpha} + \sqrt{d}C_\infty + \gamma T\Bigg)
\end{aligned}
$$

By choosing $\alpha = \max\left\{\frac{C_\infty}{\sqrt{d\log(T)}}, \sqrt{T}\right\}$ and $\eta = \min\left\{\frac{\sqrt{\log(T)}}{16C_\infty}, \sqrt{\frac{\log(T)}{T}}\right\}$, and $\gamma = \frac{d}{\sqrt{T}}$, we could ensure $\eta \leq \frac{1}{16\sqrt{d}\alpha}$. This gives the final regret $\mathcal{O}\left(d\sqrt{T}\log(T/\delta) + dC_\infty\sqrt{\log T}\right)$. The additional $\sqrt{d}$ factor comes from the additional condition for the **Stability-2** term. □

## H  Proof of Theorem 5.2

Similar to before, we define

$$
\widehat{\boldsymbol{\gamma}}_t = \begin{bmatrix} 0 & \frac{1}{2}\widehat{\theta}_t \\ \frac{1}{2}\widehat{\theta}_t^\top & 0 \end{bmatrix}, \qquad \boldsymbol{D}_t = \begin{bmatrix} \alpha B_t - \alpha B_{t-1} & 0 \\ 0 & 0 \end{bmatrix},
$$

and $x_t = \mathbb{E}_{a\sim p_t}[a]$, $\tilde{x}_t = \mathbb{E}_{a\sim\tilde{p}_t}[a]$. We perform the regret decomposition as the following.

$$
\begin{aligned}
\text{Reg}_T &= \sum_{t=1}^{T} \langle u - a_t, \theta_t \rangle \\
&= \sum_{t=1}^{T} \langle u - x_t, \theta_t \rangle + \sum_{t=1}^{T} \langle x_t - \tilde{x}_t, \theta_t \rangle + \sum_{t=1}^{T} \langle \tilde{x}_t - a_t, \theta_t \rangle \\
&= \sum_{t=1}^{T} \langle u - x_t, \theta_t \rangle + \mathcal{O}\left(\gamma T + \sqrt{T\log(1/\delta)}\right) \\
&= \underbrace{\sum_{t=1}^{T} \left\langle u - x_t, \theta_t - \mathbb{E}_t[\widehat{\theta}_t] \right\rangle}_{\textbf{Bias}} + \underbrace{\sum_{t=1}^{T} \left\langle u - x_t, \mathbb{E}_t[\widehat{\theta}_t] - \widehat{\theta}_t \right\rangle}_{\textbf{Deviation}} + \underbrace{\sum_{t=1}^{T} \left\langle u - x_t, \widehat{\theta}_t \right\rangle}_{\textbf{FTRL}} + \mathcal{O}\left(\gamma T + \sqrt{T\log(1/\delta)}\right).
\end{aligned}
$$

(23)

The **FTRL** term can be further bounded as the following.

**FTRL**

$$= \sum_{t=1}^{T} \mathbb{E}_{a \sim p_t} \left[ \left\langle u - a, \widehat{\theta}_t \right\rangle \right]$$

$$= \sum_{t=1}^{T} \mathbb{E}_{\boldsymbol{H} \sim q_t} \left[ \langle \boldsymbol{U} - \boldsymbol{H}, \widehat{\boldsymbol{\gamma}}_t \rangle \right]$$

$$\leq \frac{d^2 \log T}{\eta} + \frac{1}{\eta} \sum_{t=1}^{T} \mathbb{E}_{\boldsymbol{H} \sim q_t} \left[ \exp \left( \eta \langle \boldsymbol{H}, \widehat{\boldsymbol{\gamma}}_t \rangle \right) - \eta \langle \boldsymbol{H}, \widehat{\boldsymbol{\gamma}}_t \rangle - 1 \right] + \sum_{t=1}^{T} \mathbb{E}_{\boldsymbol{H} \sim q_t} \left[ \langle \boldsymbol{H} - \boldsymbol{U}, \boldsymbol{D}_t \rangle \right]$$

(by Theorem I.2)

$$= \frac{d^2 \log T}{\eta} + \underbrace{\frac{1}{\eta} \sum_{t=1}^{T} \mathbb{E}_{a \sim p_t} \left[ \exp \left( \eta \left\langle a, \widehat{\theta}_t \right\rangle \right) - \eta \left\langle a, \widehat{\theta}_t \right\rangle - 1 \right]}_{\textbf{Stability}} + \underbrace{\alpha \sum_{t=1}^{T} \mathbb{E}_{a \sim p_t} \left[ \|a\|^2_{B_t - B_{t-1}} \right] - \alpha \sum_{t=1}^{T} \|u\|^2_{B_T}}_{\textbf{Bonus}}.$$

(24)

In the following four lemmas, we bound the four terms **Bias**, **Deviation**, **Bonus**, **Stability**.

**Lemma H.1.**

$$\textbf{Bias} \leq \left( \max_t \|x_t - u\|_{\widetilde{\Sigma}_t^{-1}} \right) \left( \sqrt{d\gamma} T + 2\sqrt{T \log(1/\delta)} + \sqrt{d}\beta C \right).$$

*Proof.*

$$\textbf{Bias} = \sum_{t=1}^{T} \left\langle u - x_t, \theta_t - \mathbb{E}_t[\widehat{\theta}_t] \right\rangle$$

$$= \sum_{t=1}^{T} \left\langle u - x_t, \theta_t - \widetilde{\Sigma}_t^{-1} \mathbb{E}_t \left[ a_t a_t^\top \right] \theta_t + \widetilde{\Sigma}_t^{-1} \mathbb{E}_t \left[ a_t \epsilon_t(a_t) \right] \right\rangle$$

$$= \gamma \sum_{t=1}^{T} \left\langle u - x_t, \widetilde{\Sigma}_t^{-1} \theta_t \right\rangle + \sum_{t=1}^{T} \left\langle u - x_t, -\widetilde{\Sigma}_t^{-1} \mathbb{E}_t \left[ a_t \epsilon_t(a_t) \right] \right\rangle \qquad (\widetilde{\Sigma}_t = \gamma I + \mathbb{E}_t[a_t a_t^\top])$$

$$\leq \gamma \sum_{t=1}^{T} \|x_t - u\|_{\widetilde{\Sigma}_t^{-1}} \|\theta_t\|_{\widetilde{\Sigma}_t^{-1}} + \sum_{t=1}^{T} \|x_t - u\|_{\widetilde{\Sigma}_t^{-1}} \mathbb{E}_t \left[ \|a_t\|_{\widetilde{\Sigma}_t^{-1}} |\epsilon_t(a_t)| \right]$$

$$\leq \left( \max_t \|x_t - u\|_{\widetilde{\Sigma}_t^{-1}} \right) \left( \gamma \sqrt{\frac{d}{\gamma}} T + \sum_{t=1}^{T} \mathbb{E}_t \left[ \|a_t\|_{\widetilde{\Sigma}_t^{-1}} |\epsilon_t(a_t)| \right] \right)$$

$(\widetilde{\Sigma}_t \succeq \gamma I$ and $\|\theta_t\|_2 \leq \sqrt{d})$

$$\leq \left( \max_t \|x_t - u\|_{\widetilde{\Sigma}_t^{-1}} \right) \left( \sqrt{d\gamma} T + \sqrt{d}\beta \sum_{t=1}^{T} \mathbb{E}_t \left[ |\epsilon_t(a_t)| \right] \right)$$

$$\leq \left( \max_t \|x_t - u\|_{\widetilde{\Sigma}_t^{-1}} \right) \left( \sqrt{d\gamma} T + 2\sqrt{T \log(1/\delta)} + \sqrt{d}\beta \sum_{t=1}^{T} |\epsilon_t(a_t)| \right).$$

(Azuma's inequality)

□

**Lemma H.2.**

$$\textbf{Deviation} \leq \mathcal{O} \left( \max_t \|u - x_t\|_{\widetilde{\Sigma}_t^{-1}} d\beta \sqrt{T} \log(T/\delta) \right).$$

*Proof.* Notice that

$$
\begin{aligned}
\left| \langle u - x_t, \widehat{\theta}_t \rangle \right| &\leq \left| (u - x_t)^\top \widetilde{\Sigma}_t^{-1} a_t \right| \\
&\leq \| u - x_t \|_{\widetilde{\Sigma}_t^{-1}} \| a_t \|_{\widetilde{\Sigma}_t^{-1}} \\
&\leq \sqrt{d}\beta \| u - x_t \|_{\widetilde{\Sigma}_t^{-1}}.
\end{aligned}
$$

By the strengthened Freedman's inequality (Lemma M.3), with probability at least $1 - \delta$,

$$
\begin{aligned}
\textbf{Deviation} &= \sum_{t=1}^{T} \left\langle u - x_t, \mathbb{E}_t[\widehat{\theta}_t] - \widehat{\theta}_t \right\rangle \\
&\leq \mathcal{O}\left( 3 \sqrt{\sum_{t=1}^{T} \mathbb{E}_t \left[ \left\langle u - x_t, \widehat{\theta}_t \right\rangle^2 \right] \log(T^d/\delta)} + 2\sqrt{d}\beta \max_t \| u - x_t \|_{\widetilde{\Sigma}_t^{-1}} \log(T^d/\delta) \right) \\
&\leq \mathcal{O}\left( \max_t \| u - x_t \|_{\widetilde{\Sigma}_t^{-1}} d\beta \sqrt{T} \log(T/\delta) \right). \qquad \text{(using the assumption } d \leq T)
\end{aligned}
$$

$\square$

**Lemma H.3.**

$$
\textbf{Bonus} \leq 3\alpha d \log(T) - \alpha \max_t \| u \|_{\widetilde{\Sigma}_t^{-1}}^2.
$$

*Proof.* The proof the same as in the logdet case. See the proof of Lemma F.6.

$\square$

**Lemma H.4.**

$$
\textbf{Stability} \leq \mathcal{O}(\eta dT \log^2 T).
$$

*Proof.*

$$
\textbf{Stability} = \frac{1}{\eta} \sum_{t=1}^{T} \mathbb{E}_{a \sim p_t} \left[ \exp\left( \eta \left\langle a, \widehat{\theta}_t \right\rangle \right) - \eta \left\langle a, \widehat{\theta}_t \right\rangle - 1 \right]
$$

Since $q_t'$ is a log-concave distribution, so are $q_t$ and $p_t$, which further implies that $\eta \left\langle a, \widehat{\theta}_t \right\rangle$ follows a log-concave distribution. Furthermore,

$$
\mathbb{E}_{a \sim p_t} \left[ \eta^2 \left\langle a, \widehat{\theta}_t \right\rangle^2 \right] \leq \mathbb{E}_{a \sim p_t} \left[ \eta^2 a_t^\top \widetilde{\Sigma}_t^{-1} a a^\top \widetilde{\Sigma}_t^{-1} a_t \right] \leq 2\eta^2 \| a_t \|_{\widetilde{\Sigma}_t^{-1}}^2 \leq 2\eta^2 d\beta^2 \leq \frac{1}{100},
$$

where we use Lemma J.2 in the second-last inequality . By Lemma 6 of Ito et al. (2020), we have

$$
\frac{1}{\eta} \sum_{t=1}^{T} \mathbb{E}_{a \sim p_t} \left[ \exp\left( \eta \left\langle a, \widehat{\theta}_t \right\rangle \right) - \eta \left\langle a, \widehat{\theta}_t \right\rangle - 1 \right] \leq \eta \sum_{t=1}^{T} \mathbb{E}_{a \sim p_t} \left[ \left\langle a, \widehat{\theta}_t \right\rangle^2 \right] \leq 2\eta \sum_{t=1}^{T} \| a_t \|_{\widetilde{\Sigma}_t^{-1}}^2 \leq 2\eta\beta^2 dT.
$$

$\square$

*Proof of Theorem 5.2.* Combining Eq. (23), Eq. (24), and Lemma H.1, Lemma H.2, Lemma H.3, Lemma H.4, we see that the regret is bounded by

$$
\begin{aligned}
&\widetilde{\mathcal{O}}\left( \frac{d^2}{\eta} + \eta dT + \max_t \| u - x_t \|_{\widetilde{\Sigma}_t^{-1}} (d\sqrt{T} + \sqrt{d}C) + dC + \alpha d \right) - \alpha \| u \|_{B_T}^2 \\
&\leq \widetilde{\mathcal{O}}\left( \frac{d^2}{\eta} + \eta dT + \alpha d \right) + \frac{d^2 T + dC^2}{\alpha} \qquad \text{(AM-GM inequality)}
\end{aligned}
$$

Choosing optimal $\alpha$ and $\eta$ leads to $\widetilde{\mathcal{O}}(\sqrt{d^3 T} + dC)$.

$\square$

# I  Dimension Reduction for Continuous Exponential Weights

First, the intrinsic dimension of $\mathcal{X}$ can be defined as the following:

**Definition 1.** *The intrinsic dimension of $\mathcal{X}$ is defined as*

$$\dim(\mathcal{X}) = \dim\left(\operatorname{span}\left(\mathcal{X} - \mathcal{X}\right)\right),$$

*where $\mathcal{X} - \mathcal{X} \triangleq \{x - x' : x, x' \in \mathcal{X}\}$.*

A convex region $\mathcal{X} \subset \mathbb{R}^n$ can be translated and rotated so that it entirely lies in $\mathbb{R}^m$ where $m = \dim(\mathcal{X})$ and has non-zero volume in $\mathbb{R}^m$. We more precisely define this transformation below.

**Definition 2.** *Let $\mathcal{X} \subset \mathbb{R}^n$ be a convex region with $\dim(\mathcal{X}) = m$. We define $\phi : \mathbb{R}^n \to \mathbb{R}^m$ as the following linear transformation:*

$$\phi(x) \triangleq ZMx,$$

*where $M \in \mathbb{R}^{n \times n}$ is a rotation matrix (i.e., orthogonal matrix) such that for any $v \in \mathcal{X} - \mathcal{X}$, $Mv$ has non-zero elements only in the first $m$ coordinates (this is always possible by the definition of $\dim(\mathcal{X})$ in Definition 1), and*

$$Z = \begin{bmatrix} 1 & 0 & \cdots & 0 & 0 & \cdots & 0 \\ 0 & 1 & \cdots & 0 & 0 & \cdots & 0 \\ \vdots & \vdots & \ddots & \vdots & \vdots & \cdots & \vdots \\ 0 & 0 & \cdots & 1 & 0 & \cdots & 0 \end{bmatrix} \in \mathbb{R}^{m \times n}$$

*extracts the first $m$ coordinates of a given $n$-dimensional vector.*

**Lemma I.1.** *For any $x \in \mathcal{X}$ and any $\theta \in \mathbb{R}^n$,*

$$\langle x, \theta \rangle = \langle \phi(x), \phi(\theta) \rangle + f(\phi, \theta),$$

*where $f(\phi, \theta) \in \mathbb{R}$ is some quantity that only depends on $\phi$ and $\theta$ but not $x$.*

*Proof.* Let $x, x' \in \mathcal{X}$. By the definition of $\phi$, we have

$$\langle \phi(x) - \phi(x'), \phi(\theta) \rangle = \langle ZM(x - x'), ZM\theta \rangle.$$

By the choice of $M$ in Definition 2, $M(x - x')$ only has non-zero elements in the first $m$ coordinates. Furthermore, since $Z$ extracts the first $m$ coordinates, we have

$$\begin{aligned} \langle ZM(x - x'), ZM\theta \rangle &= \sum_{i=1}^{m} (M(x - x'))_i (M\theta)_i \\ &= \sum_{i=1}^{n} (M(x - x'))_i (M\theta)_i \\ &= \langle M(x - x'), M\theta \rangle \\ &= \langle x - x', \theta \rangle. \qquad (M^\top = M^{-1} \text{ because } M \text{ is a rotation matrix}) \end{aligned}$$

Thus,

$$\langle x, \theta \rangle - \langle \phi(x), \phi(\theta) \rangle = \langle x', \theta \rangle - \langle \phi(x'), \phi(\theta) \rangle,$$

meaning that the value of $\langle x, \theta \rangle - \langle \phi(x), \phi(\theta) \rangle$ is shared by all $x \in \mathcal{X}$. Defining this value as $f(\phi, \theta)$ finishes the proof. $\qquad \square$

We consider the continuous exponential weight algorithm (Algorithm 5) running on $\phi(\mathcal{X}) \subset \mathbb{R}^m$:

---

**Algorithm 5:** Exponential Weights

---

**1** Let $\mathcal{X} \subset \mathbb{R}^n$, and let $\phi(\mathcal{X}) \triangleq \{\phi(x) : x \in \mathcal{X}\}$.

**2** **for** $t = 1, 2, \ldots$ **do**

**3**     Define for $y \in \phi(\mathcal{X})$,

$$w_t(y) = \exp\left(\eta \sum_{s=1}^{t-1} \langle y, \phi(\theta_s)\rangle + \eta \sum_{s=1}^{t} \langle y, \phi(b_s)\rangle\right) \ \text{ and } \ p_t(y) = \frac{w_t(y)}{\int_{y' \in \phi(\mathcal{X})} w_t(y')\mathrm{d}y'}$$

    for some bonus term $b_t$.

**4**     Sample $y_t \sim p_t$, and play $x_t = \phi^{-1}(y_t)$, where $\phi^{-1}$ is the inverse mapping of $\phi$.

**5**     Receive $\theta_t \in \mathbb{R}^n$.

---

In Algorithm 5, we require that the inverse mapping of $\phi$ exists. This is true because for any $x, x' \in \mathcal{X}$, we have $\|\phi(x) - \phi(x')\| = \|ZM(x - x')\| = \|M(x - x')\| = \|x - x'\|$, and thus $\phi$ cannot map $x, x' \in \mathcal{X}$ with $x \neq x'$ to the same point.

**Theorem I.2.** *Let $q_t \in \Delta(\mathcal{X})$ be the distribution such that $x \sim q_t$ is equivalent to first drawing $y \sim p_t$ and then taking $x = \phi^{-1}(y)$. Algorithm 5 ensures for any $x \in \mathcal{X}$,*

$$\sum_{t=1}^{T} \langle x, \theta_t + b_t \rangle - \sum_{t=1}^{T} \mathbb{E}_{x \sim q_t}\left[\langle x, \theta_t + b_t\rangle\right] \leq \frac{m \log T}{\eta} + \frac{1}{\eta} \sum_{t=1}^{T} \mathbb{E}_{x \sim q_t}\left[\exp\left(\eta \langle x, \theta_t\rangle\right) - \eta \langle x, \theta_t\rangle - 1\right].$$

*Proof.* Note that Algorithm 5 is a standard continuous exponential weight algorithm over reward vectors $\phi(\theta_t)$ and in the space of $\phi(\mathcal{X}) \subset \mathbb{R}^m$. By the standard analysis (see, e.g., Ito et al. (2020); Zimmert and Lattimore (2022)), we have for any sequence $\lambda_1, \ldots, \lambda_T \in \mathbb{R}$ and any $y \in \phi(\mathcal{X})$,

$$\sum_{t=1}^{T} \langle y, \phi(\theta_t) + \phi(b_t)\rangle - \sum_{t=1}^{T} \mathbb{E}_{y \sim p_t}\left[\langle y, \phi(\theta_t) + \phi(b_t)\rangle\right]$$

$$\leq \frac{m \log T}{\eta} + \frac{1}{\eta} \sum_{t=1}^{T} \mathbb{E}_{y \sim p_t}\left[\exp\left(\eta \langle y, \phi(\theta_t)\rangle + \lambda_t\right) - \left(\eta \langle y, \phi(\theta_t)\rangle + \lambda_t\right) - 1\right].$$

By Lemma I.1, the above implies

$$\sum_{t=1}^{T} \langle \phi^{-1}(y), \theta_t + b_t \rangle - \sum_{t=1}^{T} \mathbb{E}_{y \sim p_t}\left[\langle \phi^{-1}(y), \theta_t + b_t\rangle\right]$$

$$\leq \frac{m \log T}{\eta} + \frac{1}{\eta} \sum_{t=1}^{T} \mathbb{E}_{y \sim p_t}\left[\exp\left(\eta \langle \phi^{-1}(y), \theta_t\rangle - \eta f(\phi, \theta_t) + \lambda_t\right) - \left(\eta \langle \phi^{-1}(y), \theta_t\rangle - \eta f(\phi, \theta_t) + \lambda_t\right) - 1\right],$$

which further implies that for any $x \in \mathcal{X}$,

$$\sum_{t=1}^{T} \langle x, \theta_t + b_t \rangle - \sum_{t=1}^{T} \mathbb{E}_{x \sim q_t}\left[\langle x, \theta_t + b_t\rangle\right] \leq \frac{m \log T}{\eta} + \frac{1}{\eta} \sum_{t=1}^{T} \mathbb{E}_{x \sim q_t}\left[\exp\left(\eta \langle x, \theta_t\rangle\right) - \eta \langle x, \theta_t\rangle - 1\right]$$

by the definition of $q_t$ and by letting $\lambda_t = \eta f(\phi, \theta_t)$. $\qquad\square$

## J    Computationally Efficient Algorithm for Adversarial $C$ Bound

In this section, we present Algorithm 6, a polynomial-time algorithm that ensures $\widetilde{\mathcal{O}}(d^3\sqrt{T} + d^{\frac{5}{2}}C)$ regret. The algorithm is based on the continuous exponential weight algorithm in the original feature space (Ito et al., 2020; Zimmert and Lattimore, 2022), with the bonus construction similar to Lee et al. (2020).

### J.1 Preliminaries for Entropic Barrier

**Entropic barrier** For any convex body $\mathcal{A}$, the family of exponential distribution is

$$p_w(x) = \frac{\exp(w^\top x)\mathbb{1}\{y \in \mathcal{A}\}}{\int_{\mathcal{A}} \exp(w^\top y)\mathrm{d}y}.$$

For any $x \in \mathcal{A}$, there is a unique $w(x)$ such that $\mathbb{E}_{y \sim p_{w(x)}}[y] = x$. The entropic barrier $F(x)$ is the negative entropy of $p_{w(x)}$. Namely

$$F(x) = \int p_{w(x)}(y) \log\left(p_{w(x)}(y)\right) \mathrm{d}y$$

We have $\nabla F(x) = w(x)$ and $\nabla^2 F(x) = \mathbb{E}_{y \sim p_{w(x)}}\left[(y-x)(y-x)^\top\right]$. We know that $F(x)$ is a $d$-self-concordant barrier on $\mathcal{A}$.

**The equivalence of mean-oriented FTRL and continuous exponential weights** Consider FTRL with entropic barrier as the regularizer that solves $x_t$ for round $t \in [T]$ following

$$x_{t+1} = \operatorname*{argmax}_{x \in \mathcal{A}} \left\{ \left\langle x, \sum_{s=1}^{t} \theta_s \right\rangle - \frac{F(x)}{\eta_t} \right\}.$$

This is equivalent to

$$\nabla F(x_{t+1}) = \eta_t \sum_{s=1}^{t} \theta_s.$$

Given that $\mathbb{E}_{y \sim p_{w(x_{t+1})}}[y] = x_{t+1}$ and $\nabla F(x_{t+1}) = w(x_{t+1})$, playing $x_{t+1}$ yields the same expected reward as playing according to distribution $p_{w(x_{t+1})}$ where $w(x_{t+1}) = \eta_t \sum_{s=1}^{t} \theta_s$. Thus, we have $p_{w(x_{t+1})}(x) \propto \exp\left(\eta_t \left\langle x, \sum_{s=1}^{t} \theta_s \right\rangle\right)$ for $x \in \mathcal{A}$.

### J.2 Auxiliary Lemmas

**Lemma J.1** (Lemma 1 of Ito et al. (2020)). *If $x$ follows a log-concave distribution $p$ over $\mathbb{R}^d$ and $\mathbb{E}_{x \sim p}[xx^\top] \preceq I$, we have*

$$\Pr\left[\|x\|_2^2 \geq d\beta^2\right] \leq d\exp(1-\beta).$$

*for arbitrary $\beta > 0$.*

**Lemma J.2.** *With the choice of $\beta \geq 4\log(10dT)$, we have*

$$|\mathbb{E}_{a \sim p_t}[f(a)] - \mathbb{E}_{a \sim \tilde{p}_t}[f(a)]| \leq 10d\exp(-\beta) \leq \frac{1}{2T^2}$$

*for any $f : \mathcal{A} \to [-1, 1]$ and*

$$\frac{3}{4}\mathbb{E}_{a \sim p_t}[aa^\top] \preceq \mathbb{E}_{a \sim \tilde{p}_t}[aa^\top] \preceq \frac{4}{3}\mathbb{E}_{a \sim p_t}[aa^\top].$$

*Proof.* The proof follows that of Lemma 4 of Ito et al. (2020), with the observation that $p_t$ is a log-concave distribution. $\qquad\square$

**Lemma J.3** (Lemma 14 of Zimmert and Lattimore (2022)). *Let $f$ be a $\nu$-self-concordant barrier for $\mathcal{A} \subset \mathbb{R}^d$. Then for any $u, x \in \mathcal{A}$,*

$$\|u - x\|_{\nabla^2 f(x)} \leq -\gamma' \langle u - x, \nabla f(x) \rangle + 4\gamma'\nu + 2\sqrt{\nu}$$

*where $\gamma' = \frac{8}{3\sqrt{3}} + \frac{7^{\frac{3}{2}}}{6\sqrt{3\nu}}$ ($\gamma' \in [1, 4]$ for $\nu \geq 1$).*

**Minkowsky Functions.** The Minkowsky function of a convex boday $\mathcal{A}$ with the pole at $w \in \mathrm{int}(\mathcal{A})$ is a function $\pi_w : \mathcal{A} \to \mathbb{R}$ defined as

$$\pi_w(u) = \inf\left\{ t > 0 \,\middle|\, w + \frac{u-w}{t} \in \mathcal{A} \right\}. \tag{25}$$

**Algorithm 6:** Continuous exponential weights (for adversarial $C$ bound)

---

**1** Let $\mathcal{A} \subset \mathbb{R}^d$ be a convex body and $F$ be its entropic barrier.

**2** **Parameters**: $\gamma = \frac{\log(T/\delta)}{T}$, $\alpha = \tilde{\Theta}(\sqrt{d}C + d\sqrt{T})$, $\eta = \min\left\{\frac{1}{160\sqrt{d^3 T}}, \frac{1}{32\sqrt{d}\alpha}\right\}$.

**3** **for** $t = 1, 2, \ldots, T$ **do**

**4** $\quad$ Define $w_t(a) = \exp\left(\eta \sum_{s=1}^{t-1}\langle a, \widehat{\theta}_s - b_s\rangle\right)$ and

$$p_t(a) = w_t(a)/\left(\int_{y\in\mathcal{A}} w_t(y)\mathrm{d}y\right), \quad \tilde{p}_t(a) = \frac{p_t(a)\mathbb{1}\{\|a\|_{\Sigma_t^{-1}} \leq \sqrt{d}\beta\}}{\int_{a'\in\mathcal{A}} p_t(a')\mathbb{1}\{\|a'\|_{\Sigma_t^{-1}} \leq \sqrt{d}\beta\}\mathrm{d}a'},$$

$\quad$ where $\Sigma_t = \mathbb{E}_{a\sim p_t}[aa^\top]$.

**5** $\quad$ Play $a_t \sim \tilde{p}_t$, and observe reward $r_t$ with $\mathbb{E}[r_t] = a_t^\top\theta_t + \epsilon_t(a_t)$.

**6** $\quad$ Construct reward estimator $\widehat{\theta}_t = \widetilde{\Sigma}_t^{-1}a_t r_t$, where $\widetilde{\Sigma}_t = \gamma I + \mathbb{E}_{a\sim\tilde{p}_t}[aa^\top]$.

**7** $\quad$ Define $\boldsymbol{B}_t = I + \begin{bmatrix} \widetilde{\Sigma}_t^{-1} & -\widetilde{\Sigma}_t^{-1}x_t \\ -x_t^\top\widetilde{\Sigma}_t^{-1} & x_t^\top\widetilde{\Sigma}_t^{-1}x_t \end{bmatrix}$, where $x_t = \mathbb{E}_{a\sim p_t}[a]$.

**8** $\quad$ **if** $\lambda_{\max}(\boldsymbol{B}_t - \sum_{\tau\in\mathcal{I}}\boldsymbol{B}_s) > 0$ **then**

**9** $\quad\quad$ $\mathcal{I} \leftarrow \mathcal{I} \cup \{t\}$.

**10** $\quad\quad$ $b_t = -\alpha\nabla F(x_t)$ where $x_t = \mathbb{E}_{a\sim p_t}[a]$ and $\nabla F(x_t) = \eta\sum_{s=1}^{t-1}(\widehat{\theta}_s - b_s)$.

**11** $\quad$ **else** $b_t = 0$.

---

**Lemma J.4** (Proposition 2.3.2 in Nesterov and Nemirovskii (1994)). *Let $f$ be a $\nu$-self-concordant barrier on $\mathcal{A} \subseteq \mathbb{R}^d$, and $u, w \in \mathrm{int}(\mathcal{A})$. Then*

$$f(u) - f(w) \leq \nu\log\left(\frac{1}{1 - \pi_w(u)}\right).$$

### J.3 Regret Analysis

We perform regret decomposition. For regret comparator $u^\star \in \mathcal{A}$, define $x^\star = \min_{x\in\mathcal{A}} F(x)$ and $u = (1 - \frac{1}{T})u^\star + \frac{1}{T}x^\star$. With probability at least $1 - \delta$,

$$
\begin{aligned}
\mathrm{Reg}_T &= \sum_{t=1}^{T}\langle u^\star - a_t, \theta_t\rangle \\
&= \sum_{t=1}^{T}\langle u - a_t, \theta_t\rangle + \frac{1}{T}\sum_{t=1}^{T}\langle u^\star - x^\star, \theta_t\rangle \\
&= \sum_{t=1}^{T}\langle u - \tilde{x}_t, \theta_t\rangle + \mathcal{O}\left(\sqrt{T\log(1/\delta)}\right) + 2 \\
&\qquad\qquad\qquad\qquad\qquad \text{(define } \tilde{x}_t = \mathbb{E}_{a\sim\tilde{p}_t}[a] \text{ and by Azuma's inequality)} \\
&= \sum_{t=1}^{T}\langle u - x_t, \theta_t\rangle + \sum_{t=1}^{T}\langle x_t - \tilde{x}_t, \theta_t\rangle + \mathcal{O}\left(\sqrt{T\log(1/\delta)}\right) \\
&= \underbrace{\sum_{t=1}^{T}\left\langle u - x_t, \theta_t - \mathbb{E}_t[\widehat{\theta}_t]\right\rangle}_{\textbf{Bias}} + \underbrace{\sum_{t=1}^{T}\left\langle u - x_t, \mathbb{E}_t[\widehat{\theta}_t] - \widehat{\theta}_t\right\rangle}_{\textbf{Deviation}} + \underbrace{\sum_{t=1}^{T}\left\langle u - x_t, \widehat{\theta}_t + b_t\right\rangle}_{\textbf{FTRL}} \\
&\quad \underbrace{- \sum_{t=1}^{T}\langle u - x_t, b_t\rangle}_{\textbf{Bonus}} + \gamma T + \mathcal{O}\left(\sqrt{T\log(1/\delta)}\right).
\end{aligned}
\tag{26}
$$

By standard FTRL analysis, we have

$$\mathbf{FTRL} \le \underbrace{\frac{F(u) - \min_{x \in \mathcal{A}} F(x)}{\eta}}_{\textbf{Penalty}} + \underbrace{\mathbb{E}\left[\sum_{t=1}^{T} \max_{x \in \mathcal{A}} \left\{ \left\langle x - x_t, \widehat{\theta}_t + b_t \right\rangle - \frac{1}{\eta} D_F(x, x_t) \right\}\right]}_{\textbf{Stability}}. \quad (27)$$

The individual terms **Bias**, **Deviation**, **Bonus**, **Penalty**, **Stability** terms are bounded in Lemma J.6, Lemma J.7, Lemma J.9, Lemma J.10, Lemma J.12.

**Lemma J.5.** *For any $t \in [T]$, if $a \sim p_t$, then with probability of at least $1 - \delta$,*

$$\|a\|_{\Sigma_t^{-1}} \le \sqrt{d} \log\left(\frac{3d}{\delta}\right).$$

*Proof.* Define $y = \Sigma_t^{-\frac{1}{2}} a$. Then $\mathbb{E}_y\left[yy^\top\right] = \Sigma_t^{-\frac{1}{2}} \mathbb{E}_{a \sim p_t}[aa^\top] \Sigma_t^{-\frac{1}{2}} = I$. Since $p_t$ is a log-concave distribution, and log-concavity is preserved under liner transformation, $y$ is also log-concave. Applying Lemma J.1 on it leads to

$$\Pr\left[\|a\|_{\Sigma_t^{-1}}^2 \ge d\beta^2\right] = \Pr\left[\|y\|_2^2 \ge d\beta^2\right] \le d\exp(1 - \beta) \le 3d\exp(-\beta).$$

Setting $\delta = 3d\exp(-\beta)$, we conclude that with probability at least $1 - \delta$, $\|a\|_{\Sigma_t^{-1}}^2 \le d\log\left(\frac{3d}{\delta}\right)^2$. $\quad\square$

**Lemma J.6.** *With probability at least $1 - \mathcal{O}(\delta)$,*

$$\mathbf{Bias} \le \left(\max_t \|x_t - u\|_{\widetilde{\Sigma}_t^{-1}}\right)\left(\sqrt{d}\gamma T + 2\sqrt{T\log(1/\delta)} + \sqrt{d}\beta C\right).$$

*Proof.* The proof is the same as that of Lemma H.1. $\quad\square$

**Lemma J.7.**
$$\mathbf{Deviation} \le \mathcal{O}\left(\max_t \|u - x_t\|_{\widetilde{\Sigma}_t^{-1}} d\beta\sqrt{T}\log(T/\delta)\right).$$

*Proof.* The proof is the same as that of Lemma H.2. $\quad\square$

**Lemma J.8.**
$$|\mathcal{I}| \le d\log_2\left(\frac{4T}{\gamma}\right).$$

*Proof.* Our proof is similar to Lemma B.12 in Lee et al. (2020). Let $\{t_1, \cdots, t_{n+1}\}$ be the rounds such that $b_t \ne 0$. Define $\boldsymbol{A}_i = \sum_{j=1}^{i} \boldsymbol{B}_{t_j}$. For any $i > 1$, since $\lambda_{\max}\left(\boldsymbol{B}_{t_i} - \boldsymbol{A}_{i-1}\right) > 0$, there exists a vector $y \in \mathbb{R}^{d+1}$ such that $y^\top \boldsymbol{B}_{t_i} y > y^\top \boldsymbol{A}_{i-1} y$. Thus, $y^\top \boldsymbol{A}_i y \ge 2y^\top \boldsymbol{A}_{i-1} y$. Let $z = \boldsymbol{A}_{i-1}^{\frac{1}{2}} y$, we have $z^\top \boldsymbol{A}_{i-1}^{-\frac{1}{2}} \boldsymbol{A}_i \boldsymbol{A}_{i-1}^{-\frac{1}{2}} z \ge 2\|z\|_2^2$. This implies $\lambda_{\max}\left(\boldsymbol{A}_{i-1}^{-\frac{1}{2}} \boldsymbol{A}_i \boldsymbol{A}_{i-1}^{-\frac{1}{2}}\right) \ge 2$. Moreover, we have $\lambda_{\min}\left(\boldsymbol{A}_{i-1}^{-\frac{1}{2}} \boldsymbol{A}_i \boldsymbol{A}_{i-1}^{-\frac{1}{2}}\right) \ge 1$ because

$$\boldsymbol{A}_{i-1}^{-\frac{1}{2}} \boldsymbol{A}_i \boldsymbol{A}_{i-1}^{-\frac{1}{2}} = \boldsymbol{A}_{i-1}^{-\frac{1}{2}}\left(\boldsymbol{A}_{i-1} + \boldsymbol{B}_{t_i}\right)\boldsymbol{A}_{i-1}^{-\frac{1}{2}} \succeq I.$$

Thus, $\frac{\det(\boldsymbol{A}_i)}{\det(\boldsymbol{A}_{i-1})} = \det\left(\boldsymbol{A}_{i-1}^{-\frac{1}{2}} \boldsymbol{A}_i \boldsymbol{A}_{i-1}^{-\frac{1}{2}}\right) \ge 2$. By induction, we have $\det\left(\boldsymbol{A}_{n+1}\right) \ge 2^n \det\left(\boldsymbol{A}_1\right)$. We now give a upper bound for $\frac{\det(\boldsymbol{A}_{n+1})}{\det(\boldsymbol{A}_1)}$. Define $\boldsymbol{a} = \begin{bmatrix} a \\ 1 \end{bmatrix}$. By AM-GM inequality, we have

$$\det\left(\boldsymbol{A}_{n+1} \boldsymbol{A}_1^{-1}\right) = \det\left(\sum_{j=1}^{n+1} \boldsymbol{B}_{t_j} \boldsymbol{B}_{t_1}^{-1}\right) \le \left(\frac{1}{d}\mathrm{Tr}\left(\sum_{j=1}^{n+1} \boldsymbol{B}_{t_j} \boldsymbol{B}_{t_1}^{-1}\right)\right)^d.$$

Notice that for any $t$, $\boldsymbol{B}_t \succeq I$ and $\text{Tr}(\boldsymbol{B}_t) = \text{Tr}(I) + \text{Tr}(\widetilde{\Sigma}_t^{-1}) + \|x_t\|_{\widetilde{\Sigma}_t^{-1}}^2 \leq \frac{2(d+1)}{\gamma}$. Thus, we can upper bound the last expression further by

$$\left(\frac{1}{d}\text{Tr}\left(\sum_{j=1}^{n+1}\boldsymbol{B}_{t_j}\right)\right)^d \leq \left(\frac{2(d+1)(n+1)}{d\gamma}\right)^d \leq \left(\frac{4T}{\gamma}\right)^d.$$

Overall, we have $2^n \leq \frac{\det(\boldsymbol{A}_{n+1})}{\det(\boldsymbol{A}_1)} \leq (4T/\gamma)^d$, and thus $n \leq d\log_2(4T/\gamma)$.

$\square$

**Lemma J.9.**

$$\textbf{Bonus} \leq -\frac{\alpha}{8}\max_t \|u - x_t\|_{\widetilde{\Sigma}_t^{-1}} + \mathcal{O}\left(\alpha d^2 \log T\right).$$

*Proof.* Let $\rho = \max_t \|u - x_t\|_{\widetilde{\Sigma}_t^{-1}}$ and $t^* = \text{argmax}_t \|u - x_t\|_{\widetilde{\Sigma}_t^{-1}}$, We discuss two conditions:

- If $t^* \in \mathcal{I}$, then $\rho^2 \leq \sum_{\tau \in \mathcal{I}} \|u - x_t\|_{\widetilde{\Sigma}_\tau^{-1}}^2$.

- If $t^* \notin \mathcal{I}$, then $\boldsymbol{B}_{t^\star} \preceq \sum_{\tau \in \mathcal{I}} \boldsymbol{B}_\tau$. Let $\boldsymbol{u} \triangleq \begin{bmatrix} u \\ 1 \end{bmatrix}$. This implies

$$\rho^2 = \|u - x_{t^*}\|_{\widetilde{\Sigma}_{t^*}^{-1}}^2 = \|\boldsymbol{u}\|_{\boldsymbol{B}_{t^\star}}^2 \leq \sum_{\tau \in \mathcal{I}} \|\boldsymbol{u}\|_{\boldsymbol{B}_\tau}^2 = \sum_{\tau \in \mathcal{I}} \|u - x_\tau\|_{\widetilde{\Sigma}_\tau^{-1}}^2,$$

where we use the definitions of $\boldsymbol{B}_t$ and $\boldsymbol{u}$ in the second and the last equality.

Thus, $\max_t \|u - x_t\|_{\widetilde{\Sigma}_t^{-1}} \leq \sum_{\tau \in \mathcal{I}} \|u - x_\tau\|_{\widetilde{\Sigma}_\tau^{-1}}$.

$$\sum_{t=1}^{T} \langle x_t - u, b_t \rangle$$
$$= \sum_{\tau \in \mathcal{I}} \langle x_\tau - u, b_\tau \rangle$$
$$= \alpha \sum_{\tau \in \mathcal{I}} \langle x_\tau - u, -\nabla F(x_\tau) \rangle$$
$$\leq -\frac{\alpha}{\gamma'} \sum_{\tau \in \mathcal{I}} \|u - x_\tau\|_{\nabla^2 F(x_\tau)} + 4\alpha d|\mathcal{I}| + \frac{2\alpha\sqrt{d}|\mathcal{I}|}{\gamma'}$$
$$\qquad\qquad\qquad\qquad\qquad (\text{Lemma J.3 and } F \text{ is } d\text{-self-concordant barrier})$$
$$\leq -\frac{\alpha}{2\gamma'} \sum_{\tau \in \mathcal{I}} \|u - x_\tau\|_{\widetilde{\Sigma}_\tau^{-1}} + \mathcal{O}(\alpha d^2 \log T) \qquad (\nabla^2 F(x_\tau) = \Sigma_\tau^{-1} \succeq \tfrac{1}{4}\widetilde{\Sigma}_\tau^{-1} \text{ and Lemma J.8})$$
$$\leq -\frac{\alpha}{2\gamma'} \max_t \|u - x_t\|_{\widetilde{\Sigma}_t^{-1}} + \mathcal{O}(\alpha d^2 \log T).$$

$\square$

**Lemma J.10.**

$$\textbf{Penalty} \leq \frac{d\log(T)}{\eta}.$$

*Proof.* Since $x^\star = \min_{x \in \mathcal{A}} F(x)$ and $\pi_{x^\star}(u) \leq 1 - \frac{1}{T}$ from Eq. (25). We have from Lemma J.4

$$\textbf{Penalty} = \frac{F(u) - F(x^\star)}{\eta} \leq \frac{d\log(T)}{\eta}.$$

$\square$

**Lemma J.11** (Lemma 17 in Zimmert and Lattimore (2022))**.** *Let $F$ be the entropic barrier and* $\|w\|_{\nabla^2 F(x_t)^{-1}} \le \frac{1}{16\eta}$, *then*

$$\max_{x \in \mathcal{A}} \left\{ \langle x - x_t, w \rangle - \frac{1}{\eta} D_F(x, x_t) \right\} \le 2\eta \|w\|_{\nabla^2 F(x_t)^{-1}}^2.$$

**Lemma J.12.** *With probability at least $1 - \delta$,*

$$\textbf{Stability} \le \mathcal{O} \left( \eta \beta^2 dT + \eta \alpha^2 d^2 \log T \right).$$

*Proof.* Since $F$ is a $d$-self-concordant barrier (Chewi, 2023), we have

$$\|b_t\|_{\nabla^2 F(x_t)^{-1}} = \alpha \|\nabla F(x_t)\|_{\nabla^2 F(x_t)^{-1}} \le \alpha \sqrt{d}.$$

By Lemma J.2, we have $\widetilde{\Sigma}_t^{-1} \preceq \left( \mathbb{E}_{a \sim \tilde{p}_t}[aa^\top] \right)^{-1} \preceq 2\Sigma_t^{-1}$, and thus

$$\|\widehat{\theta}_t\|_{\nabla^2 F(x_t)^{-1}}^2 = \|\widetilde{\Sigma}_t^{-1} a_t r_t\|_{\Sigma_t}^2 \le 2a_t^\top \widetilde{\Sigma}_t^{-1} a_t \le 2d\beta^2.$$

Thus, $\|\widehat{\theta}_t + b_t\|_{\nabla^2 F(x_t)^{-1}} \le \beta\sqrt{2d} + \alpha\sqrt{d}$. If $\eta \le \frac{1}{16(\beta\sqrt{2d} + \alpha\sqrt{d})}$, by Lemma J.11, we have

$$\begin{aligned}
\textbf{Stability} &\le 2\eta \sum_{t=1}^T \|\widehat{\theta}_t + b_t\|_{\nabla^2 F(x_t)^{-1}}^2 \\
&\le 4\eta \sum_{t=1}^T \|\widehat{\theta}_t\|_{\nabla^2 F(x_t)^{-1}}^2 + 4\eta \sum_{\tau \in \mathcal{I}} \|b_\tau\|_{\nabla^2 F(x_\tau)^{-1}}^2 \\
&\le \mathcal{O} \left( \eta \beta^2 dT + \eta \alpha^2 d |\mathcal{I}| \right) \\
&\le \mathcal{O} \left( \eta \beta^2 dT + \eta \alpha^2 d^2 \log T \right)
\end{aligned}$$

$\square$

**Theorem J.13.** *Algorithm 6 ensures with probability at least $1 - \delta$, $\mathrm{Reg}_T = \widetilde{\mathcal{O}}\left( d^3 \sqrt{T} + d^{\frac{5}{2}} C \right)$, where $\widetilde{\mathcal{O}}(\cdot)$ hides polylog$(T/\delta)$ factors.*

*Proof.* Putting Lemma J.6, Lemma J.7, Lemma J.9, Lemma J.10, Lemma J.12 into Eq. (26) and Eq. (27), with $\eta \le \frac{1}{16(\beta\sqrt{2d} + \alpha\sqrt{d})}$ and $\gamma = \frac{1}{T}$, we have with probability at least $1 - \mathcal{O}(\delta)$,

$$\mathrm{Reg} \le \max_t \|u - x_t\|_{\widetilde{\Sigma}_t^{-1}} \left( \widetilde{\mathcal{O}}(d\sqrt{T} + \sqrt{d}C) - \frac{\alpha}{8} \right) + \widetilde{\mathcal{O}} \left( \alpha d^2 + \frac{d}{\eta} + \eta \alpha^2 d^2 + \eta dT + \sqrt{T} \right).$$

By setting $\frac{\alpha}{8} = \tilde{\Theta}(d\sqrt{T} + \sqrt{d}C)$, we have

$$\mathrm{Reg} \le \widetilde{\mathcal{O}} \left( d^3 \sqrt{T} + d^{\frac{5}{2}} C + \frac{d}{\eta} + \eta d^4 T + \eta d^3 C^2 \right).$$

Setting $\eta = \frac{1}{160 d\sqrt{T} + 32\alpha\sqrt{d}}$, we get

$$\mathrm{Reg} \le \widetilde{\mathcal{O}} \left( d^3 \sqrt{T} + d^{\frac{5}{2}} C \right).$$

$\square$

# K  Gap-dependent Misspecification

We consider the same setting as Liu et al. (2023a), but remove an assumption for it. Consider bandit learning with general reward function $f_0$ where for any action $x_t \in \mathcal{X} \subset \mathbb{R}^d$ at round $t$, the learner get reward $y_t = f_0(x_t) + \eta_t$ where $\eta_t$s are zero mean, $\sigma$-sub-Gaussian noise. We assume there exists a linear function $\theta^\top x$ that could approximate $f_0(x)$ in the following manner.

**Definition 3.**

$$\sup_{x\in\mathcal{X}}\left|\frac{\theta^\top x - f_0(x)}{f_0^\star - f_0(x)}\right| \le \rho$$

where $f_0^\star = \max_{x\in\mathcal{X}} f_0(x)$ and $0 \le \rho < 1$.

The algorithm in Liu et al. (2023a) only gets $\mathcal{O}\left(\sqrt{T}\right)$ regret when $\rho \le \frac{1}{d\sqrt{\log T}}$ and we improve it to $\rho \le \frac{1}{\sqrt{d}}$ by using elimination-based methods in Algorithm 7. For any design $\pi$ on action set $\mathcal{A}$, define

$$G(\pi) = \sum_{a\in\mathcal{A}} \pi(a)aa^\top \qquad g(\pi) = \max_{a\in\mathcal{A}} \|a\|^2_{G(\pi)^{-1}}$$

---

**Algorithm 7:** Phased Elimination for Misspecification

---

1 **Input:** Action set $\mathcal{A}_1 = \mathcal{A}$. Initialize $m_1 = \lceil 64d \log\log d \log\left(\frac{|\mathcal{A}|}{\delta}\right)\rceil + 16$.

2 **for** $\ell = 1, 2, \cdots, L$ **do**

3     Find the approximate G-optimal design $\pi_\ell$ on $\mathcal{A}_\ell$ with $g(\pi) \le 2d$ and
    $|\text{Supp}(\pi)| \le 4d\log\log d + 16$

4     Compute $u_\ell(a) = \lceil m_\ell \pi_\ell(a)\rceil$ and $u_\ell = \sum_{a\in A_\ell} u_\ell(a)$

5     Take each action $a \in \mathcal{A}_\ell$ exactly $u(a)$ times with reward $y(a)$.

6     Calculate

$$\widehat{\theta}_\ell = G_\ell^{-1} \sum_{a\in A_\ell} u(a)ay(a) \quad \text{where} \quad G_\ell = \sum_{a\in A_\ell} u(a)aa^\top$$

    Update active action set

$$\mathcal{A}_{\ell+1} = \left\{ a \in A_\ell : \max_{b\in\mathcal{A}_\ell}\langle\widehat{\theta}_\ell, b\rangle - \langle\widehat{\theta}_\ell, a\rangle \le \sqrt{\frac{4d}{m_\ell}\log\left(\frac{|\mathcal{A}|}{\delta}\right)} + \frac{1}{2^\ell} \right\}$$

    $m_{\ell+1} \leftarrow 4m_\ell$

---

Define $\text{Gap}(x) = f_0^\star - f_0(x)$ as the suboptimal gap at point $x$. Definition 3 implies the true value function $f_0(x) = \theta^\top x + \Delta(x)$ where $|\Delta(x)| \le \rho(f_0^\star - f_0(x)) = \rho\text{Gap}(x)$. We further assume that $|\Delta(x)| \le \rho\text{Gap}(x)$ which captures both standard uniform misspecification and the gap-dependent misspecification. With this assumption, our main result is summarized in Theorem K.1.

**Theorem K.1.** *For action $a$, assume $y(a) = f_0(a) + \eta_a$ where $\eta_a$ is zero-mean sub-gaussian noise and $f_0(a) = \theta^\top a + \Delta(a)$ with $|\Delta(a)| \le \rho\text{Gap}(a)$. If $\rho \le \frac{1}{64\sqrt{d}}$, with probability of at least $1 - \delta$, we have*

$$\text{Reg}_T^{\mathcal{M}^\star} \le \mathcal{O}\left(\sqrt{dT \log|\mathcal{A}|/\delta}\right)$$

*Proof.* First, with probability of at least $1 - \delta$, for any $\ell$ and $b \in \mathcal{A}_\ell$, we have

$$\left|\langle b, \widehat{\theta}_\ell - \theta\rangle\right| = \left|b^\top G_\ell^{-1} \sum_{a\in A_\ell} u(a)ay(a) - b^\top\theta\right|$$

$$= \left|b^\top G_\ell^{-1} \sum_{a\in A_\ell} u(a)aa^\top\eta_a + b^\top G_\ell^{-1} \sum_{a\in A_\ell} u(a)a\Delta(a)\right|$$

$$\le \sqrt{\frac{4d}{m_\ell}\log\left(\frac{|\mathcal{A}|}{\delta}\right)} + \left|b^\top G_\ell^{-1} \sum_{a\in A_\ell} u(a)a\Delta(a)\right|$$

where in the last step, we use standard concentration by Equation (20.2) of Lattimore and Szepesvári (2020) and the apply union bound for all actions.

For the last term, we have

$$\left| b^\top G_\ell^{-1} \sum_{a \in A_\ell} u(a) a \right| \le \max_{c \in \mathcal{A}_\ell} \Delta(c) \cdot \sqrt{\left( \sum_{a \in A_\ell} u(a) \right) b^\top G_\ell^{-1} \sum_{a \in A_\ell} u(a) a a^\top \Sigma_\ell^{-1} b}$$

$$\text{(Cauchy-Schwarz)}$$

$$= \max_{c \in \mathcal{A}_\ell} \Delta(c) \cdot \sqrt{u \|b\|_{G_\ell^{-1}}^2} \le \max_{c \in \mathcal{A}_\ell} \Delta(c) \cdot \sqrt{\frac{2du}{m_\ell}} \le \max_{c \in \mathcal{A}_\ell} \Delta(c) \cdot 2\sqrt{d}.$$

Thus, for any $b \in \mathcal{A}_\ell$,

$$\left| \langle b, \widehat{\theta}_\ell - \theta \rangle \right| \le \sqrt{\frac{4d}{m_\ell} \log\left( \frac{|\mathcal{A}|}{\delta} \right)} + 2\sqrt{d} \max_{a \in \mathcal{A}_\ell} \Delta(a) = \sqrt{\frac{4d}{m_\ell} \log\left( \frac{|\mathcal{A}|}{\delta} \right)} + 2\sqrt{d}\rho \max_{a \in \mathcal{A}_\ell} \text{Gap}(a)$$

When $\ell = 1$, since $m_1 = \lceil 256 d \log \log d \log\left( \frac{|\mathcal{A}|}{\delta} \right) \rceil + 16$, we have $\sqrt{\frac{4d}{m_1} \log\left( \frac{|\mathcal{A}|}{\delta} \right)} \le \frac{1}{2^4}$. Moreover, by trivial bound, $\max_{a \in \mathcal{A}_1} \text{Gap}(a) \le 2$ and $a^\star \in \mathcal{A}_1$.

We will jointly do two inductions. Assume for round $\ell$, we have $a^\star \in \mathcal{A}_\ell$ and $\max_{a \in \mathcal{A}_\ell} \text{Gap}(a) \le \frac{1}{2^{\ell-2}}$. We first show $a^\star \in \mathcal{A}_{\ell+1}$. Thus, for any $b \in \mathcal{A}_\ell$, given $\rho \le \frac{1}{64\sqrt{d}}$, since $m_\ell = 4^{\ell-1} m_1$, we have

$$\left| \langle b, \widehat{\theta}_\ell - \theta \rangle \right| \le \sqrt{\frac{4d}{m_\ell} \log\left( \frac{|\mathcal{A}|}{\delta} \right)} + 2\sqrt{d}\rho \max_{a \in \mathcal{A}_\ell} \text{Gap}(a) \le \frac{1}{2^{\ell-1}} \frac{1}{2^4} + \frac{1}{2^{\ell+3}} = \frac{1}{2^{\ell+2}}$$

From the induction hypothesis, let $\widehat{a}_\ell = \arg\max_{b \in \mathcal{A}_\ell} \langle \widehat{\theta}_\ell, b \rangle$ we have

$$\widehat{\theta}_\ell^\top \widehat{a}_\ell - \widehat{\theta}_\ell^\top a^\star \le \theta^\top \widehat{a}_\ell - \theta^\top a^\star + \underbrace{\widehat{\theta}_\ell^\top \widehat{a}_\ell - \theta^\top \widehat{a}_\ell}_{\le \frac{1}{2^{\ell+2}}} + \underbrace{\theta^\top a^\star - \widehat{\theta}_\ell^\top a^\star}_{\le \frac{1}{2^{\ell+2}}}$$

$$\le \underbrace{f_0(\widehat{a}_\ell) - f_0(a^\star)}_{\le 0} + |\Delta(\widehat{a}_\ell)| + \frac{1}{2^{\ell+1}}$$

$$\le \rho \text{Gap}(\widehat{a}_\ell) + \frac{1}{2^{\ell+1}} \le \frac{1}{2^\ell}$$

For $\ell + 1$, the remaining actions $a \in \mathcal{A}_{\ell+1}$ satisfy

$$\max_{b \in \mathcal{A}_\ell} \langle \widehat{\theta}_\ell, b \rangle - \langle \widehat{\theta}_\ell, a \rangle \le \sqrt{\frac{4d}{m_\ell} \log\left( \frac{|\mathcal{A}|}{\delta} \right)} + \frac{1}{2^\ell} \le \frac{1}{2^{\ell+3}} + \frac{1}{2^\ell}$$

This implies $a^\star \in \mathcal{A}_{\ell+1}$. Moreover, since $a^\star \in \mathcal{A}_\ell$, for $a \in \mathcal{A}_{\ell+1}$, we have

$$\text{Gap}(a) = f_0^\star - f_0(a) = \theta^\top a^\star - \theta^\top a + |\Delta(a)|$$

$$\le \widehat{\theta}_\ell^\top a^\star - \widehat{\theta}_\ell^\top a + \rho\text{Gap}(a) + (\theta - \widehat{\theta}_\ell)^\top a^\star + (\widehat{\theta}_\ell - \theta)^\top a$$

$$\le \widehat{\theta}_\ell^\top a^\star - \widehat{\theta}_\ell^\top a + \rho\text{Gap}(a) + (\theta - \widehat{\theta}_\ell)^\top a^\star + (\widehat{\theta}_\ell - \theta)^\top a$$

$$\le \underbrace{\widehat{\theta}_\ell^\top a^\star - \widehat{\theta}_\ell^\top \widehat{a}_\ell}_{\le 0 \text{ given } a^\star \in \mathcal{A}_\ell} + \underbrace{\widehat{\theta}_\ell^\top \widehat{a}_\ell - \widehat{\theta}_\ell^\top a}_{\le \frac{1}{2^{\ell+3}} + \frac{1}{2^\ell}} + \rho\text{Gap}(a) + \frac{1}{2^{\ell+1}}$$

Given $\rho \le \frac{1}{64\sqrt{d}} \le \frac{1}{64}$, this implies

$$\text{Gap}(a) \le \frac{1}{1-\rho}\left( \frac{1}{2^\ell} + \frac{1}{2^{\ell+1}} + \frac{1}{2^{\ell+3}} \right) \le \frac{63}{64}\left( \frac{1}{2^\ell} + \frac{1}{2^{\ell+1}} + \frac{1}{2^{\ell+3}} \right) \le \frac{1}{2^{\ell-1}}$$

The above arguments show that as $\ell$ increases, $\max_{a \in \mathcal{A}_\ell} \text{Gap}(a)$ will shrink by $\frac{1}{2}$ at every step. Since for $a \in \mathcal{A}_\ell$, $\text{Gap}(a) \le \frac{1}{2^{\ell-1}} = \mathcal{O}\left( \sqrt{\frac{4d}{m_\ell} \log\left( \frac{|\mathcal{A}|}{\delta} \right)} \right)$

Finally, given $L = \log(T)$, we have

$$\text{Reg} = \sum_{\ell=1}^{L} \sum_{a \in \mathcal{A}_\ell} u_\ell(a) \text{Gap}(a) \leq \sum_{\ell=1}^{L} m_\ell \sqrt{\frac{4d}{m_\ell} \log\left(\frac{|\mathcal{A}|}{\delta}\right)} \leq \mathcal{O}\left(\sqrt{dT \log |\mathcal{A}|/\delta}\right)$$

$\square$

When $|\mathcal{A}| \geq T^d$, we can apply similar covering number arguments as in Appendix E, replacing $\mathcal{A}_1$ with a $\frac{1}{T}$-net of $\mathcal{A}$. Combined with Theorem K.1, this yields the result in Theorem 6.2.

Using the hard instance for $\epsilon$-misspecified linear bandits setting in Lattimore et al. (2020), we now show that $\rho = \tilde{\Omega}(\frac{1}{\sqrt{d}})$ for an algorithm to achieve sub-linear regret, proving the above algorithm is optimal in terms of $\rho$ assumption.

**Theorem K.2.** *If $\rho \geq \sqrt{\frac{8 \log(3T)}{d-1}}$ then there exists an instance that $R_T = \Omega(\rho T)$.*

*Proof.* Using Theorem F.5 in Lattimore et al. (2020), there exist a discrete time-invariant action space $\{a_i \in \mathbb{R}^d\}_{i=1}^{3T}$ that satisfies these two conditions:

1. $\|a_i\| = 1 \quad \forall i$

2. $\langle a_i, a_j \rangle \leq \sqrt{\frac{8 \log(3T)}{d-1}} \quad \forall i \neq j$

and let $\theta^* = \sqrt{\frac{d-1}{8 \log(3T)}} \epsilon a_{i^*}$ for some $i^*$, and let misspecification at each round for all non-optimal arms be $\epsilon$ to make the true expected reward zero. Defining $\tau := \max(t | i_s \neq i^* \quad \forall s \leq t)$, we have $\mathbb{E}[R_T] \geq \sqrt{\frac{d-1}{8 \log(3T)}} \epsilon \mathbb{E}[\tau]$. Since the observed rewards are independent of $a_{i^*}$ before time $\tau$, and $i^*$ is chosen randomly, we have $\mathbb{E}[\tau] \geq \min\{T, \frac{3T-1}{2}\}$. So,

$$\mathbb{E}[R_T] \geq \epsilon T \sqrt{\frac{d-1}{8 \log(3T)}}$$

Finally, we have $\rho \geq \frac{\epsilon}{\sqrt{\frac{d-1}{8 \log(3T)}} \epsilon - 0} = \sqrt{\frac{8 \log(3T)}{d-1}}$, so choosing $\epsilon = \min(\sqrt{\frac{8 \log(3T)}{d-1}}, \sqrt{\frac{d-1}{8 \log(3T)}})$ completes the proof showing linear regret when $\rho$ is large enough. $\square$

## L   General Reduction from Corruption-Robust Algorithms to Misspecification

In this section, we extend the results of Section 6 to the reinforcement learning setting. We consider episodic MDPs, denoted by a tuple $\mathcal{M} = (\mathcal{S}, \mathcal{A}, \{P_h\}_{h=1}^{H}, \{r_h\}_{h=1}^{H}, s_1)$ for $\mathcal{S}$ the set of states, $\mathcal{A}$ the set of actions, $P_h : \mathcal{S} \times \mathcal{A} \to \triangle_{\mathcal{S}}$ the transition kernel, $r_h : \mathcal{S} \times \mathcal{A} \to \triangle_{[0,1]}$ the reward, and $s_1$ the starting state. We assume each episode starts in state $s_1$, where the agent takes action $a_1$, transitions to $s_2 \sim P_1(\cdot \mid s_1, a_1)$ and receives reward $r_1 \sim r_1(s_1, a_1)$. This proceeds for $H$ steps at which point the episode terminates and the process resets. We assume that $\sum_{h=1}^{H} r_h \in [0, 1]$ almost surely (note that the linear bandit setting with rewards in [-1,1] can be incorporated into this with a simple rescaling).

We let $\pi$ denote a policy, $\pi_h : \mathcal{S} \to \triangle_{\mathcal{S}}$, a mapping from states to actions. We denote the value of a policy $\pi$ on MDP $\mathcal{M}$ as $V_0^{\mathcal{M},\pi} := \mathbb{E}^{\mathcal{M},\pi}[\sum_{h=1}^{H} r_h]$. We assume access to some function class $\mathcal{F} \subseteq \{\mathcal{S} \times \mathcal{A} \to \mathbb{R}\}$. In the MDP setting, we define regret on MDP $\mathcal{M}$ as:

$$\text{Reg}_T^{\mathcal{M}} := T \cdot \sup_\pi V_0^{\mathcal{M},\pi} - \sum_{t=1}^{T} V_0^{\mathcal{M},\pi_t}.$$

In the MDP setting, we consider the following notion of misspecification.

**Definition 4** (Misspecification). *For our environment of interest $\mathcal{M}^\star$, there exists some environment $\mathcal{M}_0$ such that, for each $f_{h+1} \in \mathcal{F}$, $\pi$, and $(s, a, h)$, we have:*

$$\left| \mathbb{E}^{\mathcal{M}^\star, \pi}[r_h + f_{h+1}(s_{h+1}, a_{h+1}) \mid s_h = s, a_h = a] \right.$$

$$\left. - \mathbb{E}^{\mathcal{M}_0, \pi}[r_h + f_{h+1}(s_{h+1}, a_{h+1}) \mid s_h = s, a_h = a] \right| \leq \epsilon_h^{\mathrm{mis}}(s, a)$$

*and*

$$\exists f_h \in \mathcal{F} \text{ s.t. } f_h(s, a) = \mathbb{E}^{\mathcal{M}_0}[r_h + \max_{a'} f_{h+1}(s_{h+1}, a') \mid s_h = s, a_h = a]$$

*for some $\epsilon_h^{\mathrm{mis}}(s, a) > 0$.*

We make the following assumption on *gap-dependent* misspecification.

**Assumption 4** (Gap-Dependent Misspecification). *For any policy $\pi$, we have*

$$\mathbb{E}^{\mathcal{M}^\star, \pi}\left[ \sum_{h=1}^{H} \epsilon_h^{\mathrm{mis}}(s_h, a_h) \right] \leq \rho \cdot \Delta(\pi)$$

*for some $\rho \geq [0, 1)$.*

We are interested in relating the above misspecification setting to the corruption-robust setting. In the MDP setting, we allow both the reward and transitions to be corrupted. For some MDP $\mathcal{M}$, define the corruption at episode $t$ and step $h$ as:

$$\epsilon_{t,h}(s_h^t, a_h^t) := \sup_{g \in \{\mathcal{S} \times \mathcal{A} \to [0, H]\}} |(\mathcal{T}^h g - \mathcal{T}_b^h g)(s_h^t, a_h^t)|$$

where

$$\mathcal{T}^h g(s, a) := \mathbb{E}^{\mathcal{M}}[r_h + \max_{a'} g(s_{h+1}, a') \mid s_h = s, a_h = a]$$

denotes the Bellman operator, and $\mathcal{T}_b^h$ denotes the corrupted Bellman operator, i.e. $\mathcal{T}_b^h$ denotes the expected reward and next state under the corrupted reward and transition distribution. We denote the total corruption level as

$$C := \sum_{t=1}^{T} \sum_{h=1}^{H} \epsilon_{t,h}(s_h^t, a_h^t).$$

Note that this definition of corruption encompasses both bandits and RL with function approximation.

Now assume we have access to the following oracle.

**Assumption 5.** *We have access to a regret minimization algorithm which takes as input $\mathcal{F}$ and some $C'$ and with probability at least $1 - \delta$ has regret bounded on $\mathcal{M}_0$ as*

$$\mathrm{Reg}_T^{\mathcal{M}_0} \leq \mathcal{C}_1(\delta, T)\sqrt{T} + \mathcal{C}_2(\delta, T)C' \tag{28}$$

*if $C' \geq C$, and by $HT$ otherwise, for $C$ as defined above and for (problem-dependent) constants $\mathcal{C}_1(\delta, T), \mathcal{C}_2(\delta, T)$ which may scale at most logarithmically with $T$ and $\frac{1}{\delta}$.*

Before stating our main reduction from corruption-robust to gap-dependent misspecification, we require the following assumption.

**Assumption 6.** *For any $\pi$, we have that there exists some $f \in \mathcal{F}$ such that for all $(s, a, h)$, $Q_h^{\mathcal{M}_0, \pi}(s, a) = f_h(s, a)$.*

We then have the following result.

**Theorem L.1.** *Assume our environment satisfies Assumption 4. Then under Assumption 5 and Assumption 6, as long as $\frac{\rho \mathcal{C}_2(\frac{\delta}{T}, T)}{1 - \rho} \leq 1/2$, with probability at least $1 - 2\delta$ we can achieve regret bounded as:*

$$\mathrm{Reg}_T^{\mathcal{M}^\star} \leq \frac{3}{1 - \rho} \cdot \mathcal{C}_1(\tfrac{\delta}{T}, T)\sqrt{T} + \frac{2}{1 - \rho} \cdot \left( H\sqrt{2T \log(1/\delta)} + H \right).$$

*Proof of Theorem L.1.* First, note that by Assumption 4, we can bound

$$\sum_{t=1}^{T} \mathbb{E}^{\mathcal{M}^\star, \pi_t} \left[ \sum_{h=1}^{H} \epsilon_h^{\mathrm{mis}}(s_h, a_h) \right] \leq \sum_{t=1}^{T} \rho \cdot \Delta(\pi_t) \leq \rho \cdot \mathrm{Reg}_T$$

where we abbreviate $\mathrm{Reg}_T := \mathrm{Reg}_T^{\mathcal{M}^\star}$. Furthermore, note that under Assumption 4 interacting with $\mathcal{M}^\star$ is equivalent to interacting with $\mathcal{M}_0$ but where the rewards and transitions are corrupted up to level $\epsilon_h^{\mathrm{mis}}(s, a)$ at $(s, a, h)$.

**Relating Regret on $\mathcal{M}_0$ to $\mathcal{M}^\star$.** Define the regret on $\mathcal{M}_0$ as

$$\mathrm{Reg}_T^{\mathcal{M}_0} := T \cdot \sup_{\pi} \mathbb{E}^{\mathcal{M}_0, \pi} \left[ \sum_{h=1}^{H} r_h \right] - \sum_{t=1}^{T} \mathbb{E}^{\mathcal{M}_0, \pi_t} \left[ \sum_{h=1}^{H} r_h \right].$$

Under Assumption 4, we have that $\mathbb{E}^{\mathcal{M}^\star, \pi^\star} \left[ \sum_{h=1}^{H} \epsilon_h^{\mathrm{mis}}(s_h, a_h) \right] = 0$. Lemma L.2 then implies that

$$\mathbb{E}^{\mathcal{M}^\star, \pi^\star} \left[ \sum_{h=1}^{H} r_h \right] = \mathbb{E}^{\mathcal{M}_0, \pi^\star} \left[ \sum_{h=1}^{H} r_h \right]$$

and so

$$\mathbb{E}^{\mathcal{M}^\star, \pi^\star} \left[ \sum_{h=1}^{H} r_h \right] \leq \sup_{\pi} \mathbb{E}^{\mathcal{M}_0, \pi} \left[ \sum_{h=1}^{H} r_h \right].$$

Furthermore, Lemma L.2 also implies

$$\left| \mathbb{E}^{\mathcal{M}_0, \pi_t} \left[ \sum_{h=1}^{H} r_h \right] - \mathbb{E}^{\mathcal{M}^\star, \pi_t} \left[ \sum_{h=1}^{H} r_h \right] \right| \leq \mathbb{E}^{\mathcal{M}^\star, \pi_t} \left[ \sum_{h=1}^{H} \epsilon_h^{\mathrm{mis}}(s_h, a_h) \right].$$

Putting these together we can bound

$$\mathrm{Reg}_T \leq \mathrm{Reg}_T^{\mathcal{M}_0} + \sum_{t=1}^{T} \mathbb{E}^{\mathcal{M}^\star, \pi_t} \left[ \sum_{h=1}^{H} \epsilon_h^{\mathrm{mis}}(s_h, a_h) \right] \leq \mathrm{Reg}_T^{\mathcal{M}_0} + \rho \cdot \mathrm{Reg}_T,$$

where the last inequality holds by Assumption 4. Rearranging this gives

$$\mathrm{Reg}_T \leq \frac{1}{1 - \rho} \cdot \mathrm{Reg}_T^{\mathcal{M}_0}.$$

**Bounding the Regret.** Consider running the algorithm of Assumption 5 on $\mathcal{M}^\star$ and assume we run with parameter $C' \leftarrow \beta$ which we will choose shortly. From the above observation, this is equivalent to running on $\mathcal{M}_0$ with corruption level $\epsilon_h^{\mathrm{mis}}(s, a)$ at $(s, a, h)$. Then by Assumption 5, with probability at least $1 - \delta$ we have regret on $\mathcal{M}_0$ bounded as

$$\mathrm{Reg}_T^{\mathcal{M}_0} \leq \mathcal{C}_1(\delta, T)\sqrt{T} + \mathcal{C}_2(\delta, T)\beta$$

if $\beta \geq \sum_{t=1}^{T} \sum_{h=1}^{H} \epsilon_h^{\mathrm{mis}}(s_h^t, a_h^t)$, and by $HT$ otherwise. Furthermore, by the above argument this then immediately implies a regret bound on $\mathrm{Reg}_T$.

Let $\mathcal{E}_{1,t}$ denote the event that $\{\beta \geq \sum_{t'=1}^{t} \sum_{h=1}^{H} \epsilon_h^{\mathrm{mis}}(s_h^{t'}, a_h^{t'})\}$. Let $\mathcal{E}_2$ denote the event that for all $t \leq T$, we have

$$\mathrm{Reg}_t \leq \frac{1}{1 - \rho} \cdot \left( \mathcal{C}_1(\tfrac{\delta}{T}, T)\sqrt{t} + \mathcal{C}_2(\tfrac{\delta}{T}, T)\beta \right) + \frac{TH}{1 - \rho} \cdot \mathbb{I}\{\mathcal{E}_{1,t}^c\},$$

and note that by the above and under Assumption 5 we then have that $\mathcal{E}_2$ occurs with probability at least $1 - \delta$. For simplicity, for the remainder of the proof we abbreviate $\mathcal{C}_1 := \mathcal{C}_1(\tfrac{\delta}{T}, T)$ and $\mathcal{C}_2 := \mathcal{C}_2(\tfrac{\delta}{T}, T)$.

Note that $\epsilon_h(s_h^t, a_h^t) \in [0, H]$ by construction. It follows that, with probability at least $1 - \delta$, via Azuma-Hoeffding,

$$\sum_{t=1}^{T} \sum_{h=1}^{H} \epsilon_h^{\mathrm{mis}}(s_h^t, a_h^t) \le \sum_{t=1}^{T} \mathbb{E}^{\pi_t} \left[ \sum_{h=1}^{H} \epsilon_h^{\mathrm{mis}}(s_h^t, a_h^t) \right] + H\sqrt{2T \log 1/\delta} \le \rho \cdot \mathrm{Reg}_T + H\sqrt{2T \log 1/\delta}.$$

Denote this event as $\mathcal{E}_3$.

Now consider choosing

$$\beta = \left( 1 - \frac{\rho \mathcal{C}_2}{1 - \rho} \right)^{-1} \cdot \left( \frac{\rho}{1 - \rho} \cdot \mathcal{C}_1 \sqrt{T} + H\sqrt{2T \log 1/\delta} + H \right)$$

so that

$$\beta = \frac{\rho}{1 - \rho} \cdot \left( \mathcal{C}_1 \sqrt{T} + \mathcal{C}_2 \beta \right) + H\sqrt{2T \log 1/\delta} + H.$$

On $\mathcal{E}_2 \cap \mathcal{E}_3$, assume that

$$\beta < \rho \cdot \mathrm{Reg}_T + H\sqrt{2T \log 1/\delta}. \tag{29}$$

Let $t^\star$ denote the minimum time such that

$$\sum_{t=1}^{t^\star} \rho \Delta(\pi_t) + H\sqrt{2T \log 1/\delta} > \beta \quad \text{and} \quad \sum_{t=1}^{t^\star - 1} \rho \Delta(\pi_t) + H\sqrt{2T \log 1/\delta} \le \beta,$$

and note that such a time is guaranteed to exist under (29) and since $\beta \ge H\sqrt{2T \log 1/\delta} + H$ by construction so $\rho \Delta(\pi_1) + H\sqrt{2T \log 1/\delta} \le H + H\sqrt{2T \log 1/\delta} \le \beta$. Furthermore, since $\Delta(\pi) \le H$, we have here that $\sum_{t=1}^{t^\star - 1} \rho \Delta(\pi_t) > \beta - \rho H - H\sqrt{2T \log 1/\delta}$. We then have

$$\begin{aligned}
\mathrm{Reg}_{t^\star - 1} &= \sum_{t=1}^{t^\star - 1} \Delta(\pi_t) \\
&> \frac{\beta}{\rho} - H - \frac{H}{\rho}\sqrt{2T \log 1/\delta} \\
&= \frac{1}{1 - \rho} \cdot \left( \mathcal{C}_1 \sqrt{T} + \mathcal{C}_2 \beta \right) \\
&\ge \frac{1}{1 - \rho} \cdot \left( \mathcal{C}_1 \sqrt{t^\star - 1} + \mathcal{C}_2 \beta \right).
\end{aligned}$$

However, since by assumption $\sum_{t=1}^{t^\star - 1} \rho \Delta(\pi_t) + H\sqrt{2T \log 1/\delta} \le \beta$, on $\mathcal{E}_3$ $\mathcal{E}_{1, t^\star - 1}$ holds so on $\mathcal{E}_2 \cap \mathcal{E}_3$ we have that

$$\mathrm{Reg}_{t^\star - 1} \le \frac{1}{1 - \rho} \cdot \left( \mathcal{C}_1 \sqrt{t^\star - 1} + \mathcal{C}_2 \beta \right).$$

This contradicts the above. Therefore, on $\mathcal{E}_2 \cap \mathcal{E}_3$ we must have that $\beta \ge \rho \cdot \mathrm{Reg}_T + H\sqrt{2T \log 1/\delta}$, so $\mathcal{E}_{1, T}$ holds on $\mathcal{E}_3$, and so on $\mathcal{E}_2 \cap \mathcal{E}_3$,

$$\mathrm{Reg}_T \le \frac{1}{1 - \rho} \cdot \left( \mathcal{C}_1 \sqrt{T} + \mathcal{C}_2 \beta \right).$$

From our setting of $\beta$ we can bound this as

$$\le \frac{1}{1 - \rho} \cdot \mathcal{C}_1 \sqrt{T} + \frac{1}{1 - \rho} \cdot \mathcal{C}_2 \cdot \left( 1 - \frac{\rho \mathcal{C}_2}{1 - \rho} \right)^{-1} \cdot \left( \frac{\rho}{1 - \rho} \cdot \mathcal{C}_1 \sqrt{T} + H\sqrt{2T \log 1/\delta} + H \right).$$

The result follows from some simplification.

$\square$

**Lemma L.2.** *For MDPs $\mathcal{M}^\star, \mathcal{M}_0$ satisfying Definition 4, under Assumption 6 we have*

$$V_0^{\mathcal{M}_0, \pi} - V_0^{\mathcal{M}^\star, \pi} \le \mathbb{E}^{\mathcal{M}^\star, \pi} \left[ \sum_{h=1}^{H} \epsilon_h^{\mathrm{mis}}(s_h, a_h) \right].$$

*Proof.* Let $r'$ denote the reward function on $\mathcal{M}_0$, and note that under Assumption 6 we have that there exists $f \in \mathcal{F}$ such that $V_h^{\mathcal{M}_0,\pi}(s) = f_h(s, \pi_h(s))$ for all $\pi, s, h$. Then Lemma E.15 of Dann et al. (2017) gives that

$$V_0^{\mathcal{M}_0,\pi} - V_0^{\mathcal{M}^\star,\pi} = \mathbb{E}^{\mathcal{M}^\star,\pi}\left[\sum_{h=1}^H (r'_h - r_h)\right.$$
$$\left. + \sum_{h=1}^H \mathbb{E}^{\mathcal{M}_0,\pi}[V_h^{\mathcal{M}_0,\pi}(s_{h+1}) \mid s_h] - \mathbb{E}^{\mathcal{M}^\star,\pi}[V_h^{\mathcal{M}_0,\pi}(s_{h+1}) \mid s_h]\right].$$

By Definition 4, we can bound this as

$$\leq \mathbb{E}^{\mathcal{M}^\star,\pi}\left[\sum_{h=1}^H \epsilon_h^{\mathrm{mis}}(s_h, a_h)\right].$$

$\square$

*Proof of Corollary 6.2.1.* First, note that under Assumption 3, we have that Assumption 4 holds for $\mathcal{F}$ the set of functions linear in $\phi$, $\mathcal{F} = \{\phi(s,a)^\top w \ : \ w \in \mathbb{R}^d \text{ s.t. } \phi(s,a)^\top w \in [0, H], \forall s, a\}$, and $\epsilon_h^{\mathrm{mis}}(s,a)$ of Assumption 4 set to $H\epsilon_h^{\mathrm{mis}}(s,a)$ for $\epsilon_h^{\mathrm{mis}}(s,a)$ of Assumption 3. To see this, let $\mathcal{M}_0$ be the MDP with transitions $\langle \phi(s,a), \mu_h(\cdot)\rangle$, and note that this the immediately implies linear realizability on $\mathcal{M}_0$ (and furthermore that Assumption 6 holds). Furthermore, since the total reward is at most $H$, it is easy to see that under Assumption 3, we can take $\epsilon_h^{\mathrm{mis}}(s,a) \leftarrow H\epsilon_h^{\mathrm{mis}}(s,a)$.

Next, note that Theorem 4.2 of Ye et al. (2023) gives an algorithm on $\mathcal{M}_0$ satisfying Assumption 5 with $\mathcal{C}_1 = \widetilde{\mathcal{O}}(\sqrt{H^2 d^3})$ and $\mathcal{C}_2 = \widetilde{\mathcal{O}}(Hd)$ (assuming that $\sum_{h=1}^H r_h \in [0, 1]$ almost surely). We can then apply Theorem L.1 to obtain the result.

$\square$

# M   Auxiliary Lemmas

**Lemma M.1** (Lemma 16 of Zimmert and Lattimore (2022)). *Let* $\boldsymbol{X} = \begin{bmatrix} X + xx^\top & x \\ x^\top & 1 \end{bmatrix}$ *and* $\boldsymbol{Y} = \begin{bmatrix} Y + yy^\top & y \\ y^\top & 1 \end{bmatrix}$. *Then*

$$D_G(\boldsymbol{X}, \boldsymbol{Y}) = D_G(X, Y) + \|x - y\|_{Y^{-1}}^2 \geq \|x - y\|_{Y^{-1}}^2.$$

**Lemma M.2** (Lemma 34 of Liu et al. (2023b)). *Let $G$ be the log-determinant barrier. For any matrix $\boldsymbol{D}$, if $\sqrt{\mathrm{Tr}(\boldsymbol{H}_t \boldsymbol{D} \boldsymbol{H}_t \boldsymbol{D})} \leq \frac{1}{16\eta}$, then*

$$\max_{\boldsymbol{H} \in \mathcal{H}} \langle \boldsymbol{H} - \boldsymbol{H}_t, \boldsymbol{D}\rangle - \frac{D_G(\boldsymbol{H}, \boldsymbol{H}_t)}{\eta} \leq 8\eta \mathrm{Tr}(\boldsymbol{H}_t \boldsymbol{D} \boldsymbol{H}_t \boldsymbol{D}).$$

**Lemma M.3** (Strengthened Freedman's inequality (Theorem 9 of Zimmert and Lattimore (2022))). *Let $X_1, X_2, \ldots, X_T$ be a martingale difference sequence with a filtration $\mathcal{F}_1 \subseteq \mathcal{F}_2 \subseteq \cdots$ such that $\mathbb{E}[X_t|\mathcal{F}_t] = 0$ and $\mathbb{E}[|X_t| \mid \mathcal{F}_t] < \infty$ almost surely. Then with probability at least $1 - \delta$,*

$$\sum_{t=1}^T X_t \leq 3\sqrt{V_T \log\left(\frac{2\max\{U_T, \sqrt{V_T}\}}{\delta}\right)} + 2U_T \log\left(\frac{2\max\{U_T, \sqrt{V_T}\}}{\delta}\right),$$

*where $V_T = \sum_{t=1}^T \mathbb{E}[X_t^2 \mid \mathcal{F}_t]$ and $U_T = \max\{1, \max_{t \in [T]} |X_t|\}$.*

