# OpenReview forum: "Corruption-Robust Linear Bandits: Minimax Optimality and Gap-Dependent Misspecification"
_NeurIPS.cc/2024/Conference — NeurIPS 2024 poster_

### Official Review · Reviewer_DxTV · 2024-07-09

**Soundness:** 2
**Presentation:** 2
**Contribution:** 2
**Rating:** 5
**Confidence:** 3

**Summary:**

This paper studies corruption-robust linear bandit optimization and characterizes the regret bound in terms of both weak and strong corruption measures. Under the stochastic setting, this paper proposes a phased elimination algorithm, and the regret bounds match the lower bound. Under the adversarial setting, the paper proposes two individual algorithms for the two corruption measures respectively. In addition, this paper studies gap-dependent misspecification setting through reduction, and discusses a use case for linear MDPs.

**Strengths:**

- The regret bounds in terms of both corruption measures are provided, where the regret bound depending on $C_\infty$ is first introduced in this paper.
- The theoretical results are supported with detailed proof.
- This paper is generally well-written.

**Weaknesses:**

- The algorithms are efficient regarding regret bound, but the computational complexity is not discussed.
- A conclusion section could be added.

**Questions:**

What is the computational cost of the proposed algorithms?

**Limitations:**

Some limitations are discussed.

---

> ### Author Rebuttal · Authors · 2024-08-07
>
> We thank the reviewer for the valuable feedback. The global response includes a summary of our paper and potential future work, which we will incorporate as a conclusion in the future version. Your questions are answered below.
>
>
> **Q1**:  What is the computational cost of the proposed algorithms?
>
> **A**:  All our proposed algorithms could be computed in polynomial time except Algorithm 2 because it requires solving a fixed point optimization defined in Line 6 of Algorithm 2. As a remedy, we propose Algorithm 4, which gives a computationally efficient counterpart of Algorithm 2 with a worse $dC_{\infty}$ regret.
>
> To be more specific, the main computational cost in Algorithm 1 is to get the G-optimal design and the per-step computational complexity roughly has order $|\mathcal{A}| d^2$ from lemma 3.9 of [1] (also discussed after Theorem 4.3 of [2]).
>
> For Algorithm 3, the main computational cost is solving the continuous exponential weights defined in Line 4. Assuming we get access to linear optimization oracle, the per-step computational complexity has order $poly(dT)$ as discussed on Page 6 of [3].
>
> For Algorithm 4, the main computational cost is solving the logdet optimization in Line 6 and the per-step computational complexity has order $d^4|\mathcal{A}|$ from proposition 1 of [4].
>
> Note that the $|\mathcal{A}|$ dependence is common in most linear bandit algorithms including LinUCB and phase elimination. More discussion on the $|\mathcal{A}|$-dependent computational complexity for existing linear bandit algorithms can be found in Section 1.2 of [5].
>
>
> **References**:
>
> [1] M. J. Todd. Minimum-volume ellipsoids: Theory and algorithms, volume 23. SIAM, 2016.
>
> [2] Lattimore, T., Szepesvari, C., and Weisz, G. Learning with good feature representations in bandits and in RL with a generative model. ICML, 2020.
>
> [3] Zimmert, J. and Lattimore, T. Return of the bias: Almost minimax optimal high probability bounds for adversarial linear bandits. COLT, 2022.
>
> [4] Dylan J Foster, Claudio Gentile, Mehryar Mohri, and Julian Zimmert. Adapting to misspecification in contextual bandits. NeurIPS, 2020
>
> [5] Liu, H., Wei, C.-Y., and Zimmert, J. Bypassing the simulator: Near-optimal adversarial linear contextual bandits. NeurIPS,2023

---

> > ### Author Response · Authors · 2024-08-12
> >
> > We thank the reviewer for the time and effort spent reviewing our paper. As the discussion phase is about to end, we would like to make sure our responses have sufficiently addressed your concerns. We look forward to your feedback.

---

> > > ### Comment · Reviewer_DxTV · 2024-08-13
> > > **Response to Rebuttal**
> > >
> > > Thanks for your response, and I have no further questions.

---

### Official Review · Reviewer_TGCM · 2024-07-13

**Soundness:** 3
**Presentation:** 3
**Contribution:** 3
**Rating:** 5
**Confidence:** 3

**Summary:**

In this work, the authors characterize the problem of learning the presence of reward corruption in the linear bandit setting. They provide matching upper and lower bounds in the corrupted stochastic setting, and initiate the study on the corrupted adversarial setting, for which they obtain optimal scaling in the corruption level.

Not only that, the authors prove a general reduction that efficiently handles gap-dependent misspecification with corruption-robust algorithms. They were able to show that linear MDPs with gap-dependent misspecification are efficiently learnable. While this reduction is general, interestingly they denied the possibility to obtain the tightest rate for gap-dependent misspecification. This observation leads them to develop a specialized algorithm which, in the linear bandit setting, obtains the optimal rate. According to their argument, this resolves the open problem of Liu et al. (2023a).

**Strengths:**

- Interesting results
    - Deterministic algorithm cannot avoid suboptimal regret (Proposition 1)
    - Matching upper and lower bound on the stochastic setting, by just changing deterministic sampling to stochastic.
    - Solving an open problem of instance-dependent misspecified setting.

- Clearly state the limitations of previous works and their technical novelties.
    - Easy to understand their contributions.

**Weaknesses:**

- (Minor) The algorithms are not seriously different from the previous works as they mentioned, but this is just a minor point - every theoretical improvement is important.

- Not clear what they tried to say on page 9
    - Why Theorem 6.2 shows that $\rho \leq \frac{1}{d}$ is not optimal?
    - Impossibility result (from line 304): so basically what authors are trying to say is, that 'their' reduction is not applicable for a tighter result, right? It is not about any reduction from corruption to misspecification.

- No future works.

**Questions:**

- Unclear points in the weakness section.

- It would be great if authors could explain why the gap-dependent misspecification assumption (Assumption 1) is necessary.

### Minor

- Theorem G.1 in line 296 - is it correct?

- Corollary 6.2.1 in line 209 - it seems like it is the result for the MDP...

**Limitations:**

The authors adequately addressed the limitations. It would be great if authors provide possible future works.
There is no potential negative societal impact of their work.

---

> ### Author Rebuttal · Authors · 2024-08-07
>
> We thank the reviewer for the valuable feedback. The global response includes potential future works, which we will incorporate into the future version. Your questions are answered below.
>
> **Q1**: The algorithms are not seriously different from the previous works as they mentioned
>
>
> **A**: Although our algorithmic framework inherits from previous works, the corresponding modifications offer new insights. For example, the impossible results in Proposition 1 for the deterministic algorithm lead to our randomized design in Algorithm 1. Moreover, our geometric-inspiring bonus design in Algorithm 2 does not appear in previous papers and it is the key to the tight $\sqrt{d} C_{\infty}$ regret. We believe such bonus design ideas could also inspire future work.
>
> **Q2**: Why Theorem 6.2 shows that $\rho \leq \frac{1}{d}$ is not optimal?
>
> **A**: In Theorem 6.2, we show that there exists an algorithm with which $\rho \leq \frac{1}{\sqrt{d}}$ suffices to ensure $d\sqrt{T}$ regret. This implies that the requirement $\rho \leq \frac{1}{d}$ is "too strong", and thus not optimal. The $\frac{1}{\sqrt{d}}$ factor, on the other hand, is optimal because of the lower bound given in Theorem G.2 (discussed in line 296-303).
>
>
> **Q3**:  Impossibility result (from line 304): so basically what authors are trying to say is, that 'their' reduction is not applicable for a tighter result, right? It is not about any reduction from corruption to misspecification.
>
> **A**: Yes, the impossibility result only shows our reduction is not optimal. However, we generally believe that no reductions can guarantee $\rho \le \frac{1}{\sqrt{d}}$ suffices for $d\sqrt{T}$ regret, due to the $dC$ lower bound for corruption. This will be clarified in our revision.
>
> **Q4**: Why the gap-dependent misspecification assumption (Assumption 1) is necessary?
>
> **A**: Assumption 1 comes from Liu et al. (2023a), with more motivation provided in its Introduction. Existing work on misspecification tends to assume the misspecification level is uniform over all actions, which typically yields overly pessimistic results (a regret linear in the misspecification level).  Assumption 1 is the only known assumption beyond realizability that is sufficient to obtain the minimax $d\sqrt{T}$ regret, as in the unmisspecified case. Understanding precisely what conditions on misspecification are necessary to achieve $d\sqrt{T}$ regret is an interesting open question.
>
> We also believe that this assumption is often reasonable in real-world applications. In practice, approximating a decision-making problem as a linear bandit typically requires extensive modeling and feature selection to choose the linearization. In settings that are not perfectly linear, one would naturally focus on obtaining an accurate linearization for "good" or "typical" users, and would likely be willing to tolerate a less accurate model for "bad" or "atypical" users. This is precisely the intuition that is captured in Assumption 1.
>
> **Q5**: Theorem G.1 in line 296 - is it correct?
>
> **A**: Thank you for pointing that out. It is a typo and should be Theorem 6.2. We have fixed it.
>
> **Q6**: Corollary 6.2.1 in line 290 - it seems like it is the result for the MDP...
>
> **A**: Thank you for pointing that out. It is a typo and should be Corollary 6.1.1. We have fixed it.

---

> > ### Author Response · Authors · 2024-08-12
> >
> > We thank the reviewer for the time and effort spent reviewing our paper. As the discussion phase is about to end, we would like to make sure our responses have sufficiently addressed your concerns. We look forward to your feedback.

---

> > > ### Comment · Reviewer_TGCM · 2024-08-12
> > >
> > > Thank you for your details. Now I understand the points of the authors. I will keep my score as it is.

---

### Official Review · Reviewer_QWPG · 2024-07-14

**Soundness:** 3
**Presentation:** 2
**Contribution:** 3
**Rating:** 6
**Confidence:** 3

**Summary:**

This paper studied the corrupted linear bandits. The authors propose four different metrics to evaluate the total corruption in Eq. (1). Many settings are considered in this paper. For stochastic LB, the proposed algorithm achieves a regret bound of $d\sqrt{T}+\sqrt{d} C_{\infty}$. For adversarial LB, the proposed algorithm achieves a regret bound in the order of $d\sqrt{T}+\sqrt{d} C_{\infty}$ or $d^{3}\sqrt{T}+d^{5/2} C$. The authors also consider the gap-dependent misspecification, where the misspecification level of an arm $a$ can be evaluated by $\rho$ times the gap of arm $a$.

**Strengths:**

See summary.

**Weaknesses:**

**Weaknesses and Questions:**
1. At lines 107-109, the authors claim that the strong adversary is equivalent to the CM viewpoint. This doesn't seem right. For regret, the strong adversary is harder than the CM viewpoint. Thus, it is unfair and wrong to compare He et al. (2022) in the same way.
2. At line 131, adversarial linear bandits are discussed. However, no problem definition of this problem is introduced before line 131.
3. This paper studies the fixed action set, while the previous works He et al. (2022) and Foster et al. (2020) allow the action set to be chosen by an adaptive adversary, which is much harder than this paper. Table 1 is not fair. He et al. (2022) is for the adaptive adversarial viewpoint, which is totally different from the stochastic LB. For a fixed action set, the optimal regret without $C$ should be $\sqrt{d T \log k}$, where $k$ is the number of arms.
4. Assumption 1 is not very reasonable.

**Questions:**

See weaknesses.

**Limitations:**

N.A.

---

> ### Author Rebuttal · Authors · 2024-08-07
>
> We thank the reviewer for the valuable feedback. Your questions are answered below.
>
> **Q1**: The strong adversary (AA) seems not equivalent to the CM viewpoint.
>
> **A**: The equivalence between the "strong adversary" in the AA viewpoint and the "strong measure'" in the CM viewpoint is based on the equivalence between the following two processes (note that the discussion here only concerns the adversary in choosing the "corruption"; the discussion about the adversary in choosing the "action set" is discussed in **Q3**):
>
> Process 1 (AA viewpoint):  In every round $t$, the learner first chooses an action $a_t$, then the adversary chooses the corruption $\epsilon_t$. The learner observes $r_t = a_t^\top \theta_t + \epsilon_t + \zeta_t$, where $\zeta_t$ is a zero-mean noise.  Define $C=\sum_{t=1}^T |\epsilon_t|$.
>
> Process 2 (CM viewpoint): In every round $t$, the adversary first chooses the corruption $\epsilon_t(a)$ for every action $a$, then the learner chooses an action $a_t$. The learner observes $r_t = a_t^\top \theta_t +  \epsilon_t(a_t) + \zeta_t$, where  $\zeta_t$ is a zero-mean noise. Define $C=\sum_{t=1}^T |\epsilon_t(a_t)|$.
>
> In Process 1, the adversary chooses the corruption based on the learner's action $a_t$. In Process 2, although the adversary specifies the corruption "before" seeing the learner's action, it captures the same effect as Process 1 because the applied corruption depends on the chosen action $a_t$. In other words, the amount $\epsilon_t(a)$ in Process 2 is a "plan" of the adversary (set before seeing the learner's action $a_t$) for the amount of corruption under the assumption that the learner chooses $a$. After seeing $a_t$, the adversary simply applies the planned corruption $\epsilon_t(a_t)$.
>
> The AA viewpoint is adopted by most previous works on linear bandits (Bogunovic et al., 2021) and linear contextual bandits (He et al., 2022). In this work, we argue that such results are equivalent to the standard adversary (i.e., adaptive adversary) with a fine-grained corruption measure (i.e., $C=\sum_{t=1}^T |\epsilon_t(a_t)|$) in the analysis. We believe this leads to a more unified treatment for the strong and weak adversary widely studied in the literature, which are usually separately discussed.
>
> We hope this clarifies the equivalence between the two viewpoints. If there are still concerns, we would appreciate it if the reviewer could elaborate on the potential mismatch in the difficulty between the two viewpoints.
>
> **Q2**:  No definition of adversarial linear bandits before line 131.
>
> **A**:  The general linear bandit problem is introduced in lines 71-76. Then, in lines 77-79, we describe the stochastic and adversarial settings of it. The term "adversarial linear bandits" in line 131 simply refers to the adversarial setting of linear bandits. We will make this more clear.
>
> **Q3**: This paper studies the fixed action set, while He et al. (2022) and Foster et al. (2020) allow adversarially chosen action set, which is much harder. Table 1 is not fair. He et al. (2022) is for the adaptive adversarial viewpoint, which is different from the stochastic LB.
>
> **A**: In this paper, we focus on linear bandits with fixed action sets. Note that even in this simpler setting, for the stochastic case, there is no prior work achieving $\sqrt{d}C_{\infty}$ regret; for the adversarial case, there is no prior study on
> $C_\infty$ bound. Our work obtains tight $\sqrt{d}C_{\infty}$ regret in both cases.
>
> The purpose of Table 1 is to summarize known $C_{\infty}$ and $C$ bounds in linear bandits. He et al. (2022) is included because their algorithm gives the best $C$ bound in linear bandits, even though it works for the more general linear contextual bandit setting with adversarially chosen action sets. We will clarify this in our next version. In other related work discussions (e.g. Line 120-130), we will also make similar clarification that Foster et al. 2020, Takemura et al. 2021 study the more challenging linear contextual bandit setting.
>
> **Q4**: For a fixed action set, the optimal regret without $C$ should be $\sqrt{dT\log(k)}$, where $k$ is the number of arms.
>
> **A**: Technically, there are two widely adopted lower bounds for linear bandits with fixed action sets. One is $\sqrt{dT\log k}$, which is tight for $k\leq 2^d$; the other is $d\sqrt{T}$, which is tight for $k>2^d$ (see, e.g., Section 24.1, 24.2 of Lattimore and Szepesvari, 2020, for the second lower bound). For simplicity, we adopt the second lower bound. Our analysis for the stochastic setting (Theorem 4.1) can also give the $\sqrt{dT\log k}$ bound straightforwardly. For the adversarial setting, however, we do not obtain $\sqrt{dT\log k}$ though due to the limitation of our base algorithms.
>
> (Lattimore and Szepesvari, 2020)  Bandit Algorithms.
>
> **Q5**: Assumption 1 is not very reasonable.
>
> **A**: Assumption 1 comes from Liu et al. (2023a), with more motivation provided in its Introduction. Existing work on misspecification tends to assume the misspecification level is uniform over all actions, which typically yields overly pessimistic results (a regret linear in the misspecification level).  Assumption 1 is the only known assumption beyond realizability that is sufficient to obtain the minimax $d\sqrt{T}$ regret, as in the unmisspecified case. Understanding precisely what conditions on misspecification are necessary to achieve $d\sqrt{T}$ regret is an interesting open question.
>
> We also believe that this assumption is often reasonable in real-world applications. In practice, approximating a decision-making problem as a linear bandit typically requires extensive modeling and feature selection to choose the linearization. In settings that are not perfectly linear, one would naturally focus on obtaining an accurate linearization for "good" or "typical" users, and would likely be willing to tolerate a less accurate model for "bad" or "atypical" users. This is precisely the intuition that is captured in Assumption 1.

---

> > ### Comment · Reviewer_QWPG · 2024-08-07
> >
> > Thanks for your rebuttal.
> > ---
> >
> > Regarding A1: I disagree. In the AA perspective, the adversary can first observe the chosen arm $A_t$ and then decide whether $A_t$ is the arm it wants to corrupt or not. For instance, if the adversary aims to corrupt a specific dimension of $\theta^*$, it can choose to do so to maximize its profit, given that it knows $A_t$ beforehand. If $A_t$ is not known, the only option for the adversary is to add corruption to each arm and hope the algorithm picks the arm that has a high influence on the dimension it wants to corrupt.
> >
> > ---
> >
> > Regarding A2, the term "adversarial linear bandits" in the bandit literature refers to a different setting. Please consider changing the name of "adversarial linear bandits".
> >
> > ---
> >
> > Regarding A4, I believe that $\sqrt{dT \log k}$ is significantly better than $d\sqrt{T}$ regret. It might be beneficial to present the $\sqrt{dT \log k}$ results in your main theorem and clearly discuss the differences in settings and results compared with previous work.

---

> ### Author Response · Authors · 2024-08-09
>
> Thank you for the comments! We would like to elaborate more as follows.
>
> **Regarding the first concern:**
>
> Our AA and CM viewpoints are exactly what you describe. However, we would like to draw the reviewer's attention to the definition of the corruption measure in the CM viewpoint: $C=\sum_{t=1}^T |\epsilon_t(a_t)|$. This definition only accounts for the corruption for the "chosen action" $a_t$, but not that for other actions $a\neq a_t$.
>
> How does this property allow us to simulate a strong adversary in the AA viewpoint? Below, we illustrate it through a concrete example as in your response (below, we use "he" for the adversary and "she" for the learner). Consider a $d$-dimensional linear bandit with the action set being the canonical basis $e_{1:d}$. From the AA viewpoint, suppose the adversary wants to corrupt the $i$-th dimension of $\theta^*$, and his strategy can be described as the following:
>
> *If $a_t=e_i$ is observed, then set $\epsilon_t=c$*;
>
> *If $a_t\neq e_i$ is observed, then set $\epsilon_t=0$*.
>
> Clearly, this is a strong adversary in the AA viewpoint since the adversary decides whether he wants to corrupt $a_t$, and to what extent he wants to corrupt it, after observing the chosen action $a_t$.
>
>
> From the CM viewpoint, the equivalent adversary would specify the following "corruption function" $\epsilon_t(\cdot)$ *before* seeing $a_t$:
>
> *$\epsilon_t(e_i)=c$, and $\epsilon_t(e_j)=0$ for $j\neq i$*.
>
> After the learner chooses $a_t$, the learner experiences a corruption of $\epsilon_t(a_t)$.
>
> We clarify two points:
>
> 1. The learner has exactly the same observations in the two viewpoints: if she chooses $a_t=e_i$, then she suffers a corruption of $c$; if she chooses $a_t\neq e_i$, then she suffers no corruption.
>
> 2. The total corruption $C$ is the same in the two viewpoints: In the AA viewpoint, $C=\sum_{t=1}^T |\epsilon_t|=\sum_{t=1}^T |c|\mathbf{1}\\{a_t=e_i\\}$. In the CM viewpoint, $C=\sum_{t=1}^T |\epsilon_t(a_t)|=\sum_{t=1}^T |c|\mathbf{1}\\{a_t=e_i\\}$.
>
> One key observation is that: in the CM viewpoint, even though the corruption function might be a non-zero function (i.e., $\epsilon_t(a)\neq 0$ for some $a$), the amount of corruption counted towards $C=\sum_{t=1}^T |\epsilon_t(a_t)|$ is zero as long as $\epsilon_t(a_t)=0$. In other words, it is possible that $\epsilon_t(a)=0$ and $\epsilon_t(b)\neq 0$ for some arms $a,b$, which could simulate the AA strong adversary who decides NOT TO corrupt after observing $a_t=a$, but decides TO corrupt after observing $a_t=b$.
>
> This might address the reviewer's concern, "*If $a_t$ is not known, the only option for the adversary is
> to add corruption to each arm and hope the algorithm picks the arm that has
> a high influence on the dimension it wants to corrupt.*" From the discussion above, we know that a CM adversary can simply do the following: set high corruption for the actions with high influence on the dimension, and set low/zero corruption for the actions with low influence on the dimension. Then, after $a_t$ is decided by the learner, the corruption level $|\epsilon_t(a_t)|$ will naturally adapt to the choice of $a_t$, i.e., the corruption will be high if $a_t$ has high influence on the dimension, and will be low/zero if $a_t$ has low influence on the dimension.
>
> A mathematical conversion between the two viewpoints is as follows. Assume that the strategy of the AA strong adversary can be decribed as a function $\epsilon_t = f(h_{t-1}, a_t)$, where $h_{t-1}$ is the history up to time $t-1$ and $a_t$ is the chosen action at time $t$. Then in the CM viewpoint, the "corruption function" would be defined as $\epsilon_t(a) = f(h_{t-1}, a)$ for every $a$. Notice that the function $\epsilon_t(\cdot)$ depends only on the history up to time $t-1$.
>
>
> **Regarding the second concern:**
>
> If the corruption is always zero, our formulation (Line 74 - 79) is the same as the standard adversarial linear bandit setting formulated in Section 27 of Lattimore and Szepesvari (2020), where the linear reward vector $\theta_t$ changes over time. The only extension we make is that we consider additional reward corruption. We can refer to it as "adversarial linear bandits with corruption" to make things even clearer.
>
> (Lattimore and Szepesvari, 2020) Bandit Algorithms.
>
> **Regarding the third concern:**
>
>  Thanks for your suggestions. We will add the bound $\sqrt{dT \log k}$ to our Theorem 4.1. We will also discuss more about our bounds compared with previous works.

---

> ### Author Response · Authors · 2024-08-11
>
> We thank the reviewer for the time and effort spent reviewing our paper. As the discussion phase is about to end, we would like to make sure our responses have sufficiently addressed your concerns. We look forward to your feedback.

---

> ### Comment · Reviewer_QWPG · 2024-08-13
>
> Thank you for your response.
>
> **Regarding the first concern,** I disagree with your statement. I believe the second point you made is incorrect. For example, with a probability of 1/2, the algorithm selects $a_{t} = e_i$. Assuming both AA and CM have the same influence on the algorithm, and CM has a corruption level $C$, then, since AA only corrupts the algorithm with a probability of 1/2, the corruption level of AA would be $C/2$. I look forward to your reply.

---

> ### Author Response · Authors · 2024-08-13
>
> We thank the reviewer for the feedback.
>
> Notice that in our previous response, we do not assume the type of algorithm the learner uses (i.e., whether it's deterministic or randomized). Therefore, the points we make still hold even when the learner uses a randomized algorithm.
>
> Below we demonstrate that in the example you gave (i.e., the learner chooses $a_t=e_i$ with probability $1/2$), the corruption levels are the same in the two viewpoints.
>
> **AA viewpoint:**
>
> Adversary strategy:  If $a_t=e_i$ is observed, then set $\epsilon_t=c$. If $a_t\neq e_i$ is observed, then set $\epsilon_t=0$.
> Corruption level:  $C=\sum_{t=1}^T |\epsilon_t|=\sum_{t=1}^T |c|\mathbf{1}\\{a_t=e_i\\}$.
> Thus, $\mathbb{E}[C]=\mathbb{E}\big[\sum_{t=1}^T |c|\mathbf{1}\\{a_t=e_i\\}\big]=\frac{T|c|}{2}$, where in the last equality we use $\Pr\\{a_t=e_i\\}=1/2$.
>
> **CM viewpoint:**
>
> Adversary strategy:  Set $\epsilon_t(e_i)=c$, and $\epsilon_t(e_j)=0$ for $j\neq i$ before seeing $a_t$.
> Corruption level:  $C=\sum_{t=1}^T |\epsilon_t(a_t)|=\sum_{t=1}^T |c|\mathbf{1}\\{a_t=e_i\\}$.
> Thus, $\mathbb{E}[C]=\mathbb{E}\big[\sum_{t=1}^T |c|\mathbf{1}\\{a_t=e_i\\}\big]=\frac{T|c|}{2}$, where in the last equality we use $\Pr\\{a_t=e_i\\}=1/2$.
>
> This shows that the corruption levels are the same in the two viewpoints.
>
> **Potential source of confusion:**  We point out that in the CM viewpoint, although there could be ``non-zero corruption function'' in every round (e.g.., $\epsilon_t(e_i)=c$ for all $t$ in our previous example),  the corruption measure we consider is $C = \sum_{t=1}^T |\epsilon_t(a_t)|$, which only depends on $a_t$. Thus, even though $\epsilon_t(e_i) = c$ for every round $t$, such corruption only contributes to $C$ when $a_t = e_i$, and it will not contribute to $C$ when $a_t\neq e_i$. This makes it equivalent to the strong AA adversary whose corruption adapts to the choice of the learner.
>
> We guess that the reviewer might be thinking about another corruption measure $C_\infty=\sum_{t=1}^T \max_a |\epsilon_t(a)|$, which accounts for the corruption for *every* (no matter chosen or unchosen) action.  However, this is not what we want to show equivalence for with the strong AA adversary.

---

> > ### Comment · Reviewer_QWPG · 2024-08-13
> >
> > Thank you for the detailed response. I recommend focusing solely on the Corruption Measure (CM) Viewpoint, as the AA Viewpoint introduces some confusion. The concepts would be more clearly presented without it. This paper makes a valuable contribution to the understanding of corruption across various metrics and presents solid regret bounds. One potential weakness of this paper is that it focuses solely on a fixed action set. I will be increasing the score to 6. Good luck!

---

> > > ### Author Response · Authors · 2024-08-13
> > >
> > > Thanks for your positive feedback and the adjustment for the score. We will make efforts to reduce the confusion to the reader.

---

### Author Rebuttal · Authors · 2024-08-07

## Global Response:
We thank all reviewers for their time and valuable feedback. As suggested, we summarize our paper here together with possible future directions. We will incorporate them into our future versions.

Our paper contributes to three research lines.

1. For stochastic linear bandits with corruption, we design an algorithm that achieves $\sqrt{d}C_{\infty}$ regret for the first time, filling a gap in the existing literature. We also construct a lower bound demonstrating that such regret is unattainable by any deterministic algorithm, which leads to our randomized design.

2. For adversarial bandits with corruption, previous works only focus on multi-armed bandits while we move forward to the more challenging linear bandit settings. We achieve the tight $\sqrt{d}C_{\infty}$ regret and also show it is possible to get $poly(d)C$ regret. The novel geometric-inspiring bonus design in Algorithm 2 does not appear in previous papers and it is the key to the tight $\sqrt{d}C_{\infty}$ regret. We believe such a bonus design idea could also inspire future work.

3. For gap-dependent misspecification in linear bandits, Liu et al. (2023a) give an algorithm showing that $\rho \le \frac{1}{d}$ is sufficient to achieve $d\sqrt{T}$ regret, and ask what is the optimal rate of $\rho$. We fully resolve this open problem by designing an algorithm that achieves $d\sqrt{T}$ regret with $\rho \le \frac{1}{\sqrt{d}}$ (Theorem 6.2), and provide a matching lower bound (Theorem G.2). Going beyond linear bandits, we show that such gap-dependent misspecification could be generalized to RL and also give a reduction from corruption to gap-dependent misspecification. Our reduction ensures that $\rho \le \frac{1}{d}$ is sufficient for $\sqrt{T}$ regret in linear MDPs. This is the first RL misspecification assumption beyond realizability that leads to $\sqrt{T}$ regret bound without any additional misspecification term.


There are many future works based on our paper. Firstly, the current $C$ bound for adversarial linear bandits is $d^3\sqrt{T} + d^{\frac{5}{2}}C$ while the lower bound is $d\sqrt{T} + dC$, leaving a gap to be addressed.  Secondly, getting tight $\sqrt{d}C_{\infty}$ bound beyond linear bandit (e.g. linear contextual bandits, general contextual bandits, and RL) is interesting. Thirdly, for gap-dependent misspecification, designing algorithms that achieve $\sqrt{T}$ regret for RL with $\rho \le \frac{1}{\sqrt{d}}$ could be a future direction. Moreover, finding other misspecification assumptions beyond realizability that lead to $\sqrt{T}$ regret without any additional term is also an interesting area to explore.


[Liu et al, 2023a] Liu, C., Yin, M., and Wang, Y.-X. (2023a). No-regret linear bandits beyond realizability. UAI.

---

### Decision · Program_Chairs · 2024-09-25

**Decision:**

Accept (poster)

**Comment:**

This paper resolved several important problems in the literature regarding bandit learning and corruption, including stochastic linear bandit and gap-dependent misspecification in linear bandits. It also considers the challenging setting of adversary linear bandit with corruption. After rebuttal, the authors successfully addressed the reviewers' concerns. I also find this paper solid, especially the connection between the two viewpoints of corruption.

One suggestion for the camera-ready version is to include the rebuttal's discussion (detailed explanation) on why the two viewpoints are equivalent in the appendix.